# Lifelong Open-Ended Probability Predictors

**Omid Madani** *omidmadani@yahoo.edu*

**Reviewed on OpenReview:** *https://openreview.net/forum?id=6430*

## Abstract

We advance probabilistic multiclass prediction on open-ended streams of items. In this setting, a predictor must emit items with probabilities, and adapt to significant non-stationarity, including new item appearances and frequency changes. The predictor is not given the set of items that it is to predict a priori, and moreover the totality of the items can grow unbounded: the space-limited predictor need only track the currently salient items and their probabilities. We develop Sparse Moving Average techniques (SMAs), including adaptations of sparse EMA as well as novel queue-based methods with dynamic per-item histories. For performance evaluation, to handle new items, we develop a bounded version of log-loss. Our findings, on a range of synthetic and real data streams, show that dynamic predictand-specific (per connection) parameters, such as learning rates, enhance both adaptation speed and stability.

> *"Occasionally, a new knot of significations is formed.. and our natural powers suddenly merge with a richer signification."* Merleau-Ponty (1945)

## 1 Introduction

The external world is inherently dynamic and productive, exhibiting various forms of non-stationarity under different timescales, from periodic cycles of day and night and seasonal changes (Ingold, 2000), to unforeseen phenomena including new ideas and inventions and abrupt physical and societal events. This constant change necessitates continual adaptation and renewed learning. As designers of such systems, we do not have control over such *external non-stationarity*, but hope that those changes are not too rapid and drastic that render learning ineffective (Schlimmer & Granger, 1986; Kuh et al., 1990; Helmbold & Long, 1991; Widmer & Kubát, 1996; Ditzler et al., 2015; Hinder et al., 2023). In particular, in this paper we foucs on online learning, such as in user interface agents and continual personalization, where the learner should predict with probabilities (Davison & Hirsh, 1997; Korvemaker & Greiner, 2000; Wester et al., 2011; Mazzetto, 2024): Probabilities are useful for down-stream utility-theoretic decision making. Furthermore, complex learning systems may also exhibit *internal*, in particular *developmental non-stationarity*, as their adapting subsystems come online and evolve over time. As designers, we have some control over this internal change, and an important challenge is balancing system adaptability and stability. Examples of such include systems following the constructivist theories of knowledge development (Piaget, 1970; von Glasersfeld, 1995; Fosnot, 2005), where perceptual or action patterns (schemata) are synthesized ('invented') by the learning system over time (Drescher, 1991; Thorisson, 2012; Strannegård et al., 2018; Madani, 2023). In Prediction Games in particular (Madani, 2007; 2023), *concepts*, *i.e.* explicit structures, serve as *both predictors and predictands* in the system, and the learned set of concepts grows over time. Thus, whether facing internal or external non-stationarity, the set of what is to be predicted may not be known in advance and may expand over time, *i.e.* open-endedness, and there can be many predictors (features) that do the predicting. The predictors should be space and time efficient, and converge fast when change occurs (sample efficient).

We advance online finite-memory multiclass learning methods, which we call **sparse moving averages (SMAs)**, for probability prediction under non-stationarity. We focus on learning and tracking probabilities above a minimum threshold, such as 0.001, acknowledging a trade-off between space constraints and the utility of learning lower probabilities. Key problem properties, and challenges, include:

- Lifelong and open-ended: The input stream is unbounded, and the set of items encountered is not given to the learner and the number of unique items observed can grow unbounded.

- Support non-stationarity as well as a wide probability range: Item probabilities change over time, requiring timely adaptation. Subject to the finite space constraints, strive to support several orders of magnitude of probability (*e.g.* $10^0$ down to $10^{-3}$).

- Stability vs. plasticity: Balancing fast adaptation, for items that are changing, while maintaining stable predictions (stable probabilities) for other predictands.

- Practical convergence: Achieving a balance between convergence speed and accuracy, rather than focusing on highly precise estimation in an asymptotic sense.

As a concrete example of the last point, if a new target event of interest occurs about once a day, then adequate accuracy, in the predicted probability, in a few days is strongly preferred over weeks or months. Sample efficiency is a necessity for autonomous continual learning. We summarize our contributions, to the best of our knowledge, and paper structure:

- A novel problem setting with a formalization for space-bounded predictors facing an open-ended input stream (Sect. 2): both the set of salient items and the proportion of the times that infrequent items are observed (noise for the predictor) can change over time. We use several evaluation scores, and develop a protocol of evaluation with a bounded version of log-loss for incomplete distributions handling new or infrequent items (Sect. 2.2.2), proving that the loss remains nearly proper.

- We develop several SMAs, focusing on three: sparse EMA (learning-rate based), Qs (counting based) and a hybrid DYAL (Sections 3.1-3.3). These techniques, implicitly or explicitly, need to address several subproblems such as change detection and when to allocate and deallocate parameters for items (non-stationarity and space-boundedness). We establish a bound on sparse EMA's convergence and derive a few theoretical properties of Qs. Variants of sparse EMA are used in past work, such as Davison & Hirsh (1997); Korvemaker & Greiner (2000); Madani et al. (2009a); Wester et al. (2011). With the novel idea of harmonic decaying of the learning rate (Sect. 3.1.2), we make a connection between sparse EMA and the counting technique such as Qs (counting and windowing are basic techniques for probability estimation, *e.g.* see Lewis (1998); Rosenfeld (2000); Bifet & Gavaldà (2007); Gama et al. (2014); Ditzler et al. (2015)). DYAL combines these ideas, enjoying the strengths of EMA (stability, memory efficiency) and Qs (fast implicit change detection). In particular, DYAL keeps a few extra parameters, in addition to a weight, *per predictand* (Fig. 9).

- A variety of experiments, in synthetic and real-world settings (Sections 5 and 6), shed light on the behavior of the different SMAs, providing evidence that DYAL often strikes a superior stability-plasticity tradeoff at a cost of a small, constant, overhead.

- Code, for a number of SMAs including the above three, and supporting functionality for evaluation and synthetic sequence generation, is provided.[1]

Sect. 7 discusses related work. The earlier version of this paper provides further material including more extensive technical properties and proofs (Madani, 2024).

## 2 Lifelong Open-Ended Prediction

Our setting is online multiclass probabilistic prediction under non-stationarity. Time $t$ is discrete, $t = 1, 2, 3, \cdots$. A predictor *processes* a stream of items (observations): at each time point, it predicts (outputting a semi-distribution, Sect. 2.1), then observes and updates (learns). Exactly one discrete item occurs and is observed at $t$. We use the parenthesized superscript notation for the variable corresponding to the observation at $t$, *i.e.* $o^{(t)}$, and similarly for other sequenced objects, for instance an estimated probability $\hat{p}$ at time $t$

---

[1]At Github: https://github.com/omadaniTet/sparse-moving-averages

| |
|---|
| $[o], [o]_1^N$, etc. : infinite & finite sequences, $[o]_1^N := [o^{(1)}, \cdots, o^{(N)}]$, $o^{(t)}$ is the item observed at $t$. |
| SDs and DIs: probability $\underline{s}$emi-$\underline{di}$stributions (SDs) and $\underline{di}$stributions (DIs) (Sect. 2.1). |
| SMA: a Sparse Moving Average predictor, *e.g.* sparse EMA and Qs predictors (Sections 3.1-3.3). |
| EMA : The Exponentiated Moving Average (Sect. 3.1); SMAs are sparse relatives. |
| $\mathcal{P}, \mathcal{W}, \mathcal{W}_1, \mathcal{W}_2$, etc. : $\mathcal{P}$ is used for a true underlying SD, $\mathcal{W}$ and $\mathcal{W}_i$ are (SMA's) estimates. |
| $[\mathcal{W}]_1^N$: first $N$ outputs of an SMA (predictor). Each $\mathcal{W}^{(t)}$ is a SD (or converted to one). |
| $\sup(\mathcal{W})$: the support set of SD $\mathcal{W}$. Every SD has finite support (subset of all items $\mathcal{I}$). |
| $a(\mathcal{W}), u(\mathcal{W})$: $a(\mathcal{W})$ the $\underline{a}$llocated probability mass, $\sum_{i \in \sup(\mathcal{W})} \mathcal{W}(i)$, $u(\mathcal{W}) := 1 - a(\mathcal{W})$ ($\underline{u}$nallocated) |
| $p^*$ : For a binary event, observation $o \in \{0, 1\}$, the true probability of the positive, 1, outcome. |
| positive outcome, 1 : whenever the target item of interest is observed (0 otherwise). |
| noise item: an item not seen recently in the input stream sufficiently often (Sect. 2.1.1). |
| $p_{min}$ : used for filtering & scaling, and bounding log-loss on noise items (Sect. 2.2.2). |
| $p_{min}$ can be interpreted as the minimum probability supported by the SMA. |
| $c_{NS}$: (for evaluating) count-threshold used by a practical noise-marker algorithm (2.2.2). |
| $O_{min}$ : in synthetic experiments, the minimum number of positive observations of a salient item before it can change probability (a knob on the rate of change, Sections 2.1.1 & 5.2). |
| $\beta, \beta_{min}, \beta_{max}, \ldots$ : learning rates (for sparse EMA variants), $\beta \in [0, 1]$, $\beta_{min}$ is minimum allowed, etc. |

Table 1: The main notation and terminology, abbreviations and parameters.

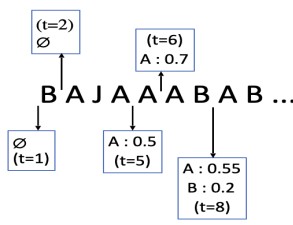

(a) A stream of observations, left to right (items: $B$, $A$, $J$, ...), and prediction outputs, the boxes, by a hypothetical predictor, shown at a few time points.

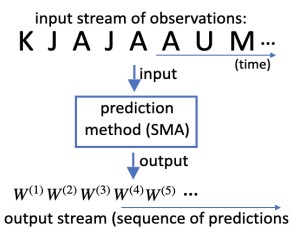

(b) An SMA emits probabilities, in effect converting an item stream (observations) into a stream of (moving) SDs, $[\mathcal{W}]$, *i.e.* the predictions sequence.

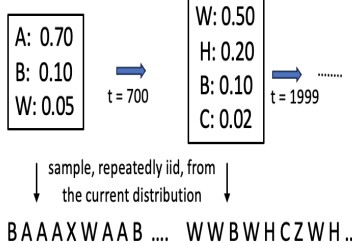

(c) A (hidden) categorical distribution generates the (synthetic) data stream, in an idealized setting (IID draws), but also itself changes every so often (here, at time $t = 700, 1999$, etc).

Figure 1: (a) An example sequence of items together with outputs of a predictor, shown for a few of the time points. The sequence is observed from left to right, thus at time $t = 1$, item $B$ is observed ($o^{(1)} = B$, $o^{(3)} = J$, etc). The predictor emits its probabilities before observing and updating. In this example, at times $t \leq 4$, nothing is predicted. At $t = 8$, the SD $\mathcal{W}^{(8)}$ is predicted, where $\mathcal{W}^{(8)} = \{A:0.55, B:0.2\}$ (Sect. 2.1). (b) An SMA, in effect, converts a stream of item observations, $[o]$, to a stream of predictions, $[\mathcal{W}]$, or $[o]_1^N \to [\mathcal{W}]_1^N$. (c) We imagine that the input sequence is generated by a SD $\mathcal{P}$ that changes from time to time (item probabilities may go up or down, including the extreme cases of removal, $A$, and insertion, $H$ and $C$, in the above) (Sect. 2.1.1).

is denoted $\hat{p}^{(t)}$. An item is identified by an integer id in the various data structures (sequences, maps, ...). Open-endedness: an item can be viewed as belonging to an infinite unordered (*i.e.* categorical and nominal) set (*e.g.* the infinite set of finite strings). We use letters $A$, $B$, $C$, $\cdots$ and also integers $0, 1, 2, 3, \cdots$ (with no ordering implied) to refer to items in examples. We use the words "sequence" and "stream" interchangeably. A sequence is denoted by the brackets notation $[\ ]$, thus $[o]_1^N$ denotes a sequence of $N$ observations: $[o^{(1)}, o^{(2)}, ..., o^{(N)}]$ (also shown as $[o^{(1)}o^{(2)} \cdots o^{(N)}]$). Fig. 1(a) gives an example sequence. Table 1 provides a summary of the main notations and terminology used.

## 2.1 Semi-Distributions (SDs)

A probability is a real number in $[0, 1]$. A prediction output is a *semi-distribution*:[2] A (categorical) semi-distribution (SD) $\mathcal{W}$, is a probability-valued function defined over an infinite item set $\mathcal{I} = \{0, 1, 2, \cdots\}$, such that $\forall i, \mathcal{W}(i) \in [0, 1]$, and $\sum_{i \in \mathcal{I}} \mathcal{W}(i) \leq 1.0$. The prediction output at $t$ is denoted $\mathcal{W}^{(t)}$. We can imagine that a prediction method is any online technique for converting an item sequence, $[o]_1^N$, one observation at a time, into a sequence of predictions $[\mathcal{W}]_1^N$ (of equal length $N$) (Fig. 1(b)). Prediction occurs before updating: $\mathcal{W}^{(t)}$ is output at time $t$ before observing $o^{(t)}$. We use $\mathcal{P}$ to refer to an underlying or actual SD generating the observations in the ideal setting and synthetic experiments (see next), and use $\mathcal{W}$ to denote estimates and prediction outputs (often strict SDs). In general, under non-stationarity, $\mathcal{W}^{(t)}$ form *moving* (changing) SDs.

Let $\mathrm{a}(\mathcal{W}) := \sum_{i \in \mathcal{I}} \mathcal{W}(i)$ (the allocated probability mass of $\mathcal{W}$), and let $\mathrm{u}(\mathcal{W}) := 1 - \mathrm{a}(\mathcal{W})$ (the unallocated or the free mass). A *probability distribution* (DI) $\mathcal{W}$ is a special case of a SD, where $\mathrm{a}(\mathcal{W}) = 1$. When $\mathrm{a}(\mathcal{W}) > 0$ we say $\mathcal{W}$ is non-empty, and when $0 < \mathrm{a}(\mathcal{W}) < 1$, we call $\mathcal{W}$ a *strict* SD. The *support*, denoted $\sup(\mathcal{W})$, is the set of items $i$ with $\mathcal{W}(i) > 0$. We simply use $|\mathcal{W}|$ for $|\sup(\mathcal{W})|$ (the number of positive entries in the map $\mathcal{W}$). The support is finite in this paper: $|\sup(\mathcal{W}^{(t)})|$ is up to 100s to 1000s in our experiments depending on the predictor's space limits (or algorithmic parameters, *e.g.* see Sect. 3.2.1). Let $\min(\mathcal{W}) := \min_{i \in \sup(\mathcal{W}(i))} \mathcal{W}(i)$, and defined as 0 when $\mathcal{W}$ is empty. A map data structure is used to implement a SD, pruned periodically. $\mathcal{W}(i)$ is 0 if item $i$ is not in the map.

As an example SD, for $\mathcal{P} = \{0{:}0.5,\ 1{:}0.2\}$, $\mathcal{P}$ has support of size 2, corresponding to a binary event, and item 0 has probability 0.5, and 1 has probability 0.2, $\min(\mathcal{P}) = 0.2$, $\mathrm{a}(\mathcal{P}) = 0.7$, and $\mathrm{u}(\mathcal{P}) = 0.3$, so $\mathcal{P}$ is a strict SD in this example.

### 2.1.1 Idealized Streams and Salience

We have the following idealized non-stationarity in mind for algorithm design. The input stream is the concatenation of *stable subsequences*, each generated in a stationary IID manner (independent and identically distributed), sampling from an underlying (semi-)distribution (Fig. 1(c)).[3] The underlying distributions change over time, but each stable period is assumed to be sufficiently long for learning those item probabilities that have changed, thus we assume *per-item* stability. Concretely, in synthetic experiments, we use a parameter, $O_{min}$, to control the degree of non-stationarity, ensuring a (*salient*) item (described next) occurs $\geq O_{min}$ times (*e.g.* $O_{min} = 5$) before its probability changes. This allows us to evaluate prediction algorithms under controlled, changing distributions, reflecting real-world scenarios where periods of stability are followed by change. But real-world streams can also deviate from this idealization in various ways (*e.g.* periodicities) and it is important to evaluate on non-synthesized sequences as well.

Our predictors are space-bounded and cannot learn probabilities that are too small. We evaluate the probabilities above a minimum threshold $p_{min}$ (*e.g.* $p_{min} \in \{0.01, 0.001\}$ in our experiments). Items with probability above $p_{min}$ are *salient*, otherwise non-salient or noise. Good predictors should quickly detect and learn new salient items while, in effect, ignoring noise (infrequent items).[4] For sequence generation using SD $\mathcal{P}$ (*e.g.* in our synthetic experiments), non-salient items are drawn with probability $\mathrm{u}(\mathcal{P}) = 1 - \mathrm{a}(\mathcal{P})$, often with a unique noise-item id: the item will appear only once in the sequence.

A stationary binary sequence, a special case useful in theoretical and empirical analyses, can be generated via IID drawing from a DI $\mathcal{P} = \{\ 1{:}\ p^*,\ 0{:}\ 1 - p^*\ \}$, where $p^* > 0$ is the *true or target* probability to estimate well via a predictor that processes the binary sequence. A binary sequence also arises whenever we focus on a single item in a given original sequence containing different items.[5]

---

[2] We are borrowing the naming used in (Dupont et al., 2005). Other names include sub-distribution and partial distribution.

[3] The concatenation of stable subsequences is similar to the model of Plasse & Adams (2019) who studied change detection for a multinomial (whose item-set is known).

[4] The definition of noise is relative to the finite space of the predictor.

[5] The original sequence $[o]$ is converted to a binary $[o_b]$, $[o] \to [o_b]$, in the following manner, say for item $A$: if $o^{(t)} = A$ then $o_b^{(t)} = 1$, and $o_b^{(t)} = 0$ otherwise (or $o_b^{(t)} = \left[\left[o^{(t)} = A\right]\right]$, where $[[x]]$ denotes the Iverson bracket on the Boolean condition $x$).

### 2.1.2 Two Example Scenarios (Streams)

Even the ideal generation model described above contains rich challenging possibilities. We describe two simple scenarios below.

**Stable-Changing.** One item $A$ is stable at a constant probability of say 0.5, while other (salient) items change (remaining stable for only a relatively short time span). For instance, there could be repeated switching between two underlying distributions, $\mathcal{P}_1$ and $\mathcal{P}_2$, $\mathcal{P}_1 \leftrightarrows \mathcal{P}_2$, where $\mathcal{P}_1 = \{A{:}0.5, B{:}0.5, C{:}0\}$ and $\mathcal{P}_2 = \{A{:}0.5, B{:}0, C{:}0.5\}$ (so items $B$ and $C$ swap probabilities from time to time). Sect. 5.1 explores a similar scenario involving oscillations between two probability values for the same item.

**Noise-Portion Change.** Note that the noise portion of the input stream, *i.e.* the proportion of the items not predictable by the learner, can range anywhere from 0 to 1, and can change too. Say a predictor has space for keeping track of only a 100 items, but for some time, *e.g.* till $t_1 = 10^7$, all items in the stream occur with probability below 0.001 say. Afterwards, for say next 100 time points till time $t_2$, a single item $A$ becomes salient with $\mathcal{P}(A) = 0.7$. This period is followed by another one where $B$ becomes salient, at $\mathcal{P}(B) = 0.4$, and $A$ dropping to 0, till $t_3$. The noise portion of the stream, for this predictor, is then 1.0 for $t \leq t_1$, then goes down to 0.3 for $t \in (t_1, t_2]$, and then goes up, to 0.6, for $t \in (t_2, t_3]$ (the transitions in the modeled item-probabilities, for a good predictor, are $\{\} \rightarrow \{A{:}0.7\} \rightarrow \{B{:}0.4\}$).

## 2.2 Evaluating Probabilistic Predictors

A prediction method, in effect, converts an observed sequence $[o]_1^N$, to a sequence of predictions, SDs $[\mathcal{W}]_1^N$ (Fig. 1(b) and Sect. 2.1). Here, we develop and discuss methods for evaluating $[\mathcal{W}]_1^N$ for empirical comparison of different SMAs.

### 2.2.1 When True Probabilities are Known: Deviation Rates

Whenever the probability of an item $p^*$ is not changing (in an stable period, 2.1.1), the predictor's probability outputs for that item should quickly converge to $p^*$. We first consider synthetic streams, *i.e.* when we know the underlying probabilities. Let $[\hat{p}]_1^N$ denote the sequence of output probabilities for a specific item. For a choice of a deviation bound $d > 1$, eg $d = 2$, we define the *deviation-rate* $dev([\hat{p}]_1^N, p^*, d)$ as the following sequence average:

$$dev([\hat{p}]_1^N, p^*, d) := \frac{1}{N} \sum_{t=1}^{N} deviates(\hat{p}^{(t)}, p^*, d) \quad \text{(the deviation rate, with } p^* > 0) \tag{1}$$

$$deviates(\hat{p}, p^*, d) := \left[\!\left[ \hat{p} = 0, \text{ or (otherwise), } \max\left(\frac{p^*}{\hat{p}}, \frac{\hat{p}}{p^*}\right) > d \right]\!\right] \quad \text{(it is 0 or 1)} \tag{2}$$

where $[\![x]\!]$ is the Iverson bracket (1 when condition $x$ is true, otherwise 0). Note that when $\hat{p}$ is 0, the condition is true and $deviates(0, p^*, d)$ is 1 for any $d$. The smaller the deviation-rate the better, and we report rates for a few choices of $d$ (Sect. 5). More generally, we track multiple items, and we report two variants: at any time $t$, when the observed item's estimated probability deviates, and when *any* (salient) item in $\mathcal{P}$ suffers too large of a deviation (Sect. 5.2). Deviation rates are easy to understand (interpretable), but often in real-world settings, true probabilities are not available.

### 2.2.2 Unknown True Probabilities: Adapting log-loss

There exist much work in diverse domains such as meteorology and finance addressing the challenge of unknown underlying probabilities. In particular, the concept of *proper scoring rules* has been developed to encourage *propriety*, *i.e.* reliable probability forecasts (Brier, 1950; Good, 1952; Toda, 1963; Winkler, 1969; Gneiting et al., 2007; Filho et al., 2021; Tyralis & Papacharalampous, 2022). A proper rule would lead to the best possible score (lowest if defined as a cost, as we do here) if the technique's probability outputs matched the true probabilities (assumed to exist). In particular, the logarithmic loss (log-loss) incurred by a candidate SD $\mathcal{W}$, given an underlying DI $\mathcal{P}$, is proper:

$$\text{LogLoss}(\mathcal{W}|\mathcal{P}) := \mathbb{E}_{o \sim \mathcal{P}} - \ln(\mathcal{W}(o)) \quad \text{(logarithmic loss of } \mathcal{W} \text{ given } \mathcal{P}) \tag{3}$$

**FC**($\mathcal{W}$) // Filter & cap $\mathcal{W}$.
  // $\mathcal{W}$ is an item to probability map.
  // Parameter: $p_{min} \in [0, 1)$
  $\mathcal{W}' \leftarrow$ ScaleDrop($\mathcal{W}$, 1.0, $p_{min}$) // Filter
  If a($\mathcal{W}'$) $\leq 1.0 - p_{min}$: // Already capped?
    Return $\mathcal{W}'$ // Nothing left to do.
  $\alpha \leftarrow \frac{1-p_{min}}{a(\mathcal{W}')}$ // Scale down by $\alpha$.
  Return ScaleDrop($\mathcal{W}'$, $\alpha$, $p_{min}$)

**ScaleDrop**($\mathcal{W}$, $\alpha$, $p_{min}$) // Filter and scale.
  $scaled \leftarrow \{\}$
  For item $i$, probability $prob$ in $\mathcal{W}$:
    $prob \leftarrow \alpha * prob$
    If $prob \geq p_{min}$ :   // Keep the salient.
      $scaled[i] \leftarrow prob$
  Return $scaled$

(a) Filtering & capping the predictions $\mathcal{W}$.

**loglossRuleNS**($o$, $\mathcal{W}$, $markedNS$)
  // Parameters: $p_{min}$. The current
  // observation is $o$, and $\mathcal{W}$ is the
  // predictions (an item to probability map).
  $\mathcal{W}' \leftarrow FC(\mathcal{W})$ // filter and cap $\mathcal{W}$.
  $prob \leftarrow \mathcal{W}'$.get($o$, 0.0)
  If $prob \geq p_{min}$: // $o \in$ sup($\mathcal{W}'$)?
    Return $-\ln(prob)$ // plain log-loss.
  // $o \notin$ sup($\mathcal{W}'$): the predictor treats $o$ as
  // a new/noise item. The noise-marker
  If not $markedNS$: // referee disagrees?
    Return $-\ln(p_{min})$ // incur maximum loss.
  // They both agree $o$ is noise, so use the
  // unallocated mass (potentially lower loss).
  $p_{noise} \leftarrow$ u($\mathcal{W}'$) // note: u($\mathcal{W}'$) $\geq p_{min}$
  Return $-log(p_{noise})$

(b) When computing log-loss, handling new or noise items (bounded loss when $p_{min} > 0$).

Figure 2: For evaluating the output probabilities: (a) CapAndFilter() is applied to the output of any predictor, at every time point $t$, before evaluation. It performs filtering (dropping small probabilities below $p_{min}$) and, if necessary, explicit capping, *i.e.* normalizing or scaling down, meaning that the final output will be a SD $\mathcal{W}'$, where a($\mathcal{W}'$) $\leq 1 - p_{min}$ (or u($\mathcal{W}'$) $\geq p_{min}$). $p_{min} = 0.01$ in our experiments. (b) Scoring via bounded log-loss, handling noise items: the maximum incurred loss has the ceiling $-\ln(p_{min})$.

The loss is the expected value of the natural logarithm $-\ln(\mathcal{W}(o))$, where the expectation is taken with respect to (wrt) the true DI $\mathcal{P}$. This loss remains sensitive (useful) for assessing smaller probabilities, unlike quadratic (Brier) loss, another commonly used proper loss. But plain log-loss is infinite on 0 probabilities and more generally, large losses on infrequent or new items can dominate overall log-loss making it uninformative for comparing predictors. We also want to preserve propriety as much as possible (including adequately estimating the noise portion of the stream, Sect. 2.1.2). We next explain how we adapt log-loss to handle such cases. Our evaluation protocol has 3 components:

1. Ensuring that the output map of any predictor is a SD $\mathcal{W}$ such that a($\mathcal{W}$) $\leq 1 - p_{min}$, with $p_{min} > 0$ ($p_{min} = 0.01$ in our experiments), thus some mass, u($\mathcal{W}$) $\geq p_{min}$, is left for observing a noise item (via filtering and normalizing, the function FC() in Fig. 2(a)).

2. Use of a *simple noise-marker*, a *"referee"*, for noise judgement, such as in Fig. 3.

3. Specification of how to score in the various cases, *e.g.* whether or not an item is marked noise by the referee (Fig. 2(b), a bounded version of log-loss).

Fig. 2(a) shows the pseudocode for filtering and capping, FC(), applied to the output $\mathcal{W}$ of any SMA, before we evaluate it. Letting $\mathcal{W}' := FC(\mathcal{W})$, we have a($\mathcal{W}'$) $\leq 1 - p_{min}$ and min($\mathcal{W}'$) $\geq p_{min}$.[6] Once we have a noise-marker and the FC() function, given any observation $o^{(t)}$, and the capped predictions $\mathcal{W}^{(t)}$ (where we have applied FC() to get the SD $\mathcal{W}^{(t)}$) and given the noise status of $o^{(t)}$ (via the noise-marker), we evaluate $\mathcal{W}^{(t)}$ using loglossRuleNS(), and take the average over the sequence:

$$\text{AvgLogLossNS}([\mathcal{W}]_1^N, [o]_1^N) = \frac{1}{N} \sum_{t=1}^{N} \text{loglossRuleNS}(o^{(t)}, \mathcal{W}^{(t)}, \textbf{isNS}(o^{(t)})) \tag{4}$$

---

[6]One can make a distinction between the minimum probability modeled and the minimum mass kept for noise items (in FC()). When evaluating SMAs, it can be argued they should be the same (Madani, 2024), and we use one parameter $p_{min}$.

**isNS**($o$) // whether observation $o$ is noise or not.
   // Parameters: $c_{NS}$ (noise count threshold).
   $flaggedNS \leftarrow recentFrqMap.get(o, 0) \leq c_{NS}$
   // Increment count of $o$.
   Increment($o, recentFrqMap$)
   Return $flaggedNS$ // Return the 0 or 1 decision.

Figure 3: A simple noise-marker, a referee for comparing different SMAs, to mark whether an item is noise, via a count map, and unlimited history in this picture.

The scoring rule loglossRuleNS($o, \mathcal{W}, \textbf{isNS}(o)$) checks if there is a probability $p, p > 0$ assigned to $o$, $p = \mathcal{W}(o)$. If so, we must have $p \geq p_{min}$ (from the definition of FC()), and the loss is $-\ln(p)$. Otherwise, if $o$ is also marked noise (there is agreement), the loss is $\ln(\text{u}(\mathcal{W}))$. From the definition of FC(), $\text{u}(\mathcal{W}) = 1 - \text{a}(\mathcal{W}) \geq p_{min}$. Finally, if $o$ is not marked noise, the maximum possible loss, $-\ln(p_{min})$, is incurred.

There are two parameters for the noise-marker, the size of the history kept (window size) and the count threshold $c_{NS}$. For simplicity, by default no bound on length of the history is imposed in our experiments (unlimited window), though we report on finite referee window sizes too (*e.g.* counts of items seen limited to the last 500 time points). By default, we set $c_{NS} = 2$, meaning an item is marked noise iff it has been seen $k \leq 2$ so far in the sequence (so the 3rd occurrence is marked noise too): therefore, to incur a loss lower than the maximum $-\ln(p_{min})$, a predictor should provide a probability above $p_{min}$ for the 4th and subsequent occurrences.[7] When comparing different predictors, via AvgLogLossNS(), we compare on the same exact sequences (with an identical noise-marker). We summarize the technical properties of LogLossNS (near propriety), in Appendix C and briefly discuss alternatives we considered in Sect. C.2. A basic question is whether the evaluation protocol can be further simplified.

### 2.2.3 Evaluating a Space-Bounded Predictor under Nonstationarity

We can use the same empirical evaluation log-loss formula of AvgLogLossNS() (Eq. 4) in the idealized non-stationary case, and as long as each subsequence is sufficiently long, a fast converging prediction method should do well. In synthetic experiments, we try different minimum frequency requirements, $O_{min}$: an item needs to occur $O_{min}$ times in a sequence before its probability can be changed. The lower the underlying $p^*$ the more time points required before $p^*$ is changed, as we expect one positive observation every $\frac{1}{p^*}$ time points. Note that if $O_{min}$ is set too low, *e.g.* below 5, this would not allow sufficient time for learning a probability well, and the underlying $p^*$ loses its meaning. We want a predictor that tracks changes in salient probabilities well: a high probability item or items (*e.g.* above 0.1) may change in probability (*e.g.* disappear), while a low probability item (*e.g.* 0.03) may remain stable for a long time, or vice versa (see Sect. 2.1.2). Good predictors should track such too (and, for example, not allocate all their mass onto the salient items). A good evaluation technique should score honest and adaptive predictors well (see Appendix C.1).

With real-world sequences, a variety of phenomena such as periodicity and other dependencies can violate the IID assumptions, and it is an empirical question whether, on real sequences, the predictors compare as anticipated[8] based on their various, presumed or expected, strengths and weaknesses stemming from consideration of the ideal setting of Sect. 2.1.1. This underscores the importance of empirical experiments on different real-world sequences.

---

[7]As another example, if $c_{NS} = 0$, the predictor should provide a probability on the 2nd occurrence of an item (otherwise it incurs the maximum loss $-\ln(p_{min})$).

[8]Note the caveat that propriety is developed under IID assumptions, and we utilize propriety for non-stationary streams composed of stationary subsequences (the assumption of stability per salient item, Sect. 2.1.1). If a sequence substantially violates the IID assumptions, it is possible that basing comparisons on proper losses could be misleading.

# 3 Sparse Moving Average (SMA) Predictors

A moving average in its most basic form tracks the changes of a scalar value by keeping a recency-biased average. Here the observation at time t can be viewed as a sparse (possibly infinite) vector of 0s with a single 1 at the dimension equal to the id of the observed item (Madani & Huang, 2008; Wei, 2012), and the techniques developed and studied here are different ways of explicitly keeping several moving averages, i.e. estimated proportions of the items deemed salient, to predict the future probabilistically. The number of proportions that are tracked is kept under a limit for efficiency, i.e. kept sparse. Note that there are two major motivations for having a limited (sparse) memory: one stemming from resource boundedness (space/time efficiency), and another for faster adaptation to non-stationarities, or sample efficiency for accurate prediction (which can sometimes trade off). We next present and develop three SMAs: one based on learning rates, another using counting (and queuing based), and, in order to more fully address the challenges of open-ended non-stationarity, a combination.

## 3.1 The Sparse EMA Predictor

The exponentiated moving average (EMA) update is a convex combination of the present (scalar) observation with the past (moving) average (effectively exponentially decaying the influence of the past values). The EMA update form has been found beneficial for online incremental learning, adapting to change and providing probabilities to predict discrete items, such as actions in adaptive user-interface agents and in large-scale classification and speculative execution (Davison & Hirsh, 1997; Korvemaker & Greiner, 2000; Madani & Huang, 2008; Wester et al., 2011).[9] Here we further develop and analyze sparse EMA, focusing on its *probability prediction* capability and convergence speed. In particular, the analysis and theorem 1 (next) and the technique of decaying the rate and its properties, Sect. 3.1.2 (and used in Sect. 3.3 as well), are new to the best of our knowledge.

Fig. 4 presents pseudo code for sparse EMA. The SMA keeps a map $\mathcal{W}$, of item to probability. Initially, at $t = 1$, $\mathcal{W}$ is empty. Each entry in $\mathcal{W}$ can be viewed as a connection, or a prediction relation, a weighted directed edge, from the predictor to a predictand, where the weight is the current probability estimate for the predictand. Prediction is straightforward: output the map's key-value (item-probability) pairs. The update can be broken into two phases, a **weakening** followed by a **boost** (or strengthening) phase, as in Fig. 5. This picture is useful when we extend EMA (Sect. 3.3). Predicting and updating take the same $O(|\mathcal{W}|)$ time, and the map $\mathcal{W}$ can be pruned periodically (see also Sect. 3.2.2 and 3.3.3). We next describe probability and SD estimation and accompanied challenges, in particular the speed of convergence *vs.* variance, and handling change (versions of plasticity *vs.* stability trade-offs (Abraham & Robins, 2005; Mohri & Medina, 2012; Mermillod et al., 2013)),[10] motivating extensions (Sect. 3.1.2 and 3.3).

### 3.1.1 Convergence

The sparse EMA update, with $\beta \in [0, 1]$, preserves the SD property: when the probability map before the update corresponds to a SD $\mathcal{W}^{(t)}$, $\mathcal{W}^{(t+1)}$ (the map after the update) is a SD too. Furthermore, EMA enjoys several convergence properties under appropriate conditions, *e.g.* the sum of the edge weights (map entries), a($\mathcal{W}^{(t)}$), increases, converging to 1.0 (*i.e.* to a DI), and an EMA update can be viewed as following the gradient of quadratic loss (Madani & Huang, 2008). Here, we are interested in convergence speed of the map weights as individual probability estimates.

In the stationary IID setting (Sect. 2.1.1, the stability period), we show that the probability estimates of EMA converge, probabilistically, to the vicinity of the true probabilities. We focus on the EMA estimates for one item or predictand, call it $A$: at each time point $t$, a **positive update** occurs when $A$ is observed, and otherwise it is a **negative update** (a weakening), which leads to the stationary binary setting (Sect.

---

[9]EMA (also, EWMA) and other moving averages can be viewed as a type of filter in signal theory and a (time) convolution (Contributors, 2024), and find a variety of applications, such as in time-series analysis and forecasting (*e.g.* (Weiß, 2018; Maiti & Biswas, 2015)), and in reinforcement learning for value function updates (in temporal difference learning, (Sutton & Barto, 1998)).

[10]In computational learning theory a similar trade-off has been termed drift error *vs.* statistical (variance) error (Mohri & Medina, 2012; Mazzetto, 2024).

**EmaUpdate**($o$) // latest observation $o$.
    // *EmaMap* is item to probability, learning rate $\beta \in (0, 1]$.
    // Other param: *doHarmonicDecay* flag.
    For each item $i$ in *EmaMap*:
        $EmaMap[i] \leftarrow (1 - \beta) * EmaMap[i]$ // Weaken.
    // Strengthen edge to $o$ (insert edge if not there).
    $EmaMap[o] \leftarrow EmaMap[o] + \beta$ // Boost.
    If *doHarmonicDecay*: // reduce rate?
        $\beta \leftarrow DecayRate(\beta)$ // see (b) and Sect. 3.1.2

(a) Plain EMA updating.

**DecayRate**($\beta$)
    // Other parameters: $\beta_{min} \in (0, 1)$, is
    // the minimum allowed learning rate.
    // $\beta_{max} \in (0, 1]$, is the maximum and
    // the initial learning rate for EMA with
    // harmonic decay.
    $\beta \leftarrow \frac{1}{\frac{1}{\beta}+1.0}$ // harmonic decay.
    Return $\max(\beta, \beta_{min})$

(b) Harmonic decay of rate.

Figure 4: Pseudo code of sparse EMA: (a) with a single learning-rate $\beta$, either fixed ("static" EMA), or (b) decayed with a harmonic schedule down to a minimum $\beta_{min}$ ("harmonic" EMA, Sect. 3.1.2). An edge (entry in map $\mathcal{W}$) is created if it doesn't already exist (*e.g.* there are no edges at $t = 1$). See also Fig. 5.

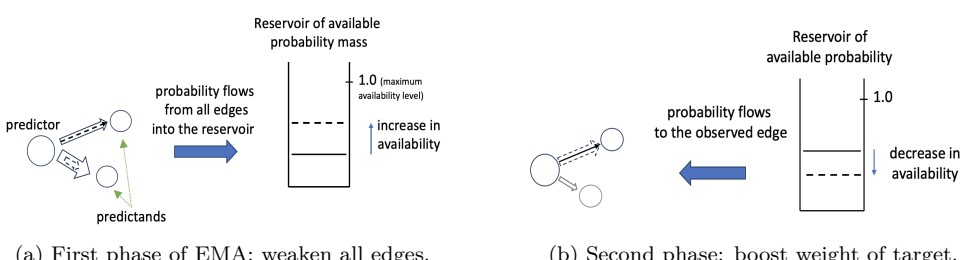

(a) First phase of EMA: weaken all edges.    (b) Second phase: boost weight of target.

Figure 5: An EMA update can be viewed as having two phases: *weaken* then *boost*. In the first weakening phase, all existing edges from the predictor to the predictands, entries in the map $\mathcal{W}$, are weakened. This can be viewed as probability flowing from the edges to the implicit reservoir of (unused or available) probability mass ($u(\mathcal{W})$). In the 2nd boost phase, the edge to the observed item is strengthened.

2.1.1). Let $p^*$ denote the target probability. The estimates of EMA for item $A$, denoted $\hat{p}^{(t)}$, form a random walk. While often, *i.e.* at many time points, it can be more likely that $\hat{p}^{(t)}$ moves away from $p^*$, such as when $\hat{p} \leq p^* < 0.5$, when $\hat{p}$ does move closer to $p^*$ (upon a positive observation), the gap reduces by a relatively large amount. The following property, in particular, helps us show that the expected gap, $|p^* - \hat{p}|$, shrinks with an EMA update, when $\hat{p}$ is not too close to $p^*$. It also connects EMA's random walk, the evolution of the gaps for instance, to discrete-time martingales, fundamental to the analysis of probability and stochastic processes (*e.g.* the gap is a supermartingale when $\hat{p}^{(t)} < p^*$) (Williams, 1991; Shafer & Vovk, 2001).

**Lemma 1.** *EMA's movements,* i.e. *changes in the estimate* $\hat{p}^{(t)}$*, satisfy the following properties, where* $\beta \in [0, 1]$*:*

1. *Maximum movement, or step size, no more than* $\beta$*:* $\forall t, |\hat{p}^{(t+1)} - \hat{p}^{(t)}| \leq \beta$*.*

2. *Expected movement is toward* $p^*$*: Let* $\Delta^{(t)} := p^* - \hat{p}^{(t)}$*. Then,* $\mathbb{E}(\Delta^{(t+1)}|\hat{p}^{(t)} = p) = (1 - \beta)(p^* - p) = (1 - \beta)\Delta^{(t)}$*.*

3. *Minimum expected progress size: With* $\delta^{(t)} := |\Delta^{(t)}| - |\Delta^{(t+1)}|$*,* $\mathbb{E}(\delta^{(t)}) \geq \beta^2$ *whenever* $|\Delta^{(t)}| \geq \beta$ *(*i.e. *whenever* $\hat{p}$ *is not very near* $p^*$*).*

The lemma is established by writing down the expressions, for an EMA update and the gap expectation, and simplifying (see Appendix A). It follows that $\mathbb{E}(\hat{p}^{(t+1)}|\hat{p}^{(t)} = p^*) = p^*$ (or $\mathbb{E}(\Delta^{(t+1)}|\Delta^{(t)} = 0) = 0$, and the *expected direction* of movement is always toward $p^*$. The maximum movement property implies that when $\hat{p}$ is sufficiently far, at least $\beta$ away from $p^*$, the sign of $\Delta^{(t)}$ does not change, and property 2 implies that the gap is indeed reduced in expectation (property 3), which we can use to show probabilistic

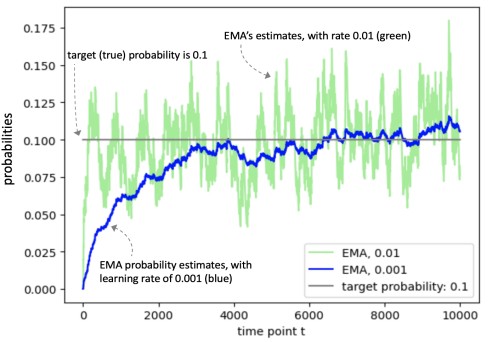 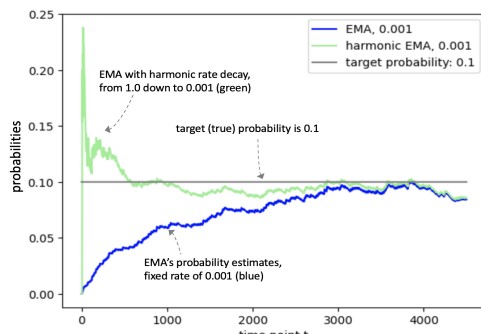

Figure 6: Given a single binary sequence, 10k long, with target $p^* = 0.1$ (the true probability of observing 1), convergence of the probability estimates, $\hat{p}^{(t)}$, by (sparse) EMA variants: (a) for the static or fixed-rate version of EMA, under two different learning rates, $\beta = 0.01$ and $\beta = 0.001$. A higher rate, $\beta = 0.01$, can lead to faster convergence, but also causes high variance. (b) Convergence is faster with harmonic decay.

convergence:

**Theorem 1.** *Sparse EMA, with a fixed rate of $\beta \in (0, 1]$, has an expected first-visit time bounded by $O(\beta^{-2})$ to within the band $p^* \pm \beta$. The required number of updates, for first-visit time, is lower bounded below by $\Omega(\beta^{-1})$.*

The proof, presented in Appendix A, works by using property 3 that expected progress toward $p^*$ is at least $\beta^2$ while our random walker $\hat{p}^{(t)}$ is outside the band ($|p^* - \hat{p}^{(t)}| > \beta$). It would be good to tighten the gap between the lower and upper bounds. We leave further characterizing the random walk, to future work.

The smaller the rate $\beta$ the longer it takes for $\hat{p}$ to get close to $p^*$. A higher rate $\beta$, *e.g.* 0.01, require 10 times fewer observations than 0.001, or leads to faster convergence, for higher target probabilities (with $p^* \gg \beta$). On the other hand, once sufficiently near the target probability $p^*$, we desire a low rate for EMA, to keep estimating the probability well (Fig. 6), or a high rate would causes unwanted jitter or variance. When we want to learn target probabilities in a diverse range, such as $[0.001, 1.0]$, we need to use a low rate to make sure smaller probabilities are learned sufficiently well. So, with a fixed rate (static EMA), accuracy on small probabilities incurs the cost of slow convergence on large probabilities, motivating consideration of changing the rate.

### 3.1.2 Harmonic Decay and a Connection to Counting

A variant of EMA, which we will refer to as *harmonic EMA*, has a rate decaying over time with each update, in a *harmonic* manner (Fig. 4(b)). The rate $\beta^{(t)}$ starts at a high value $\beta^{(1)} = \beta_{max}$ (*e.g.* $\beta_{max} = 1.0$), and is reduced gradually with each (positive or negative) update, via:

$$\beta^{(t+1)} \leftarrow \left(\frac{1}{\beta^{(t)}} + 1\right)^{-1} \quad \text{(harmonic decay of the learning rate: a double reciprocal)}$$

For instance, with $\beta^{(1)} = 1$, then $\beta^{(2)} = \frac{1}{2}$, and in general $\beta^{(t)} = \frac{1}{t}, t = 1, 2, 3, \cdots$, with some floor $\beta_{min} \in (0, 1)$ as shown in Fig. 4(b). Such a decay regime is beneficial for faster learning of the higher probabilities (*e.g.* $p^* \gtrsim 0.1$), as it is equivalent to simple counting and averaging to compute proportions (connecting the rate-based EMA to the counting-based methods of the next section).[11] This does not negatively impact convergence or the error-rate on the lower probabilities (such as $p^* \lesssim 0.05$). In particular,

---

[11]This manner of reducing the rate is equivalent to plain averaging: one can expand and note the telescoping product, or use induction: The estimate after the update at $t, t \geq 2$ is $\hat{p}^{(t)} = \beta^{(t)} o^{(t)} + (1 - \beta^{(t)}) \hat{p}^{(t-1)}$, where $\beta^{(1)} = 1, \cdots, \beta^{(t)} = \frac{1}{t}$, or $1 - \beta^{(t)} = \frac{t-1}{t}$, and we have $\hat{p}^{(t-1)} = \frac{\sum_{i=1}^{t-1} o^{(i)}}{t-1}$ (induction hypothesis), and combining, we obtain the simple average:

with harmonic-decay, one requires $O(\frac{1}{p^*})$ time points instead of $\Omega(\frac{1}{\beta})$ for convergence to within a positive (constant) multiplicative deviation $d$. Note that the changing rate $\beta$ also indicates the predictor's confidence in its current estimate, assuming the target probability does not change: the lower the $\beta$ compared to an estimate $\hat{p}$, the less likely $\hat{p}$ will change substantially in subsequent updates.

Plain harmonic-decay does not completely address the challenges of open-ended non-stationarity however, as the rate is not raised once lowered: in general one needs to raise the rate to learn fast when there is (substantial) change, *e.g.* whenever new salient items are observed along the input stream. On the other hand, ideally, the learning of one item should not interfere with the probability already learned for another unrelated stable item (*e.g.* see the Stable-Changing scenario in Sect. 2.1.2). Therefore, we propose per-edge (per-predictand) learning rates along with (implicit) change detection to address these challenges next.

### 3.2 SMAs Based on Counting

A simple count-based approach to computing the probability of an item is to keep the count of (positive) observations of the item and divide it by the current time (*i.e.* the total count). This proportion estimator is unbiased with minimum variance (MVUE), and is also the maximum likelihood estimator (MLE) (Hogg et al., 2018; Devore, 2016) (see also Appendix B). This simple running-average approach is not sufficiently sensitive to changes in the proportion of a target item (and assumes we know which items to keep track of). We need a way to keep track of only recent history or limiting the window over which we do the averaging. Windowing is a basic tool used for non-stationarity, *e.g.* (Bifet & Gavaldà, 2007; Gama et al., 2014; Mazzetto, 2024). A challenge is that we are interested in a fairly wide range of probabilities, *e.g.* from 1.0 to 0.001, and a fixed history window of size $k$ of all last $k$ observations, a "box" predictor (Sect. B.2), is spatially wasteful and slow to adapt.

#### 3.2.1 The Qs SMA (Count Queues)

The Qs predictor keeps a map, $qMap$, of item to queue, where each queue is a small list of counts (Fig. 7). At each time $t$, after outputting predictions using the existing queues, it updates the queues: First, if the observation $o^{(t)}$ does not have a queue, a queue is allocated for it and inserted in $qMap$. For any queue corresponding to an item $i \neq o^{(t)}$, $o^{(t)}$ is a **negative observation**, and a ***negative update*** is performed, while a ***positive update*** is performed for the observed item $o^{(t)}$ (for a *positive observation*). Every so often, the map is pruned (Sect. 3.2.2). Operations on a single queue are described next (Fig. 7(b)).[12]

A new cell, cell0, at the back of the queue, is allocated each time a positive observation occurs, and its counter is initialized to 1. With every subsequent negative observation, until the next positive observation, cell0 increments its counter, $C_0$. The other queue cells are ***frozen*** (their counts, $C_i$, not changed). Upon a positive observation, before a new back cell (cell0) is allocated, the existing cell0, if any, is now frozen (becomes cell1), and the oldest cell, cellk, is discarded if the queue is at capacity **qcap**. Thus, each cell corresponds to one positive observation and its count is the *time 'gap'* between one positive observation and the next. Queues of different items with different probabilities, correspond to different lengths of history, the lower the probability, the longer the history (but the same maximum number, qcap, of positive observations).

Recall that a value $\hat{p}$ derived from a statistical experiment (a random variable) is called an *unbiased estimate* of $p^*$ if $\mathbb{E}(\hat{p}) = p^*$, and we call it an *upper estimate* if $\mathbb{E}(\hat{p}) > p^*$ (a biased estimator). Lemma 2 summarizes a few properties on a few ways that probabilities could be derived from the queue cells, which informs algorithm design (*e.g.* when to drop items, Sect. 3.2.2). In particular: A single cell's count yields an upper estimate, while we can obtain an unbiased estimate from two or more *frozen* cells.

**Lemma 2.** *Assume a salient item has positive probability $p^* < 1$ in a time span $T$, and its queue cells were allocated and updated in the span $T$, cell $i$ having count $C_i$.*

- *For any queue cell $i$, $\frac{1}{C_i}$ is an upper estimate of $p^*$.*

---

$\hat{p}^{(t)} = (\sum_{i=1}^{t} o^{(i)})/t$. Note that the naming 'harmonic EMA' is somewhat inaccurate: this is no longer exponentially weighting of the history (only when the floor $\beta_{min} > 0$ is reached).

[12]ADWIN (Bifet & Gavaldà, 2007), also based on queuing, could be used here instead of our simpler counts queue. ADWIN is designed for computing mean and variances on more general numeric (not Boolean) sequences. We compare with ADWIN on Boolean sequences in Appendix B.3.

**PredictUpdateViaQs**($[o]$)
   // Input is sequence $[o] = [o^{(1)}, o^{(2)}, \cdots]$.
   $qMap \leftarrow \{\}$ // An empty map, item→queue.
   $t \leftarrow 0$ // Discrete time.
   Repeat // Increment time, predict, observe.
      $t \leftarrow t + 1$ // Increment time.
      GetPredictions($qMap$) // Get the predictions.
      // Use observation at time $t$, $o^{(t)}$, to
      UpdateQueues($qMap, o^{(t)}$) update $qMap$.
      If t % 1000 == 0: // Periodically prune $qMap$.
         PruneQs($qMap$) // (a heart-beat method).

**GetPredictions**($qMap$) // Emit probabilities.
   // Returns a map: item → probability .
   $\mathcal{W} \leftarrow \{\}$ // allocate an empty map.
   For each item $i$ and its queue $q$ in $qMap$:
      // One could remove 0 probabilities here.
      $\mathcal{W}[i] \leftarrow$ GetProb($q$)
   Return $\mathcal{W}$ // The final predictions.

**UpdateQueues**($qMap, o$) // latest observation $o$.
   If item $o \notin qMap$: // when $o \notin qMap$, insert.
      // Allocate & insert q for $o$.
      $qMap[o] \leftarrow$ Queue()
   For each item $i$ and its queue $q$ in $qMap$:
      If $i \neq o$:
         // All but one will be negative updates.
         NegativeUpdate(q) // Increments a count.
      Else: // Exactly one positive update.
         // Add a new cell, with count 1.
         PositiveUpdate(q)

**Queue**($cap = 3$) // Allocates a queue object,
   //   with various fields (capacity, cells, etc.)
   $q.qcap \leftarrow cap$ // max size $cap, cap > 1$.
   // Array (or linked list) of counts.
   $q.cells \leftarrow [0, \cdots, 0]$
   // Current size or number of cells ($\leq cap$).
   $q.nc \leftarrow 0$
   Return $q$

**GetProb**($q$) // Extract a probability from the
   // queue $q$.
   If $q.nc \leq 1$: // Too few cells (grace period).
      Return 0
   Return $\frac{q.nc-1}{GetCount(q)-1}$

**GetCount**($q$) // Get total count of all cells in $q$.
   Return $\sum\limits_{0 \leq j < q.nc} q.cells[j]$ // sum all cells.

**NegativeUpdate**($q$) // Increments $cell0$ count.
   // The back (latest) cell of $q$ is incremented.
   If $q.nc \geq 1$: // Do nothing, if no cells.
      q.cell[0] $\leftarrow$ q.cells[0] + 1

**PositiveUpdate**($q$)  // Adds a new (back) cell.
   // Existing cells shift one position. Oldest cell is
   // discarded, in effect, when $q$ is at capacity.
   If $q.nc < q.qcap$:
      $q.nc \leftarrow q.nc + 1$ // Grow the queue $q$.
   For $i$ in $[1, \cdots, q.nc - 1]$: // Inclusive.
      $q.cells[i] \leftarrow q.cells[i-1]$ // shift (counts).
   $q.cells[0] \leftarrow 1$ // set newest, $cell0$, to count 1.

Figure 7: The main functions of the Qs predictor (left), and individual queue operations (right). The Qs SMA keeps a one-to-one map of items to queues, $qMap$: item → queue, where each queue is a small list of counts. The counts of a queue are combined to yield a probability for the corresponding item.

- *For a set $S$ of $k \geq 2$ cells, let $Ratio(S) := \frac{k-1}{-1+\sum_{i \in S} C_i}$. If $cell0 \in S$, then $Ratio(S)$ is an upper estimate of $p$, and if $cell0 \notin S$ ($S$ contains frozen cells only), then $Ratio(S)$ is unbiased, in particular the minimum-variance unbiased estimator of $p^*$.*

The lemma is established by connecting the cell counts to geometric and negative binomial distributions (Appendix B, and Madani (2024)). To output a probability, for each item that has a queue, we currently use the upper estimate $Ratio(S)$, where $S$ is *all* the cells of the queue (including cell0). With our GetProb() function, the Qs technique needs to see two observations of an item, in sufficiently close time proximity, to start emiting (positive) probabilities for the item (for an unbiased estimate we would need 3). Our experiments did not show a significant difference between including *vs.* excluding cell0 (possibly lower variance when including cell0). The probabilities over all the items in $qMap$ may sum to more than 1, violating the SD property. For evaluations, we apply the FC() function in Fig. 2(a). Appendix B presents a few variants of queuing.

### 3.2.2 Map Pruning in Qs, and Computational Complexity

The least frequent items are removed from the map every so often, for instance via a method that works like a "heart-beat" pattern: map expansion continues until the size reaches or exceeds a maximum allowed capacity $2s_1$, *e.g.* $s_1 := \frac{2}{p_{min}}$ (some multiple of $\frac{1}{p_{min}}$). We can then remove the items in order of least frequency, *i.e.* rank by descending count $C_0$ in cell0 of each queue, and remove until the size is shrunk back to $s_1$. Lemma 3 is useful in justifying how we prune:

**Lemma 3.** *At any time $t$, across all items with a queue, there is exactly one item, with cell0 count $C_0 = 1$. For any integer $c > 1$, there is at most one item with $C_0 = c$.*

**UpdateDyal**($o$) // latest observation $o$.
  // Data structures: $qMap$, $EmaMap$, $rateMap$.
  $qPR, qcount \leftarrow$ GetQinfo($qMap, o$)
  UpdateQueues($qMap, o$)
  WeakenEdges($o$) // Weaken, except for $o$.
  If $qPR == 0$: // item is currently noise?
    Return
  $emaPR \leftarrow EmaMap.get(o, 0.0)$ // 0 if $o \notin$ Map.
  // listen to queue?
  If Q_High($emaPR, qPR, qcount$):
    // set initial rate and probability using queue for $o$.
    $rateMap[o] \leftarrow qcount^{-1}$
    $\delta \leftarrow qPR - emaPR$
  Else: // EMA update with harmonic decay.
    $\beta \leftarrow rateMap[o]$
    $\delta \leftarrow (1 - emaPR) * \beta$
    $rateMap[o] \leftarrow DecayRate(\beta)$ // Sect. 3.1.2
  $EmaMap[o] \leftarrow \delta + emaPR$

**Q_High**($emaPR, qPR, qcount$) // Uses Binomial Tail
  // Parameter: $sig\_thrsh$ (significance threshold).
  If $emaPR == 0$: // when 0, listen to Qs .
    Return True
  If $qPR \leq emaPR$:
    Return False
  // Use (approximate) binomial tail (Sect. 3.3.1).
  Return $qcount * KL(qPR, emaPR) \geq sig\_thrsh$

**WeakenEdges**($o$)
  // Weaken weights except for item $o$.
  // Data structures: $qMap$, $EmaMap$,
  // and $rateMap$. Parameter: $p_{min}$.
  For each item $i$ and rate $\beta$ in $rateMap$:
    If $i == o$: // Don't weaken $o$
      Continue
    // Possibly listen (switch) to the queue.
    $qPR, qcount \leftarrow$ GetQinfo($qMap, i$)
    If Q_Low($EmaMap[i], qPR, qcount$):
      $EmaMap[i] \leftarrow qPR$ // Set to q info.
      $rateMap[i] \leftarrow qcount^{-1}$
    Else: // weaken as in EMA.
      $EmaMap[i] \leftarrow (1 - \beta) * EmaMap[i]$
      $rateMap[i] \leftarrow DecayRate(\beta)$

**GetQInfo**($qMap, o$)
  If $o \notin qMap$: // no queue for $o$?
    Return 0, 0
  $q \leftarrow qMap.get(o)$
  Return GetProb($q$), GetCount($q$)

**Q_Low**($emaPR, qPR, qcount$) // Uses Binomial Tail
  // Parameter: $sig\_thrsh$ (significance threshold).
  If $emaPR \leq qPR$:
    Return False
  Return $qcount * KL(qPR, emaPR) \geq sig\_thrsh$

Figure 8: The DYAL SMA where the weaken-and-boost update logic is similar to sparse EMA. Here each prediction edge has a small queue and its own learning rate, in addition to the probability estimate (a weight) (see Fig. 9), and during weakening and boosting, the queue estimates could be used to reset both the weight and the learning-rate of the edge.

The proof follows from how the SMA operates: exactly one item is observed at any time (one, new, cell0 with value 1) all existing cell0 counts are incremented. Some count values may be 'missed' (*i.e.* cell0 values may not form a consecutive increasing integer sequence) because the same item can be observed at different times. The complete proof and further properties on the spread of the counts, across all the cells in all the queues (not just cell0), appear in Madani (2024). Therefore, when pruning, an item in position $k$ in the sorted list has $c_0 \geq k$. One can use the binomial tail to see that the probability that a salient item with true probability $p$ ($p \geq p_{min}$) is kept and tracked, rapidly goes up the farther $p$ is from $p_{min}$. For instance, for an item with $p \geq 0.01$ the probability that the item in not observed in $\frac{10}{p}$ consecutive time points is below 0.01 (sorting and keeping a multiple of $\frac{1}{p_{min}}$ is unlikely to prune salient items). See Sect. 3.3.1 for further on the binomial tail. Note that an item that is dropped could be added later. Temporary but possibly long streaks of certain events (*e.g.* due to non-IID causes) can lead to poor estimates or incorrect pruning, and to address such possibilities techniques such as multilevel updating may be useful (*e.g.* see Madani et al. (2026)).

Count values in queue cells can also be periodically reset to prevent numeric overflow. Predicting and updating take the same $O(qcap|qMap|) = O(\frac{qcap}{p_{min}})$ (or just $O(\frac{1}{p_{min}})$), where we assume $O(1)$ time for summing numbers and taking ratios (qcap is small constant, such as 3 or 5).

## 3.3 DYAL: Combining Sparse EMA and Qs

The Qs technique can give us a good roughly unbiased initial estimate of the probability of an item, but it has a high variance (*e.g.* see Fig. 10(c)) unless we use considerable queue space (many queue cells, and trade off plasticity for stability). With sparse EMA (Sect. 3.1), and with our assumption that there will be a period of stability when a salient item is observed (stability for that item), one could begin with the Qs estimate and fine tune it to achieve lower variance in the estimation. We next show that these complementary benefits of each approach can indeed be put together, and we call the combination **DYAL**, for *dynamic adjustment of the learning* (rate), *i.e. dialing* up and down the learning (Fig. 8).

An update in the DYAL SMA has the simple logical structure of the plain (sparse) EMA, *i.e.* weaken-and-boost (Fig. 5): weaken all existing edges, then boost (strengthen) the edge to the observed item (Sect. 3.1). However each edge in DYAL, in addition to its weight (a scalar probability), also has other parameters, in particular its own learning rate, which is used for the weakening and boosting of that edge. Fig. 9 shows the parameters kept with each edge, for the three main SMAs of this paper. Associated with each DYAL edge, there is also a queue, and during weakening or boosting an edge, it is possible that the queue estimates are used: a probability as well as a learning rate are derived from the queue. We refer to this possibility as *listening* to the queue (Sect. 3.3.1). Each learning rate is decayed, via harmonic decay (Sect. 3.1.2), after an edge update. For a visualization of the change patterns in learning rates, see Fig. 10(e) and (f) and Fig. 15. The queues are also used for deciding when an item should be discarded. Thus the queues are used as "gates", the interface to the external observed stream, determining what is kept track of and what is discarded, and providing rough initial estimates, of the probability and $\beta$, whenever the probability of an item needs to substantially change. Note that in the pseudo code of Fig. 8 we use three maps ($EmaMap, rateMap, qMap$), while in the Fig. 9(c) picture they are combined into one logical graph.

Prediction in DYAL is as in plain EMA, and the $EmaMap$ is output. Like Qs, DYAL does not ensure that the probabilities in $EmaMap$ form an SD, and we apply the normalizing FC() function for evaluations.[13] When updating, all the three maps are updated in general, as given in the UpdateDyal() function. Upon each observation $o$, first the queues information on $o$ is obtained. This is the current queue probability $qPR$ (possibly 0) on $o$, and the $qcount$ (via GetCount()). Then $qMap$ is updated (a Qs-type update). Finally, all the edges (entries) in the $EmaMap$, except for $o$, are weakened. If there was no queue for $o$, *i.e.* when $qPR$ is 0, $o$ must be new, or noise in general, and nothing more is done, *i.e.* no (EMA) edge strengthening is done (note that a queue is allocated for $o$ upon the $qMap$ update). Weakening, for each edge, is either a plain EMA weakening, or the queue estimates are used, covered next in Sect. 3.3.1. Similarly, for strengthening, the condition for listening to the queue is examined, as described next, and the appropriate strengthening action is taken, *i.e.* plain EMA strengthening or listening to the queue.

We can verify from the logic of DYAL that in the stationary setting the probabilities in $EmaMap$ should converge to the true probabilities, except there is some low probability that once in a while resetting to the queue may occur (Sect. 3.3.1) (the queues contain higher variance estimates). We expect that applying the normalizing FC() does not affect the stable salient or higher probabilities substantially (whenever every salient item is estimated well), but specially at times of change and resets to a queue estimate, where high variance (inaccurate) estimates for one or a few items are used, the effect on stable can be higher. We leave a careful analysis of interactions amongst the probability estimates to future work.

### 3.3.1   When and How to Listen (Reset) to the Queue Estimates

The queue for each item provides the number of positives observed recently and the total count across cells *i.e.* the number of trials (roughly, the GetCount() function). A probability estimate $qPR$ is derived from these numbers (GetProb()). We also have an estimate of probability, emaPR, from the EMA weights $EmaMap$, which we expect to be generally more accurate with lower variance than $qPR$, but from time to time change occurs and this estimate becomes out-dated and should be discarded (reinitialized). Based on the counts and the queue estimate $qPR$, we can perform a binomial-tail test that asks whether, when assuming $emaPR$ is the true probability, one can observe the alternative probability $qPR$ in $qcount$ trials, with some reasonable likelihood. The tail can be approximated efficiently in $O(1)$ time when one has the number of trials and the observed probability $qPR$ (Arratia & Gordon, 1989; Ash, 1990), yielding a basic tool with diverse applications such as in text analysis (Madani, 2021). The tail tells us how likely it is that a binary event with assumed probability $emaPR$ could lead to the observed qPR proportion. As seen in Fig. 8, the approximation is based on the KL divergence. When this event is sufficiently unlikely, or, equivalently, when the binomial-tail score surpasses a threshold (default 5) DYAL resets to the queue estimate and sets a new rate $\beta$ accordingly too (Sect. 3.1.2) Thresholds of 3, 5 (default), and 7 correspond to roughly 90%, 99%, and 99.9% confidences respectively (99% means 0.01 chance of making a false-alarm error, *i.e.* resetting

---

[13]In an earlier version of DYAL it was ensured that DYAL's output would be an SD by transferring probabilities among the predictands more carefully, requiring extra logic (Madani, 2024). However leaving this operation to FC() leads to a significantly simpler technique, and our experiments do not show a large uniform difference between the two versions.

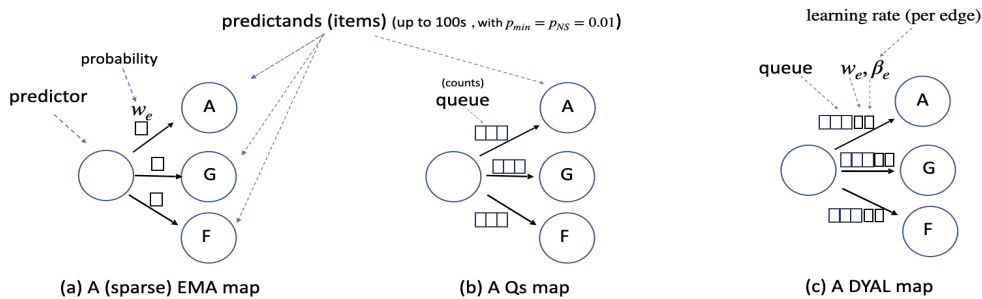

Figure 9: Comparing the parameters of each prediction edge by the main SMAs of this paper (3 predictands shown in each map: items $A$, $G$, and $F$). (a) Sparse EMA maintains a single weight, *i.e.* probability $w_e$, with each edge, and uses a single learning rate independent of the edges. (b) The Qs SMA (center), keeps and updates a small queue of counts for each edge (integers, up to 3 queue cells in this picture), and has no (explicit) learning-rate. (c) DYAL, a combination of EMA and Qs, keeps a probability $w_e$, a learning-rate $\beta_e$, and a small queue for each edge. The rate $\beta_e$ normally decays with each update, but at times $\beta_e$ along with $w_e$ are reset to the estimates derived by the edge's queue (Sect. 3.3).

while the EMA estimate should have been kept).[14] See Sect. 5.1.2. Lowering the threshold moves DYAL closer to Qs (higher variance, but faster reaction to change), and increasing the threshold takes it closer to the static sparse EMA (lowers false positives but increases the miss possibility, or the run length), a tradeoff common in change detection tasks (Aminikhanghahi & Cook, 2016; van den Burg & Williams, 2022).

### 3.3.2 Queue Size and the Granularity of Probabilities

A single queue cell can only provide (over-estimate) probabilities in $\{1, 1/2, 1/3, ...\}$ (many fractions such as 0.75 are not not representable). Using multiple queue cells generates more variety (Madani, 2024). The EMA estimate can also better estimate finer details, as long as the learning rate is appropriately small. However, if the true probability of an item goes from say 0.5 to 1.0, the EMA part of DYAL may not adapt sufficiently fast if it has a low learning rate at time of change. The binomial tail test with a conservative threshold (such as 5) and a small queue may not detect the change either. This motivates additional queue cells (*e.g.* 5 queue cells can be better than 3, *e.g.* see Fig. 19(b)) as well as keeping a positive floor on the learning rates (the harmonic decay should not go to zero when change is possible).[15]

### 3.3.3 Computational Complexity of DYAL

The pruning logic of $qMap$ in DYAL is identical to the Qs (Sect. 3.2.2), except that whenever an entry is deleted from the queue, its corresponding entries (key-value pairs) in $rateMap$ and $EmaMap$ are also deleted, thus $|qMap| \geq |rateMap| = |EmaMap|$. Consequently asymptotic space and update times are the same as for Qs and sparse EMA (a fixed overhead).

## 4 Experimental Set Up

Filtering and normalizing (*i.e.* the FC() function of Fig. 2, default $p_{min} = 0.01$) is applied to the output of all predictors, for evaluations at every time point. The default parameters for DYAL are qcap of 3 and binomial-tail threshold of 5, and we report the performance of DYAL for different $\beta_{min}$, often set at 0.01 or 0.001. We are interested in $\beta_{min} \leq 0.01$ because our target range is learning probabilities in roughly $[0.01, 1.0]$ well. For the static (sparse) EMA, we report and change the (fixed) $\beta$ used, for harmonic EMA

---

[14]As in Sect. 3.1.2, with each edge having its own rate, the rate $rateMap[i]$ can be used as a measure of the predictor's uncertainty around the probability estimate $EmaMap[i]$. Initially, when set to the queue estimate, the rate can be relatively high, and is lowered over time (increased confidence in the estimated probability) with rate decay.

[15]The floor $\beta_{min}$ on the learning rate could be a function of the current probability estimate $\hat{p}$ (*e.g.* $\frac{\hat{p}}{10}$), rather than a constant (which is the case in this paper). This provides more plasticity (faster convergence), but at the cost of higher variance.

| deviation threshold→ | 1.5 | 2 | 1.5 | 2 |
|---|---|---|---|---|
| change type and $O_{min}$ ↓ | Qs, 5 | | Qs, 10 | |
| $0.025 \leftrightarrows 0.25$, 10 | $0.423 \pm 0.029$ | $0.189 \pm 0.023$ | $0.395 \pm 0.021$ | $0.234 \pm 0.014$ |
| $0.025 \leftrightarrows 0.25$, 50 | $0.382 \pm 0.038$ | $0.131 \pm 0.028$ | $0.222 \pm 0.039$ | $0.062 \pm 0.017$ |
| $\mathcal{U}(0.01, 1.0)$, 10 | $0.443 \pm 0.030$ | $0.234 \pm 0.042$ | $0.468 \pm 0.028$ | $0.284 \pm 0.035$ |
| $\mathcal{U}(0.01, 1.0)$, 50 | $0.380 \pm 0.050$ | $0.173 \pm 0.080$ | $0.247 \pm 0.060$ | $0.102 \pm 0.076$ |
| | static EMA, 0.01 | | static EMA, 0.001 | |
| $0.025 \leftrightarrows 0.25$, 10 | $0.510 \pm 0.024$ | $0.357 \pm 0.020$ | $0.996 \pm 0.006$ | $0.760 \pm 0.043$ |
| $0.025 \leftrightarrows 0.25$, 50 | $0.255 \pm 0.045$ | $0.129 \pm 0.028$ | $0.705 \pm 0.032$ | $0.560 \pm 0.023$ |
| $\mathcal{U}(0.01, 1.0)$, 10 | $0.686 \pm 0.030$ | $0.477 \pm 0.035$ | $0.818 \pm 0.038$ | $0.693 \pm 0.054$ |
| $\mathcal{U}(0.01, 1.0)$, 50 | $0.397 \pm 0.056$ | $0.209 \pm 0.045$ | $0.775 \pm 0.085$ | $0.602 \pm 0.099$ |
| | harmonic EMA, 0.01 | | harmonic EMA, 0.001 | |
| $0.025 \leftrightarrows 0.25$, 10 | $0.502 \pm 0.025$ | $0.351 \pm 0.021$ | $0.957 \pm 0.021$ | $0.668 \pm 0.053$ |
| $0.025 \leftrightarrows 0.25$, 50 | $0.247 \pm 0.046$ | $0.123 \pm 0.028$ | $0.606 \pm 0.031$ | $0.494 \pm 0.018$ |
| $\mathcal{U}(0.01, 1.0)$, 10 | $0.684 \pm 0.033$ | $0.476 \pm 0.034$ | $0.813 \pm 0.036$ | $0.683 \pm 0.050$ |
| $\mathcal{U}(0.01, 1.0)$, 50 | $0.395 \pm 0.056$ | $0.204 \pm 0.043$ | $0.761 \pm 0.079$ | $0.592 \pm 0.097$ |
| | DYAL, 0.01 | | DYAL, 0.001 | |
| $0.025 \leftrightarrows 0.25$, 10 | $0.480 \pm 0.029$ | $0.332 \pm 0.025$ | $0.586 \pm 0.066$ | $0.408 \pm 0.061$ |
| $0.025 \leftrightarrows 0.25$, 50 | $0.251 \pm 0.048$ | $0.150 \pm 0.031$ | $0.099 \pm 0.066$ | $0.053 \pm 0.037$ |
| $\mathcal{U}(0.01, 1.0)$, 10 | $0.560 \pm 0.035$ | $0.351 \pm 0.054$ | $0.529 \pm 0.052$ | $0.313 \pm 0.045$ |
| $\mathcal{U}(0.01, 1.0)$, 50 | $0.320 \pm 0.11$ | $0.189 \pm 0.128$ | $0.246 \pm 0.081$ | $0.128 \pm 0.083$ |

Table 2: Synthetic single-item non-stationary: Deviation-rates (lower is better) where $p^*$ either oscillates back and forth between 0.025 and 0.25 ($0.025 \leftrightarrows 0.25$), or otherwise is drawn uniformly at random from $[0.01, 1.0]$ ($p^* \sim \mathcal{U}(0.01, 1.0)$). Each sequence is 10k long, and the deviation-rate is averaged over 500 such sequences. For $0.025 \leftrightarrows 0.25$, each subsequence (wherein $p^*$ is constant) is roughly same length: 400 long for $O_{min} = 10$, and 2k for $O_{min} = 50$. For $\mathcal{U}(0.01, 1.0)$, each subsequence only has to meet the $O_{min}$ constraint (see Sect. 5.1). Qs SMAs do best here, and DYAL variants do equally well or come close (in terms of both their best- and worst-case performance).

and DYAL, we change the $\beta_{min}$, and for Qs, the qcap. The referee (noise-marker) uses default $c_{NS} = 2$ and log-loss (in tables and figures) refers to AvgLogLossNS (Sect. 2.2.2). The map size is periodically checked and if above 225 entries, is pruned to 150 (a multiple of $p_{min}^{-1}$, Sect. 3.3.3). All code is in Python (Madani, 2024). all paired comparisons, comparing two SMAs on tens of 10k long sequences, take no longer than a minute on an Apple laptop. DYAL is roughly 5x slower than sparse EMA (neither optimized).

## 5   Synthetic Experiments

Here, we generate sequences knowing the true SDs $\mathcal{P}^{(t)}$. Appendix D.1 reports deviation rates when tracking a single item in a stationary setting. In such stationary setting, we find (sparse) EMA variants, including DYAL, do significantly better than Qs, although, for static EMA, the appropriate setting of the fixed rate is an issue (Sect. 3.1.1). We next closely examine the behavior of the SMAs, specially DYAL (Fig. 10), when the probability of a single item repeatedly changes, before looking at performance on a multi-item scenario.

### 5.1   Tracking a Single Item, Non-Stationary

Here, sequences of 10000 items are generated in each trial. We report on two main settings for non-stationarity: In the first setting, $p^*$ oscillates between, 0.25 and 0.025, thus an abrupt 10x change is guaranteed to occur and frequently (the Stable-Changing example scenario of Sect. 2.1.2, focused on one changing salient item). This oscillation is shown as $0.25 \leftrightarrows 0.025$ in Table 2. In the second 'uniform' setting, each time $p^*$ is to change, we draw a new $p^*$ uniformly at random from the interval $[0.01, 1.0]$, shown as $\mathcal{U}(0.01, 1.0)$, and in this setting, some changes are large, others small and could be viewed as "drifts". The stable period,

during which $p^*$ cannot change (to allow time for learning), is set as follows. For both settings, within a stable period, the target item (item 1) has to be observed at least $O_{min}$ times before $p^*$ is eligible to change, where results for $O_{min} \in \{10, 50\}$ are shown in Table 2. Additionally, we impose a general minimum-length constraint (not just on positive observations) for the $0.25 \leftrightarrows 0.025$ setting: each stable period has to be $\frac{O_{min}}{\min(0.025, 0.25)}$, so that the different periods would have similar length (expected 400 time points when $O_{min} = 10$, and 2000 when $O_{min} = 50$). In this way, subsequences corresponding to $p^* = 0.25$ would not be too short (otherwise, deviation-rate performance when $p^* = 0.025$ dominates). Thus, with $0.25 \leftrightarrows 0.025$, we get an expected 25 stable subsequences (or changes in $p^*$) in 10k long sequences with $O_{min} = 10$, and 5 switches in $p^*$ when $O_{min} = 50$. For the uniform setting, we did not impose any extra constraint, and respectively with $O_{min}$ of 10 and 50, we get around 200 and 50 stable subsequences in 10k time points. In this setting, performances in periods when $p^*$ is low do dominate the deviation rates.

Here the simple Qs SMA excels, specially when $O_{min}=10$. With such non-stationarity, qcap $=5$ can even work better than qcap $=10$. In this non-stationary setting, DYAL is the 2nd best, and performs substantially better than the other EMA variants. $O_{min}$ of 10 may still be considered on the low side (*i.e.* relatively high non-stationarity). With $O_{min} = 50$, every predictor's deviation-rate improves, compared to $O_{min}=10$ setting. DYAL, with $\beta_{min}=0.001$, performs best for the $0.25 \leftrightarrow 0.025$ and Qs, qcap $=10$, works best with uniform $p^*$, DYAL remaining second best. When picking from uniform, the change in $p$ may not be high, Qs can pick up such change faster, and more naturally, than DYAL, which performs explicit tests and may or may not raise its learning rate depending on the extent of change. However, Qs with 10, despite its simplicity, does require significant extra memory (and we'll find that in the multiple items settings, no setting for Qs works better than DYAL). Appendix B.3 contains comparisons to ADWIN (Bifet & Gavaldà, 2007) in this setting.

### 5.1.1 Behavior of the Output Estimates of the Various SMAs

Figs. 10 shows plots of the output estimates $\hat{p}^{(t)}$, along one of the oscillating sequences ($0.25 \leftrightarrow 0.025$). As expected, we observe that the Qs technique leads to high variance during stable periods, qcap of 5 substantially higher than qcap of 10 (Fig. 10(d)). Similarity for both (static) EMA and DYAL, the rate of 0.01 exhibits higher variance than 0.001 (Figs. 10(b) and (c)). Static EMA and DYAL with 0.01 are virtually identical in Fig. 10(b), except that DYAL can have discontinuities when $p^*$ changes (when it listen to its queue estimates, as discussed next), and converges from both sides of the target $p^*$, while static EMA converges from one side.

### 5.1.2 Change Causes Spikes in the Learning Rates of DYAL

Fig. 10(e) shows the evolution of the learning rate of DYAL, and in particular when DYAL detects a (substantial or sufficiently significant) change (shown via red dots near the x-axis). At the scale shown, rate increases look like pulses: they go up (whenever DYAL listens to a queue) and then with harmonic decay, they fairly quickly come back down (resembling spikes at the scale of a few hundred time points, for probabilities in $[0.01, 1.0]$). We also note that this rate increase can happen when a high probability (0.25) changes to a low probability (0.025) as well, though the rate increase is not as large, as would be expected. DYAL is not perfect: Not all the changes are caught (via switching to its queues' estimates, Sect. 3.3.1), in particular, there are a few "false negatives" (misses), when a change in $p^*$ occurs in the sequence and DYAL continues to use its existing learning rates, slowly adapting its estimates. This may suffice when the $\beta$ at the time is sufficiently high or when the EMA estimate is sufficiently close. In particular, we can observe in Figs. 10(b) and (c) that a higher rate, $\beta_{min} = 0.01$, leads to fewer switches compared to 0.001, and in Figs. 10(b) we observe that DYAL behaves almost identically to static EMA when both have 0.01 (there appears to be one switch only at $t = 15000$ for DYAL). Fig. 10(f) presents a close up of when a (significant) change is detected and a switch does occur, from a low true probability of $p^* = 0.025$ to $p^* = 0.25$ in the picture. Note that before the switch, the estimates (the blue line) start rising, but this rise may be too slow. Once sufficient evidence is collected, both the estimate and the learning rate are set to the queues' estimate, which is seen as a more abrupt or discontinuous change in the picture. The estimates then more quickly converge to the new true probability of 0.25. While we have not shown the actual negative and positive occurrences in the picture, it is easy to deduce where they are from the behavior of $\hat{p}^{(t)}$ (specially after the switch, where the estimates are high): the positives occur when there is an increase in the estimate $\hat{p}^{(t)}$ (from $t$ to $t+1$).

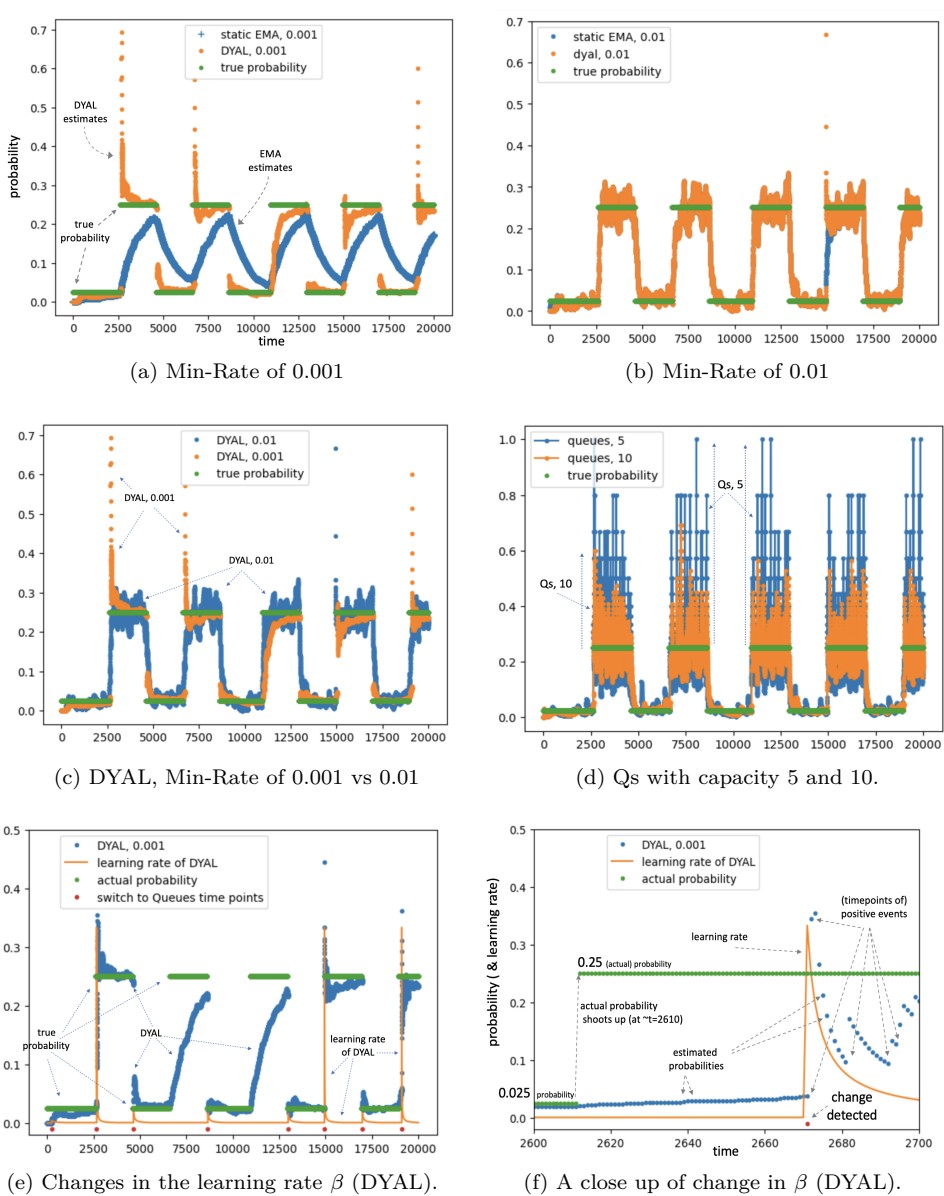

Figure 10: Synthetic single-item oscillation experiments, $0.025 \leftrightarrows 0.25$, on one sequence. Parts (a) to (d) show the probability estimates of EMA, Qs, and DYAL, blue and orange colors, around the true/actual probability $p^*$ (green, up down steps). Note the convergences and variances, *e.g.* Qs with qcap 5 is more variant than 10, in (d) and more variant than EMA or DYAL (note also the y-axes in different figures have different ranges). Parts (e) and (f) show evolution of the learning rate and the rate spikes of DYAL (for a single prediction target).

Fig. 15 and Fig. 21 show additional patterns of change in the learning rate in multi-item real sequences. Appendix D contains a few additional variations.

## 5.2  Synthetic, Non-Stationary, Multi-Item

Here, a non-stationary sequence is a concatenation of "stable" (stationary) subsequences (similar to above), each corresponding to a true underlying SD $\mathcal{P}^{(j)}$ and long enough that all its items are observed sufficiently often before changing. We may have a minimum (overall) length requirement as well. Appendix D.2 describes

the generation process GenSD() which also includes generating noise item. Whenever the underlying SD $\mathcal{P}^{(t)}$ is changed, we can either use new items (**'new items'** setting), or reuse but assign new probabilities (**'recycle items'** in Table 3).

| | 1.5any | 1.5obs | logloss | 1.5any | 1.5obs | logloss | optimal loss |
|---|---|---|---|---|---|---|---|
| new items ↓ | Qs, 5 | | | Qs, 10 | | | |
| $O_{min} = 10$ | 0.94 | 0.24 | 1.17 | 0.88 | 0.17 | 1.19 | 1.040 ±0.11 |
| $O_{min} = 50$ | 0.92 | 0.22 | 1.13 | 0.78 | 0.10 | 1.09 | 1.028 ±0.22 |
| | static EMA, 0.01 | | | static EMA, 0.001 | | | |
| 10 | 0.86 | 0.20 | 1.26 | 1.00 | 0.95 | 2.33 | 1.040 |
| 50 | 0.80 | 0.06 | 1.07 | 0.70 | 0.32 | 1.44 | 1.028 |
| | harmonic EMA, 0.01 | | | harmonic EMA, 0.001 | | | |
| 10 | 0.86 | 0.19 | 1.25 | 0.98 | 0.88 | 2.25 | 1.040 |
| 50 | 0.80 | 0.06 | 1.07 | 0.62 | 0.23 | 1.34 | 1.028 |
| | DYAL, 0.01 | | | DYAL, 0.001 | | | |
| 10 | 0.94 | 0.09 | 1.11 | 0.89 | 0.09 | 1.14 | 1.040 |
| 50 | 0.89 | 0.05 | 1.05 | 0.59 | 0.03 | 1.06 | 1.028 |
| recycle items ↓ | Qs, 5 | | | Qs, 10 | | | |
| $O_{min} = 10$ | 0.91 | 0.23 | 1.16 | 0.83 | 0.13 | 1.14 | 1.040 ±0.11 |
| $O_{min} = 50$ | 0.91 | 0.22 | 1.12 | 0.76 | 0.10 | 1.08 | 1.028 ±0.22 |
| | static EMA, 0.01 | | | static EMA, 0.001 | | | |
| 10 | 0.87 | 0.16 | 1.16 | 1.00 | 0.73 | 1.65 | 1.040 |
| 50 | 0.80 | 0.06 | 1.06 | 0.77 | 0.26 | 1.28 | 1.028 |
| | harmonic EMA, 0.01 | | | harmonic EMA, 0.001 | | | |
| 10 | 0.87 | 0.15 | 1.15 | 0.98 | 0.66 | 1.54 | 1.040 |
| 50 | 0.80 | 0.05 | 1.05 | 0.71 | 0.18 | 1.18 | 1.028 |
| | DYAL, 0.01 | | | DYAL, 0.001 | | | |
| 10 | 0.91 | 0.10 | 1.11 | 0.83 | 0.16 | 1.18 | 1.040 |
| 50 | 0.88 | 0.05 | 1.04 | 0.56 | 0.04 | 1.06 | 1.028 |

Table 3: Synthetic multi-item experiments: performance averages over 50 sequences, ∼10k length each, generated using GenSD: the underlying SD $\mathcal{P}$ is changed whenever *all* salient items in $\mathcal{P}$ are observed $\geq O_{min}$ times ($O_{min} \in \{10, 50\}$). In top half, items are new when underlying SD changes, and in the bottom half, items are 'recycled' (Sect. 5.2).

Table 3 presents deviation rates, generalized to multiple items (described next), as well as log-loss of various SMAs and the optimal (lowest achievable) log-loss: given item $o^{(t)}$ and the underlying SD $\mathcal{P}^{(t)}$, if $o^{(t)} \in \mathcal{P}^{(t)}$ ($o^{(t)}$ is salient), the optimal loss at $t$ is $-\ln(\mathcal{P}^{(t)}(o^{(t)}))$, and if $o^{(t)}$ is noise ($o^{(t)} \notin \mathcal{P}^{(t)}$), then the loss is $-\ln(u(\mathcal{P}^{(t)}))$. The optimal loss is simply the average of this measure over the entire sequence.[16] Note that this loss does not change whether we use new items or recycle items (Table 3), as we evaluate on the same 50 sequences to evaluate the different methods/settings, so the corresponding optimal losses are identical.

The (multi-item) deviation-rate, given the sequence of underlying SDs $[\mathcal{P}]_1^N$, is defined as:

$$dev([\mathcal{W}]_1^N, [o]_1^N, [\mathcal{P}]_1^N, d) = \frac{1}{N} \sum_{t=1}^N multidev(o^{(t)}, \mathcal{W}^{(t)}, \mathcal{P}^{(t)}, d) \;\; \text{(multi-item deviation)},$$

where for the multidev() function, we explored two options: under the more lenient **'obs'** setting, we score based on the observation $o$ at time $t$: $multidev_{obs}(o, \mathcal{W}, \mathcal{P}, d) := deviates(\mathcal{W}(o), \mathcal{P}(o), d)$. Under the more demanding **'any'** setting, we count as deviation if any estimate in $\mathcal{W}$ has high deviation:

---

[16]If a single SD $\mathcal{P}$ generated $[o]_1^N$, what we described is an empirical estimate of the SD entropy, *i.e.* (an estimate of) $u(\mathcal{P}) \ln(u(\mathcal{P})) + \sum_{i \in \mathcal{P}} -\mathcal{P}(i) \ln(\mathcal{P}(i))$. When the underlying SD $\mathcal{P}$ changes from time to time, the computed optimal loss is the weighted average of the entropies, weighted by the length of the subsequence each SD was responsible for.

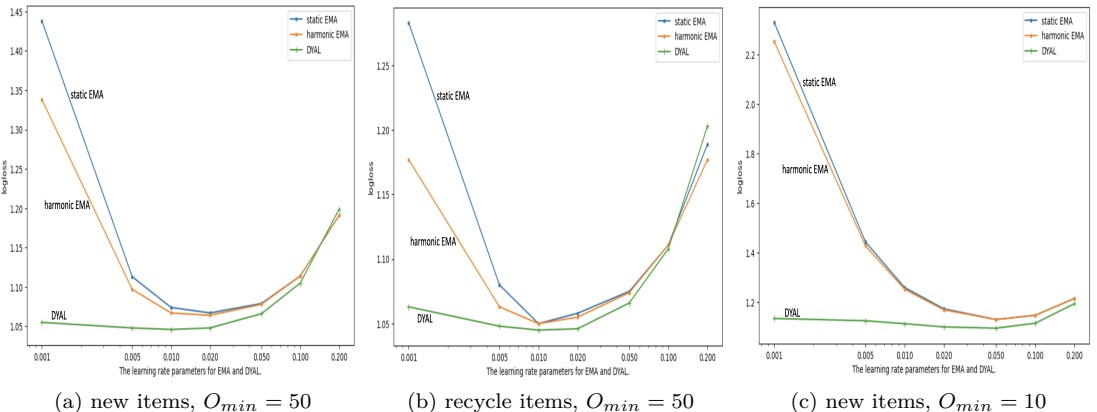

(a) new items, $O_{min} = 50$    (b) recycle items, $O_{min} = 50$    (c) new items, $O_{min} = 10$

Figure 11: log-loss performance, as the learning rate is changed, in synthetic multi-item experiments, $O_{min} = 50$. DYAL is least sensitive to $\beta_{min}$.

$multidev_{any}(o, \mathcal{W}, \mathcal{P}, d) := \max_{i \in \mathcal{P}} deviates(\mathcal{W}(i), \mathcal{P}(i), d)$ ($o$ is not used). $multidev_{obs}()$ is closer to log-loss, as it only considers the observation, and like log-loss, is therefore more sensitive to items with higher probability in the underlying $\mathcal{P}$. $multidev_{any}()$ is a more strict performance measure. Table 3 shows both deviation types with $d = 1.5$.

As expected, when $O_{min}$, *i.e.* the length of the stability period, is increased from 10 to 50, all performances improve, even for Qs with qcap =5.[17] The performance of Qs with qcap =10 is better than qcap =5, even for high non-stationarity $O_{min}$=10, as we change the underlying SD $\mathcal{P}$ only when *all* salient items of SD $\mathcal{P}$ pass the $O_{min}$ threshold. Nevertheless, even with qcap of 10, Qs often trails the best of the sparse EMA variants significantly. Fig. 11 shows that DYAL is not as sensitive to the setting of the (minimum) rate. We also note that while the log-loss values can seem close, the deviation rates can explain or reveal better why DYAL often does perform better overall. We can also pair two methods and count the number of wins (lower log-loss on the same sequence), and DYAL gets substantially more wins in all pairings. Appendix D.3 contains these paired results and additional sensitivity plots.

## 6 Experiments On Real-World Data Sources

A variety of complex phenomena combine in the real world to generate the sequences of observations (not necessarily IID), and even if we assume stable distributions generate the data over some durations, in general we do not have access to the underlying probabilities in order to compare predictors. We use log-loss (AvgLogLossNS of Sect. 2.2.2) and compare different SMAs on the same sequences. We begin with sequences that contain external non-stationarity (Unix commands), and conclude with a source reflecting internal system non-stationarity.

### 6.1 Unix Commands Data

The sequences in our Unix data exhibit significant external non-stationarity (Sect. 6.1): each person's pattern of command usage changes over days and months, as daily patterns of activities change (project changes, vacation, ...). We look at two sequence datasets here: which we refer to as the *52-scientists* data, a subset of data collected by Greenberg (Greenberg, 1988), and the Masquerade sequences (Schonlau et al.,

---

[17]In our setup, imagine $O_{min}$ is lowered to say 1 or 0, an extreme case of non-stationarity, and the item probabilities may lose their meaning. On the other hand, another stationary regime may arise, for instance if $p^*$ is picked uniformly from [0, 1] each time, with a short enough stable period, it is as if $p^* \approx 0.5$ on the whole sequence. A learner making the stationary assumption, with an appropriate learning rate (or otherwise harmonic EMA), may work best (resembling the other extreme of high $O_{min}$ and convergence to full stationarity).

| 52-scientists (52 sequences) | | | | | | Masquerade (50 sequences, 5k each) | | |
|---|---|---|---|---|---|---|---|---|
| sequence length | | | # unique commands per user | | | # unique commands per user | | |
| median | min | max | median | min | max | median | min | max |
| 1.8k | 205 | 12k | 106 | 22 | 359 | 101 | 5 | 138 |

Table 4: Statistics on Unix commands sequences, from the two sources.

2001) (Table 4). Diverse contextual features, derived not just from recently typed commands but also temporal attributes such as time of day and day of week, can serve as predictors of the next command (Davison & Hirsh, 1997; Korvemaker & Greiner, 2000; Madani et al., 2009a). As the sequences would be relatively short if we focused on individual commands as predictors (at most 100s long in almost all cases), we look at the performance of the "always-active" predictor that tries to predict a "moving prior" of which next command is typed.[18]

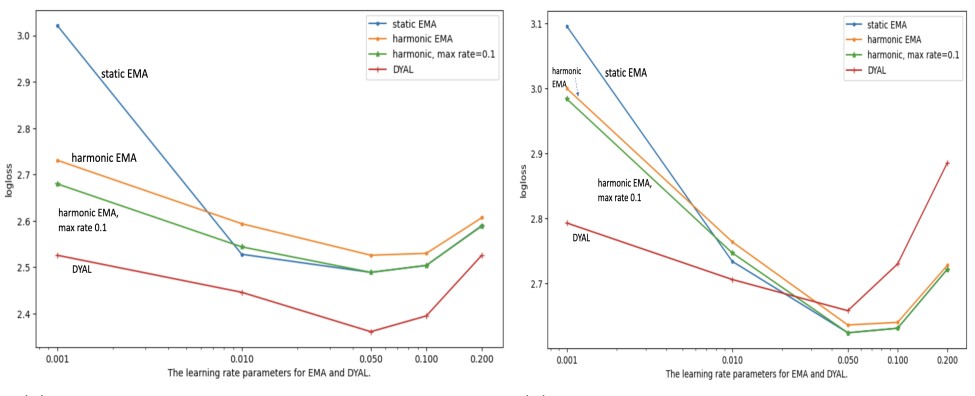

(a) log-loss vs rate, Unix 52-scientists sequences.   (b) log-loss vs rate, on Masquerade sequences.

Figure 12: Losses (AvgLogLossNS()) *vs.* $\beta$ on the Unix sequences.

Fig. 12 shows the performance of the (sparse) EMA variants as a function of the learning rate, and Table 5 reports on Qs performance with a few qcap values, and includes the best of the EMA variants. We observe a similar pattern (compare to Fig. 11): a v-shaped performance, and for DYAL, as before, the plots show less of a dependence on $\beta_{min}$ compared to other EMA variants. However, the degradation in performance as we lower $\beta_{min}$ is more noticeable here. DYAL performs better on 52-scientists compared to others, but is beaten by static EMA on Masquerade, unless we restrict the history kept by the referee, discussed next.

### 6.1.1  Pairing and Sign-Tests on Unix Sequences

We pair and compare DYAL to static EMA as harmonic and static behave similarly. At $\beta$ of 0.05, best for both DYAL and static EMA, on the 52-scientist sequences, we get 46 wins for DYAL over static (lower log-loss for DYAL), and 6 wins for static. On average, 13% of a sequence is marked noise by the referee. As we increase the referee threshold $c_{NS}$ from 2 to 3 to 4, we get additional wins for DYAL and the log-loss performances improve for all methods, and the fraction of sequence marked noise goes up, reaching 18% at $c_{NS} = 4$. Conversely if we lower the referee threshold to 1, we get fewer 35 wins for DYAL over static EMA (and 11% marked noise). If we impose a window, of say 500, for the referee, the head-to-head wins of DYAL over other SMAs remains the same or improves.

---

[18]We also only predict the command *stubs*, *i.e.* Unix commands such as "ls", "cat", "more", and so on, without their arguments (filenames, options, etc.). Command arguments are not available in the Masquarade data and our experiments with full commands, *vs.* stubs only, on 52-scientists yielded similar findings.

| 52-Scientists | | | | | Masquerade | | | | | | |
|---|---|---|---|---|---|---|---|---|---|---|---|
| Qs | | | | DYAL | Qs | | | | static | static* | DYAL* |
| 2 | 3 | 5 | 10 | 0.05 | 2 | 3 | 5 | 10 | 0.05 | 0.05 | 0.05 |
| 2.581±0.28 | 2.586 | 2.629 | 2.686 | 2.362 | 2.769±0.5 | 2.735 | 2.754 | 2.830 | 2.624 | 2.56 | 2.42 |

Table 5: On Unix sequences, performances of Qs and the best of EMA variants. Lower qcaps work better, suggesting significant (external) non-stationarity. The right-most two columns report log-loss when referee window size is 500 instead of the default unlimited size (Sect. 6.1.1).

On Masquerade's 50 sequences, we get only 15 wins for DYAL *vs.* 35 for static (EMA), again at $\beta_{min} = 0.05$, where both do their best, which is statistically significant. With the default of $c_{NS} = 2$ only 5% of a sequence is marked noise on average. As in the case of 52-scientists, when we increase the referee threshold from 2 to 3 to 4, we get additional wins for DYAL, and at $c_{NS} = 3$, DYAL has 32 wins over 18 wins for static (8% marked noise with $c_{NS} = 4$). Similar to the above 52-scientists case, lowering the threshold to 1 leads to more wins for static. Importantly, if we use a referee that is based on a finite window of the last say 500 (or 200, etc) (rather than our simple unlimited window size) again DYAL becomes significantly superior: 50 wins for DYAL, and 9% marked noise (compared to 5% with unlimited referee window).

Why a fixed and relatively high learning rate of 0.05 does relatively well here compared to the more dynamic DYAL, on Masquerade sequences (under unlimited referee window)? Any assumption behind the design of DYAL, in particular the sufficient stability assumption, may be partially failing here. For many items, their probability, or appearance frequency, may be high once they appear, but their stability period may be too short, and a simple high learning-rate of 0.05 may work just as well, compared to the slow two-tiered approach of first detection and estimation via the queues, and at some point, switching to the queues estimate. As we increase the referee threshold $c_{NS}$, we focus or bias the evaluation further on the more stable items in the sequences, and we get better relative results for DYAL, giving more credence to the stability explanation. Limiting the window size (from unlimited to a few hundreds) also lowers the impact of noise (recently unseen) items.

Compared to the Expedition sequences (next, Sect. 6.2), the Unix sequences exhibit higher non-stationarity, when we consider a few indicators: the best performance occurs when $\beta_{min}$ is relatively high at $\approx 0.05$ (Fig. 12). Table 5 also shows that Qs does best here with smaller qcap values. Appendix E.1 presents additional visualizations (learning-rate spikes) for further evidence of external non-stationarity.

## 6.2   104 Expedition Sequences

Expedition is an exploratory system developed for investigating self-supervised cumulative learning of explicitly represented structures, referred to as *concepts* (n-grams of characters) (Madani, 2023). The sequences extracted from running Expedition exhibit internal non-stationarity as we explain.[19] The Expedition system repeatedly inputs a random line of text (averaging ∼50 characters)[20] and *interprets* it, involving searching, predicting, and matching of its concepts (Fig. 13). The probabilistic predictions of the concepts are used to pick a high scoring interpretation. The winning interpretation in turn is used to update prediction relations of those concepts participating in the interpretation. Over time, the system also composes new n-grams out of existing ones, and uses them, which improves the subsequent predictions, and thereby the interpretations (*e.g.* 'new' is more predictive than 'n' or 'ne' in a corpus of English text). Generating and using n-grams is the source of (internal) non-stationarity: the system begins at the low level of characters, its starter-set of concepts, and gradually grows its concept set (higher-level concepts). In every episode, once an interpretation, *i.e.* a final concept sequence, is selected, each concept in the sequence acts as a predictor and updates its weight for predicting what concept comes immediately after it to the right (in the final selected interpretation). For example, in the interpretation of Fig. 13(b), the predictor (concept) 'r' observes the

---

[19]Expedition sequences are made available with the code, and Unix sequences are publicly available.

[20]A text corpus of NSF abstracts was used in extracting the Expedition sequences (Dua & Graff, 2017). The NSF dataset contains approximately 120k research paper abstracts, yielding 2.5 million English text lines, over 20 million term occurrences, and just under 100 unique characters.

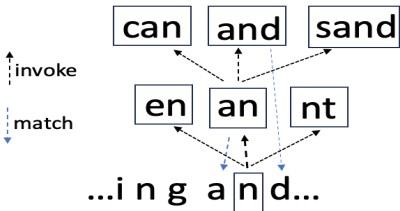
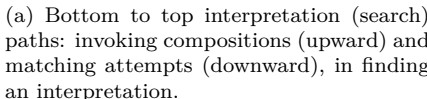

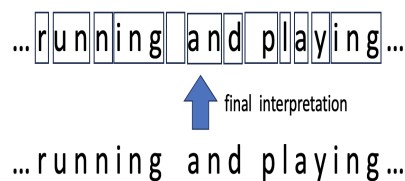

(a) Bottom to top interpretation (search) paths: invoking compositions (upward) and matching attempts (downward), in finding an interpretation.

(b) An example final interpretation, given the input fragment "running and playing".

Figure 13: (a) The interpretation search process consists of invoking (up arrows) and matching attempts (down arrows), until no more compositions that match the input remain. (b) The final selected interpretation is a sequence of highest level available concepts, n-grams, that match the given input line well.

concept 'un' to its right (and not 'u') and updates accordingly, and (the predictor) 'un' observes 'n' to its right and updates accordingly, and so on.

The sequences were obtained from running the Expedition system for four trials, in each trial starting from scratch (character level) and with a different random seed, *i.e.* different sequences of lines are input to the system. Each trial was stopped after 20k to 30k episodes, taking under an hour each. In the course of a trial, thousands of new concepts (bigrams, trigrams, ..) are generated and used. Id generation for concepts is incremental via a simple counter, and as most characters are seen first, before any higher-level n-gram is generated, concepts with lower ids tend to correspond to characters (unigrams) and ids above 95 correspond to bigrams and trigrams. In each trial, for each of a few arbitrary concept (ids) being tracked[21] (a few below id 20, a few around 100 and 500), we collected and created a sequence from what comes after it in the episodes it is active in, *i.e.* it appears in one or more places in the final selected interpretation. For instance, in one trial, the concept (corresponding to) 't' was tracked, and we obtained a sequence of 25k items long for 't', each observation being a concept id.[22] Initially, in the first few 100s of episodes say, 't' will only see single characters (unigrams) immediately next to its right (concept ids below 95). Later on, as bigrams and higher n-grams are generated, a mix of unigrams along with higher level n-grams are observed (after 't'). We tracked a few concepts in each run and collected 104 sequences over the different trials, with median sequence size of 1.2k observations, minimum size of 75, and maximum sequence size of 48k. For the less frequent and newer concepts (as predictors) we get shorter sequences. There are nearly 1000 unique concepts (concept ids) in the longest sequences, and 10s of unique concepts in the shorter ones.

Note that if Expedition did not generate new concepts, *i.e.* both the predictors and predictands remained characters (unigrams), this would be a stationary task, as input lines are randomly sampled from a finite corpus (*e.g.* the edge or conditional probability $P('b'|'a')$ would not change).[23] The non-stationarity here is internal, due to then generation and use of new concepts.

### 6.2.1 Overall Performance on 104 Expedition Sequences

Table 6 shows the losses of our 4 predictors, averaged over the 104 Expedition sequences. As before, all the parameters are at their default when not specified, and log-loss refers to AvgLogLossNS() (Sect. 4). We observe that DYAL does best on average, and as Table 7 shows, pairing and performing sign tests indicates that log-loss of DYAL (with $\beta_{min} = 0.001$) outperforms others (yields the lowest loss) over the great majority of the sequences (high statistical confidence). We also used a referee window size of 500, and obtained similar results (the referee window size is unlimited by default).

---

[21]The same character, and in general concept, gets a different id in different trial runs.

[22]This implies 't' was active in up to 25k episodes (in some episodes, 't' may appear more than once).

[23]We also experimented under this stationary multiclass setting and DYAL had a faster (superior) convergence curve than the other SMAs and Qs (Madani, 2024).

| | Qs | | | | static | | harmonic | DYAL |
|---|---|---|---|---|---|---|---|---|
| | 2 | 3 | 5 | 10 | 0.001 | 0.01 | 0.01 | 0.001 |
| logloss | 2.65 | 2.60 | 2.61 | 2.70 | 2.80 | 2.56 | 2.71 | 2.42 |

Table 6: Loss on 104 Expedition sequences (1.2k median length).

| DYAL, 0.001 vs. $\rightarrow$ | Qs, 3 | static, 0.01 | static, 0.005 | harmonic, 0.01 |
|---|---|---|---|---|
| $c_{NS} = 3$ | 1, 103 | 3, 101 | 24, 80 | 1, 103 |
| $c_{NS} = 2$ (default) | 1, 103 | 3, 101 | 25, 79 | 1, 103 |
| $c_{NS} = 1$ | 22, 82 | 35, 69 | 14, 90 | 17, 87 |
| $c_{NS} = 0$ | 21, 83 | 21, 83 | 5, 99 | 39, 65 |

Table 7: Number of losses and wins of DYAL, with $\beta_{min} = 0.001$, pairing it against a few other techniques, on the 104 Expedition sequences, as we alter the $c_{NS}$ threshold. If observation count $\leq c_{NS}$ then it is marked noise (Sect. 2.2.2). Thus DYAL wins over Qs with (qcap = 3) on 103 of 104 sequences (top left). The number of wins of DYAL is significant at over 99.9% confidence level in all cases. Also, DYAL wins over static with $\beta \in \{0.01, 0.005\}$, on all the 19 longest sequences, at default $c_{NS} = 2$.

### 6.2.2 Sensitivity to Parameters

Table 7 also changes the $c_{NS}$ to assess sensitivity to what is considered noise. Lowering the $c_{NS}$ makes the problem harder, and we have observed here and in other settings, that log-loss goes up. For instance, the log-loss performance of DYAL goes from 2.93 at $c_{NS} = 0$ down to 2.3 with $c_{NS} = 3$. Of course, with $c_{NS} = 1$, we are expecting a technique to provide a good probability estimate even though the item has occurred only twice before![24]

Fig. 14(a) shows the sensitivity to $\beta_{min}$ for DYAL and harmonic EMA, and $\beta$ for static EMA. We observe, as in Sect. 5.2 for the case of multi-item synthetic experiments, that DYAL is less sensitive than both of harmonic and static EMA, while harmonic is less sensitive than static when $\beta$ is set low (and otherwise, similar performance to static). In particular, for longer sequences, lower rates can be better (see Appendix E), but for other EMA variants, low (fixed or minimum) rates remain an issue when faced with non-stationarity (*i.e.* new salient items).

Fig. 14(b) shows the sensitivity of DYAL to the choice of the binomial threshold, which controls when DYAL resets to the queue estimates (Sect. 3.3.1). There is some sensitivity, but we posit that at 3 and 5, DYAL performs relatively well. In particular, we see in Appendix E that when we look at performance on the longer sequences, the performance as a function of binomial threshold becomes more stable, and thresholds 3 and above do best. We also ran DYAL using different queue capacities with $\beta_{min} = 0.001$ (default is $qcap = 3$), and obtained similar log-loss results (*e.g.* 2.38 at $qcap = 2$, and 2.4 for $qcap = 5$).

### 6.2.3 Evolution of the Learning Rates, Degrees, etc.

Fig. 15 shows plots of the evolution of maximum (max-rate) and median of the learning rates in the $rateMap$ of DYAL for two predictors (two sequences), the concept "ten", with just over 200 episodes, and the concept "l" with over 12000 episodes. The number of entries (edges) in the $rateMap$ (and the $EmaMap$), or the out-degree, is also reported. We observe that the maximum over the learning rates contain bursts every so often indicating new concepts need to be learned (predicted), while the median rate converges to the minimum, indicating that most predictands at any given time are in a stable state. On the long 12k sequence, we also see the effect of pruning the map every so often (Sect. 3.2.2 and 3.3.3): the number of map entries remain below 100 as old and low probability items are pruned. Appendix E.1 also includes plots of the max-rate on a few additional sequences (self-concatenations), in exploring evidence for non-stationarity.

---

[24]Without extra information or assumptions, such as making the stationarity assumption and assuming that there are no noise items, or using global statistics on similar situations (*e.g.* past items that were seen once), this appears impossible.

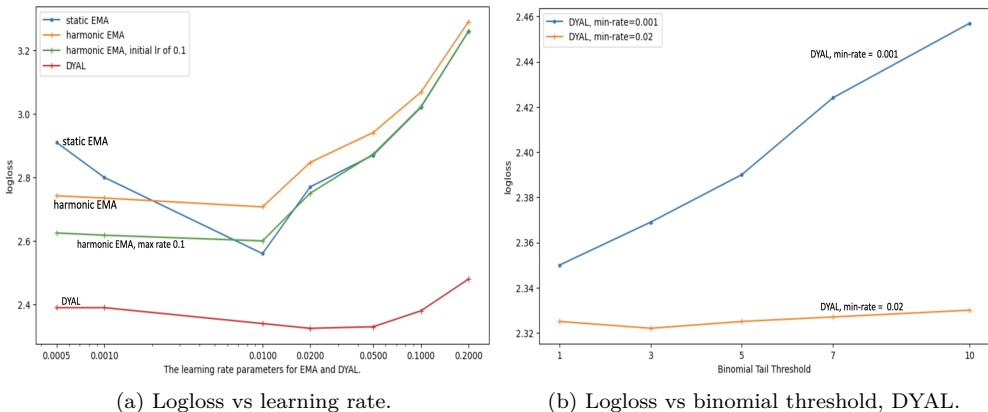

(a) Logloss vs learning rate.

(b) Logloss vs binomial threshold, DYAL.

Figure 14: Expedition sequences: Changing the learning rate (left), and the binomial threshold (right) in DYAL, and plotting the AvgLogLossNS performance.

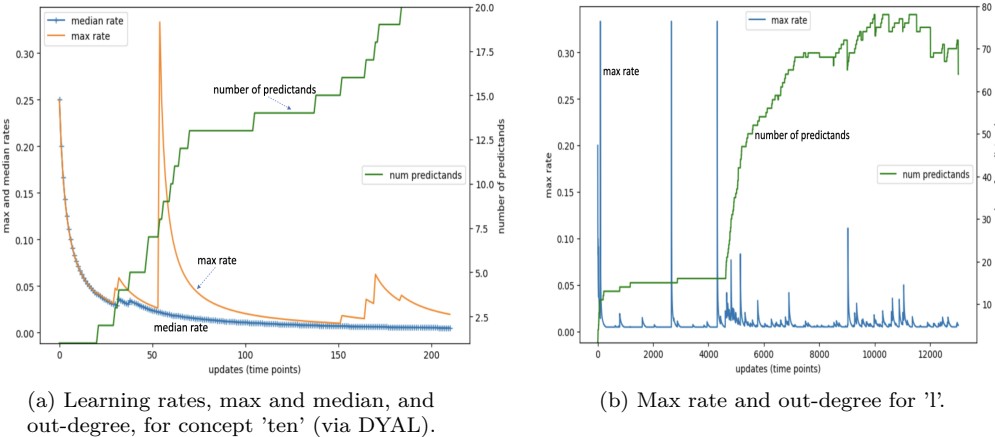

(a) Learning rates, max and median, and out-degree, for concept 'ten' (via DYAL).

(b) Max rate and out-degree for 'l'.

Figure 15: Examples of evolution of the learning rates of DYAL and the number of predictands, or out-degree ($|EmaMap|$), over time. (a) Maximum (max-rate) and median learning rates and out-degree for the concept 'ten'. (b) Maximum learning rates and number of predictands for the 13k-long sequence for the concept 'l'.

## 7   Related Work

Our problem lies in the areas of non-stationarity and learning under concept drift (Ditzler et al., 2015; Gama et al., 2014; Hinder et al., 2023), online multiclass learning (Hoi et al., 2018; Madani et al., 2009b;a; Wester et al., 2011), (non-parametric) density estimation and distribution learning (Davison & Hirsh, 1997; Korvemaker & Greiner, 2000; Mazzetto, 2024; Mazzetto & Upfal, 2023; Cohen et al., 2020), probability forecasting and assessing quality of output probabilities (propriety, calibration) (Filho et al., 2021; Tyralis & Papacharalampous, 2022; Brier, 1950; Good, 1952; Dawid, 1984), change detection (Aminikhanghahi & Cook, 2016; van den Burg & Williams, 2022; Ditzler et al., 2015), streaming data structures and algorithms (Gaber et al., 2005; Gama et al., 2009) and time-series analysis (Weiß, 2018; Maiti & Biswas, 2015). To the best of our knowledge, this combination of efficient open-ended probability prediction under non-stationarity has not been studied before. We provided pointers on relevant work for the tools and techniques we used throughout the paper. We further situate our work within the broader context of similar tasks and problem domains, and provide a short history and discussion here.

Non-parametric density estimation techniques are often based on kernels or keeping track of specific episodes and assume stationarity (*e.g.* Cohen et al. (2020); Devroye & Györfi (1987)). Recent work of Mazzetto (2024), in computational-learning theory, comes closest to ours and extends the density estimation task to an open-ended, in particular with infinite-support, non-stationary setting, but space efficiency and empirical performance of the predictor is not a focus, and the kept history grows logarithmically in stream length. Furthermore, absolute loss (total-variational distance) is used to assess the theoretical quality of algorithms (limiting applicability to estimating only high probabilities well, above say 0.1 (see also Sect. B.2 on the Box SMA).

Our task and the DYAL solution involves a kind of *implicit* change detection (CD) (Aminikhanghahi & Cook, 2016; Gama et al., 2014; Ditzler et al., 2015; Ziffer et al., 2020): Two main categories of adaptation strategy to change are blind (or implicit) and informed (or explicit) (Gama et al., 2014; Ditzler et al., 2015). In all of Qs, sparse EMA (static or harmonic-decay), and Box, adaptation to non-stationarity is done automatically or without explicit detection. DYAL comes closest to explicit detection, and it uses the implicit Qs to do so. In our task, the system need only adjust its output probabilities in a timely manner. In some tasks, such as monitoring for safety and potential attacks, explicitly pinpointing the (approximate) time of change can be important too (Ditzler et al., 2015; Plasse & Adams, 2019). Change detection is a diverse subject studied in several fields such as system monitoring, psychology (*e.g.* within human vision) and image processing and time series analysis (*e.g.* Plasse & Adams (2019); Tartakovsky et al. (2014); van den Burg & Williams (2022); Atto et al. (2021)). Here, we seek a (timely) response and adaptation to a change. Many variants of moving averages are also used in time-series analysis (*e.g.* the ARMA model). There, the observations are ordinal (such as counts) even if discrete (*e.g.* Weiß (2018); Maiti & Biswas (2015)), and stationarity or limited non-stationarity is typically assumed. Similarly, much past work, *e.g.* on variable window sizes, has addressed numeric data, for instance tracking means and variances (Bifet & Gavaldà, 2007; Sebastião et al., 2008). Plasse & Adams (2019) observe that CD is understudied for categorical data, and the authors develop efficient model-based explicit CD techniques for streaming data: the set of items is known and fixed in that work (a multinomial distribution). Their evaluation is based on optimizing change-detection rates (*e.g.* maximize true-positive rates for an acceptable fixed false-positive rate).

The online observe-update cycle, updating a semi-distribution, has a resemblance to online (belief) state estimation, *e.g.* in Kalman filters and partially observed Markov models for control and decision making, in particular with a discrete state space (Dean & Wellman, 1991). Here, the goal is pure prediction (*vs.* action selection or control), though some of the techniques developed here may be useful to that setting, in a changing world. Extensions of the filter, via dynamic window sizes (for numeric sequences), has proved useful for adapting supervised and unsupervised learning techniques, such as Naive Bayes, to non-stationarity (Bifet & Gavaldà, 2007; Ziffer et al., 2021). More broadly, streaming algorithms aim to compute useful summary statistics, such as unique counts and averages, while being space and time efficient, in particular often requiring a single pass over a large data set or sequence, such as the count-min sketch algorithm (Gaber et al., 2005; Gama et al., 2009). Here, we have been interested in computing recent proportions in a non-stationary setting, for continual prediction, carried out by each of *many* (severely) resource-bounded predictors. Learning finite state machines in a streaming manner shares similar philosophy and a similar subproblem of change detection (in particular, Fig. 1, "system for continuous learning", Balle et al. (2014)).

In typical online (supervised) learning, *e.g.* Hoi et al. (2018); Rosenblatt (1958); Littlestone (1988), the focus is on how best to aggregate the predictors (features) for prediction and, for example, on learning a good weighting for a linear model. Prior work of Madani et al. (2009b) on large-scale multiclass learning is also open-ended, including handling when parameters are allocated and deallocated. Subsequent work tackles non-stationarity (Madani et al., 2009a). However, the focus had been on aggregating and ranking classes, while we have focused on learning good *independent probability* predictors, with improved justification for when allocation and deallocation decisions are made (Sect. 3.2.2). The independence is akin to the (multiclass) Naive Bayes model (Lewis, 1998) and the counting techniques for n-gram language models (Rosenfeld, 2000), but in a non-stationary setting. Another avenue of work is calibrating classifiers after training (trained on a ranking or accuracy-related objective) (Filho et al., 2021), and there may be extensions applicable to our non-stationary open-ended online setting. Similarly, investigating online techniques for

learning good mixing weights, but also handling non-stationarities, may prove a fruitful future direction, for instance in the mold of (sleeping) experts algorithms (Freund et al., 1997; Hoi et al., 2018; Dani et al., 2006).

Learning-rate decay has been beneficial, *e.g.* for backpropagation, and there is research work at explaining the reasons (You et al., 2019; Smith, 2018). Here, we motivated decay variants in the context of sparse EMA updates and learning good probabilities fast, and motivated predictand-specific (per connection) rates.

## 8 Summary and Future Directions

We formulated the problem of online open-ended probability prediction, where our underlying assumption is that *for adequately long spans of time, there exist sufficiently many stable (predictable) items* for this task of learning and tracking changing probabilities to be useful. We developed a number of sparse moving average (SMA) techniques for finite-space predictors. We described the challenges of assessing probability outputs under noise and non-stationarity, and developed a method for evaluating the probabilities, based on bounding log-loss. We showed that different predictors work best for different regimes of non-stationarity, but provided evidence that in the regime where the probabilities can change substantially but only after intermittent periods of stability, the DYAL SMA which is a combination of the sparse EMA and the Qs technique, is more flexible and has advantages over either of the simpler methods: the Qs predictor has good sensitivity to abrupt changes (can adapt fast), but also has higher variance, while plain (static) EMA is slower but is more stable. Combining the two, with predictand-specific (per connection) parameters, yielding dynamic learning rates, often leads to faster more robust convergence in the face of non-stationarities.

We touched on a number of open problems in the course of the paper. We hope to further develop and compare various SMAs in different tasks, *e.g.* in the unsupervised concept learning of the Expedition system, and for alternative problem formulations, such as when items have different rewards.

## Acknowledgments

Thanks to Jana Radhkrishinan for granting the freedom conducive to this work while at Cisco, and to Tom Dean's SAIRG reading group, including Brian Burns, Reza Eghbali, Sean Kugele, Georgi Georgiev, Gene Lewis, and Justin Wang, for continued discussions and feedback. Thanks also to James Tee and Aaditya Ramdas for discussions and pointers on online state estimation and change detection. I am grateful to John Bowman for a proof sketch of unbiased estimation based on Rao-Blackwellization, and related pointers. Many thanks to the anonymous reviewers whose many comments and suggestions led to a better paper.

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

## A  Convergence Properties of Sparse EMA

The proof of Lemma 1 follows. Here, we are in the stationary binary setting (Sect. 2.1.1), where the target probability to learn, $\mathcal{P}(1)$, is denoted $p^*$, and the estimates of EMA for item 1, denoted $\hat{p}^{(t)}$, form a random walk ($\hat{p}^{(t)}$ is the estimate immediately after the update at time $t$). Fig. 16 shows the main ideas of the proofs, *e.g.* the expected step size is $\beta^2$ towards $p^*$ when $\hat{p}$ is not too close to $p^*$.

**Lemma.** *EMA's movements,* i.e. *changes in the estimate $\hat{p}^{(t)}$, enjoy the following properties, where $\beta \in [0, 1]$:*

1. *Maximum movement, or step size, no more than $\beta$: $\forall t, |\hat{p}^{(t+1)} - \hat{p}^{(t)}| \leq \beta$.*

2. *Expected movement is toward $p^*$: Let $\Delta^{(t)} := p^* - \hat{p}^{(t)}$. Then, $\mathbb{E}(\Delta^{(t+1)}|\hat{p}^{(t)} = p) = (1 - \beta)(p^* - p) = (1 - \beta)\Delta^{(t)}$.*

3. *Minimum expected progress size: With $\delta^{(t)} := |\Delta^{(t)}| - |\Delta^{(t+1)}|$, $\mathbb{E}(\delta^{(t)}) \geq \beta^2$ whenever $|\Delta^{(t)}| \geq \beta$ (i.e. whenever $\hat{p}$ is sufficiently far from $p^*$).*

*Proof.* (proof of part 1) On a negative update, $\hat{p}^{(t)} - \hat{p}^{(t+1)} = \hat{p}^{(t)} - (1 - \beta)\hat{p}^{(t)} = \beta\hat{p}^{(t)} \leq \beta$, and on a positive update, $\hat{p}^{(t+1)} - \hat{p}^{(t)} = (1 - \beta)\hat{p}^{(t)} + \beta - \hat{p}^{(t)} = \beta - \hat{p}^{(t)}\beta \leq \beta$ (as $\hat{p}^{(t)} \in [0, 1]$).

(part 2) We write the expression for the expectation and simplify: $p^*$ of the time, we have a positive update, *i.e.* both weaken and boost ($(1 - \beta)p + \beta$), and the rest, $1 - p^*$ of the time, we have weaken only ($(1 - \beta)p$). In both cases, the term $(1 - \beta)p$, is common and is factored:

$$
\begin{aligned}
\mathbb{E}(\Delta^{(t+1)}|\hat{p}^{(t)} = p) &= p^*\left(p^* - ((1 - \beta)p + \beta)\right) + (1 - p^*)\left(p^* - (1 - \beta)p\right) \\
&\quad \text{(next, the term, } p^* - (1 - \beta)p \text{, is common and is factored )} \\
&= (p^* + (1 - p^*))(p^* - (1 - \beta)p) - p^*\beta \\
&= p^* - (1 - \beta)p - p^*\beta = p^*(1 - \beta) - (1 - \beta)p \\
&= (1 - \beta)(p^* - p)
\end{aligned}
$$

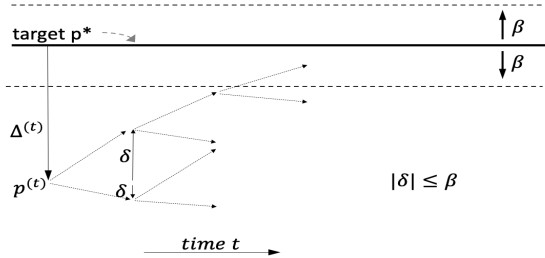
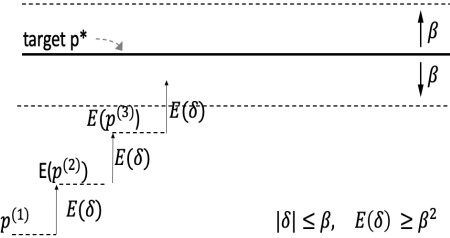

(a) Convergence to within band (here, from below $p^*$)     (b) Expected steps and expected positions

Figure 16: A picture of the properties in Lemma 1, upper bounding the expected number of time steps to enter the band $p^* \pm \beta$ by $\beta^{-2}$, whether from below or above the band (Theorem 1), where $\Delta^{(t)} := p^* - \hat{p}^{(t)}$ and $\delta^{(t)} := |\Delta^{(t)}| - |\Delta^{(t+1)}|$. (a) At any point, there are two possible outcomes after an update (weaken or boost), and the movement, $\delta$, could be toward or away from $p^*$ (e.g. $\delta < 0$), but always $|\delta| \leq \beta$. (b) As long as $\hat{p}$ is not in the band, there is an expected movement, $\mathbb{E}(\delta)$, of at least $\beta^2$, toward $p^*$. $\mathbb{E}(\delta)$ is smallest when $\hat{p}$ is near $p^*$ and largest, up to $\beta$, when $\hat{p}$ is farthest ($\hat{p} = 0$ or $\hat{p} = 1$).

(part 3) Note from our definition of $\delta^{(t)}$, $\delta^{(t)} > 0$ when distance to $p^*$ is reduced (when $|\Delta^{(t+1)}| < |\Delta^{(t)}|$). When $\hat{p}$ is close to $p^*$, e.g. $\hat{p} = p^*$, the expected distance may not shrink, but when outside the band, we can show a minimum positive progress: Assume $\hat{p}^{(t)} \leq p^* - \beta$, then $\delta^{(t)} = p^* - \hat{p}^{(t)} - (p^* - \hat{p}^{(t+1)})$ ($\hat{p}^{(t+1)}$ is also below $p^*$) or $\delta^{(t)} = \hat{p}^{(t+1)} - \hat{p}^{(t)}$, and:

$$\mathbb{E}(\hat{p}^{(t+1)} - \hat{p}^{(t)}) = \mathbb{E}(\delta^{(t)}) = \mathbb{E}(p^* - \hat{p}^{(t)} - (p^* - \hat{p}^{(t+1)})) = \mathbb{E}(p^* - \hat{p}^{(t)}) - \mathbb{E}(p^* - \hat{p}^{(t+1)})$$

$$\text{(the above rewrite used the linearity of expectation)}$$

$$\text{(via part 2)} \quad = \mathbb{E}(p^* - \hat{p}^{(t)}) - (1 - \beta)\mathbb{E}(p^* - \hat{p}^{(t)}) = \beta\mathbb{E}(p^* - \hat{p}^{(t)})$$

$$\geq \beta^2 \quad \text{(via the assumption } |p^* - \hat{p}^{(t)}| \geq \beta)$$

And similarly for when $\hat{p}^{(t)} \geq p^* + \beta$, then $\delta^{(t)} = \hat{p}^{(t)} - \hat{p}^{(t+1)}$. □

Note that from property 2 above, there is always progress towards $p^*$ *in expectation*, meaning that if $\hat{p}^{(t)} < p^*$, then $E(\hat{p}^{(t+1)}) > \hat{p}^{(t)}$, and if $\hat{p}^{(t)} > p^*$, then $E(\hat{p}^{(t+1)}) < \hat{p}^{(t)}$. This is the case even if the probability of moving away is higher than 0.5 ($\hat{p} \leq p^* < 0.5$). We note however, that the result being in expectation, both the actual outcomes for $\hat{p}^{(t+1)}$ can be father from $p^*$ than $\hat{p}^{(t)}$ (consider when $\hat{p}^{(t)} = p^*$). Property 3 puts a floor (a minimum) on amount of the progress towards $p^*$ in expectation, when $\hat{p} \notin [p^* - \beta, p^* + \beta]$, For instance, when $\hat{p}^{(t)} \leq p^* - \beta$, it puts a minimum on the expected positions $\mathbb{E}(\hat{p}^{(t+1)}), \mathbb{E}(\hat{p}^{(t+2)}), \mathbb{E}(\hat{p}^{(t+3)}), \cdots$, until one such point crosses the band, Fig. 16(b), and the Theorem follows.

**Theorem.** *EMA, with a fixed rate of $\beta \in (0, 1]$, has an expected first-visit time bounded by $O(\beta^{-2})$ to within the band $p^* \pm \beta$. The required number of updates, for first-visit time, is lower bounded below by $\Omega(\beta^{-1})$.*

*Proof.* We are interested in the maximum of first-time $k$ when the expected $\mathbb{E}(\hat{p}^{(k)}) \in [p^* - \beta, p^* + \beta]$. Using the maximum movement constraint, as long as $|p^* - \hat{p}^{(t)}| > \beta$, an EMA update does not change the sign of $p^* - \hat{p}^{(t)}$ ($\hat{p}$ does not switch sides wrt $p^*$, e.g. if greater than $p^*$, it remains greater after the update). Thus, before an estimate $\hat{p}^{(t)} < p^*$ changes sides, and exceed $p^*$, it has to be within or come within the band $p^* \pm \beta$. Therefore, start with an arbitrary location $\hat{p}^{(1)}$ outside the band, say $\hat{p}^{(1)} < p^* - \beta$ (similar arguments apply when $\hat{p}^{(1)} > p^* + \beta$), and consider the sequence, $\hat{p}^{(1)}, \hat{p}^{(2)}, \hat{p}^{(3)}, \cdots, \hat{p}^{(k)}$, where $\forall t, 1 \leq t \leq k, \hat{p}^{(t)} < p^* - \beta$. We can now lower bound the expected position of $\hat{p}^{(k)}, k \geq 2$ wrt $\hat{p}^{(1)}$, to be at least $(k-1)\beta^2$ above $\hat{p}^{(1)}$:

$$\mathbb{E}(\hat{p}^{(k)} - \hat{p}^{(1)}|\hat{p}^{(1)} = p) = \mathbb{E}(\hat{p}^{(k)} - \hat{p}^{(k-1)} + \hat{p}^{(k-1)} - \hat{p}^{(1)}|\hat{p}^{(1)} = p)$$

$$\text{(insert all intermediate sequence members)}$$

$$= \mathbb{E}(\sum_{2 \leq t \leq k} \hat{p}^{(t)} - \hat{p}^{(t-1)}|\hat{p}^{(1)} = p)$$

$$= \sum_{2 \leq t \leq k} \mathbb{E}(\hat{p}^{(t)} - \hat{p}^{(t-1)}|\hat{p}^{(1)} = p) \geq (k-1)\beta^2,$$

where we used the linearity of expectation, and the $\beta^2$ lower bound for the last line. With $p^* \leq 1$, an upper bound of $\frac{1}{\beta^2}$ on maximum first-visit time follows: $k$ cannot be larger than $\frac{1}{\beta^2}$ if we want to satisfy $\forall 1 \leq t \leq k, \mathbb{E}(\hat{p}^{(t)}) < p^* - \beta$, or one of $t \in \{1, \cdots, \frac{1}{\beta^2} + 1\}$ has to be within the band $p^* \pm \beta$.

The lower bound $\Omega(\beta^{-1})$ on $k$ follows from the upperbound of $\beta$ on any advancement towards $p^*$. $\qquad \square$

The dynamics of the estimates $\hat{p}^{(t)}$ can be likened, in some respects, to oscillatory physical motions such as the motion of a pendulum and a vertically hung spring: the expected movement is 0 at target $p^*$, corresponding to the resting length of the spring, or its equilibrium length, where spring acceleration is 0, and the expected movement is highest at the extremes (farthest from $p^*$), akin to the acceleration (vector) of the spring being highest when it's most stretched or compressed.

## B  Further Properties of Qs, and Several Qs Variants

We begin this section by reviewing a few properties of Qs, then present two variants, time stamping and Box, and conclude with a comparison to ADWIN (Bifet & Gavaldà, 2007).

### B.1  Statistical Properties of Qs

A single completed (frozen) cell of a queue in the Qs method, in the ideal stationary setting (Sect. 2.1.1), corresponds to the experiment of tossing a biased two-sided coin, with unknown heads probability $p^* > 0$ (the target of estimation) until a heads is observed (shift the cell counts by one). Let $c$ be the number of tosses until and including the first heads outcome. Then $c$ follows the geometric distribution, and $\frac{1}{c}$ is the MLE estimate for $p^*$ (Hogg et al., 2018; Devore, 2016), but it over estimates and the bias gets worse, in the ratio sense, $\frac{1/c}{p^*}$, as $p^* \to 0$. (Madani, 2024). We could repeat the tossing experiment until we get $k > 1$ heads outcomes. This corresponds to multiple completed queue cells. The total number of trials, or the total count over all completed cells, minus the number of heads, follows the more general negative binomial distribution (with parameters $k$ and $p^*$), and Rao-Blackwellization can be used to derive unbiased (and minimum variance) estimates as well as simple upper and lower bounds on $p^*$ (Marengo & Farnsworth, 2021; Rao, 1945; Blackwell, 1947; Lehmann & Scheffé, 1950). Further details and other properties of Qs (such as the sum and the spread of the probabilities derived from the queues) appear in Madani (2024).

### B.2  Variants of the Qs SMA: Time-Stamping and Box

A small variation, which we call the *time-stamp* Qs, allows for a more efficient $O(1)$ updating. Here, the predictor also keeps a single counter, or its own private *clock* reflecting (update) time, and a queue cell, upon allocation, simply records its creation time using the clock. The time gap (difference) between two consecutive queue cells reflects the count (gap), and can be used to derive probability (similar to Sect. 3.2).

The plain Box predictor, also known as a sliding window predictor, keeps a history window of fixed size $K$, of the last $K$ observations for our multiclass setting. The sliding window idea (of fixed or variable size) is a common tool used in various non-stationary problems (Bifet & Gavaldà, 2007; Gama et al., 2014; Ditzler et al., 2015). For our task, the Box SMA can be implemented relatively efficiently via a single queue (plus a count hash map), and thus it has similarities to the Qs technique (one long queue *vs.* several small queues). However, the space consumption is a rigid $\Theta(K)$, and unlike the Qs predictor, $K$ could be relatively large

such as 100 or 1000, depending on how small one wants to go in tracking probabilities. This predictor, similar to the static sparse EMA, has a convergence *vs.* stability problem:[25] depending on $K$ either response time for new items suffers or small probabilities cannot be modeled well (see Sect. 3.1.1 and Table 10). The dynamic Qs is more suited to non-stationarity. Our GitHub code implements these variants.

### B.3   Numeric *vs.* Boolean Streams: A Comparison to ADWIN

The ADWIN (Adaptive Windows) technique of Bifet & Gavaldà (2007) is a streaming method designed for keeping track of a possibly changing mean and variance on numeric sequences, while the input to SMAs are (many-dimensional) sparse Boolean vectors (Sect. 3). ADWIN has similarities to Qs: it uses adjustable window sizes, also making use of queues, of windows, for detecting and responding to change. Instead of the simple count queues (Sect. 3.2.1), for each item, ADWIN could be used (Bifet & Gavaldà, 2007). We compare ADWIN's accuracy in emitting probabilities on binary sequences: the estimated mean $\hat{\mu}$ of a recent subsequence is the probability of observing the item (outcome 1), and we use $1 - \hat{\mu}$ for the probability of 0. The true probability $p^*$ of outcome 1 is picked at random in [0, 1] for periods governed by $O_{min}$, as in Table 2. We looked at oscillation settings ($p_1 \leftrightarrow p_2$, *e.g.* $p_1 = 0.5$ and $p_2 = 0.05$) as well, and obtained similar results. We briefly discuss ADWIN's computational complexity (space/time) below. We used the River online suite of algorithms.[26] We used a maximum loss of 5 for logarithmic loss (but 10 and a few higher values do not change results). Therefore, we did not use a referee (no noise items). We normalized outputs of Qs and DYAL to make sure the two probabilities add to 1 (no higher, no less, as for ADWIN). We also include comparisons based on Brier (quadratic) loss (Sect. C.2).

We experimented with all the parameters of ADWIN (grace period, $M$, clock,...), but the confidence parameter $\delta$ made the most difference (we also needed to use a relatively low grace period and clock such as 5, etc). DYAL ($\beta_{min} = 0.001$) and Qs (qcap of 5) have default parameters below (wins are head-to-head *vs.* ADWIN, eg DYAL *vs.* ADWIN).

|  | $\delta = 0.01$ | $\delta = 0.1$ | $\delta = 0.5$ | Qs | DYAL | wins *vs.* ADWIN |
|---|---|---|---|---|---|---|
| $O_{min}$=10, log | 0.317 | 0.310 | 0.303 | 0.268 | 0.289 | DYAL & Qs win $\sim$30 |
| $O_{min}$=50, log | 0.368 | 0.360 | 0.354 | 0.340 | 0.345 | DYAL & Qs win $\sim$30 |
| $O_{min}$=500, log | 0.388 | 0.387 | 0.386 | 0.402 | 0.388 | mixed |
| $O_{min}$=5000, log | 0.541 | 0.541 | 0.542 | 0.576 | 0.545 | ADWIN wins $\sim$30 |
| $O_{min}$=10, Brier | 0.200 | 0.194 | 0.189 | 0.166 | 0.183 | DYAL & Qs win $\sim$30 |
| $O_{min}$=50, Brier | 0.233 | 0.226 | 0.220 | 0.213 | 0.217 | DYAL & Qs win $\sim$30 |

Table 8: Comparisons with ADWIN, on 30 binary non-stationary sequences of length 10k each (probability of outcome 1 changes, as in Table 2): 'log' is log-loss capped at 5 and Brier is quadratic loss. With fast ($O_{min}$=10) and moderate change ($O_{min}$=50), DYAL and Qs incur lower losses compared to ADWIN.

Very high $\delta$ (above 0.1, *i.e.* assuming change when confidence on change is low) loses its meaning for ADWIN (or any statistical-significance technique): we need $\delta$ to be below 0.1, *e.g.* 0.01 for reduced false positives (the default in River is 0.002). But with low $\delta$ ADWIN is slower to emit good changed probabilities compared to Qs and DYAL, under fast or moderate non-stationarity. With large, *e.g.* $\delta > 0.1$, ADWIN basically reduces to a regiment akin to Qs (reverting to a small subwindow) and we note Qs does well in these binary experiments. In multi-item synthetic experiments (not 'closed-world') Qs begins to trail DYAL substantially. The main ADWIN window can grow unbounded (see below), which is probably the main reason ADWIN has an advantage as we transition to basically stationary settings (*e.g.* $O_{min}$=5000 above).[27]

---

[25]But EMA can be more space efficient, while Box has a rigid $O(K)$ space requirement. On the other hand, update time of Box is $O(1)$ when implemented efficiently, while EMA's is $|\mathcal{W}|$ (the size of its map).

[26]https://riverml.xyz/latest/api/drift/ADWIN (where the better ADWIN2 from the paper is implemented).

[27]The losses in Table 8 increase as we increase $O_{min}$ in these experiments (even though the problem becomes more stable): when $O_{min}$ is small, subsequences corresponding to lower probabilities for outcome 1 are longer, and their losses (lower due to lower entropy) tend to dominate the overall loss for each sequence (see also Sect. 5.2 on comments on log-loss and entropy, and Sect. 5.1 on varying length subsequences).

### B.3.1 Computational Complexity

The main window in ADWIN, of size $W$, grows if there is no (detected) change in the stream, and the memory requirement is $O(M \ln \frac{W}{M})$, where $M \geq 2$ is the maximum number of buckets (cells) kept with size *at each power of 2* up to $W$ (if $M = 2$ there are up to 2 windows of size $2^i$, for each $i$, until $W$) (note: each (sub)window is compact keeping summary stats). ADWIN needs to be modified (eg. largest main window cut at a max size of a few 1000s if we are interested in say $p_{min} = 0.01$) so that its memory (and process time) doesn't grow unbounded. We expect this could be done (River doesn't allow us to limit the total number of windows). We saw a noticeable slow down when testing on 100k sequences. However, even with such capping of main window to say the last 1000 elements observed, the memory requirement tends to be higher than our techniques (over 20 buckets or cells *vs.* say $\leq 7$ cells for DYAL or Qs), and the test performance (to detect change and to pick a subwindow) are more complex in ADWIN (designed for handling the more general non-binary case).

## C    The Near Propriety of Bounded Log Loss

Here we present our analysis that points to the approximate propriety of LogLossNS(). Sect. C.2 briefly discusses a few alternatives that we considered for evaluating SMAs.

In the IID generation setting based on drawing from a SD $\mathcal{P}$ (defined below), we establish how log-loss is affected under various transformations (*e.g.* , due to applying FC()) via the connection of log-loss with the Kulbeck-Leibler (KL) divergence. Under a few assumptions such as small $p_{min}$ ($p_{min} = p_{min}$ used in FC()), the analysis sheds light on when $\mathcal{P}$ remains optimal (*i.e.* leads to lowest loss) or near optimal. Specifically, we look at how far a minimizer $\mathcal{W}^*$ of the bounded-loss can be from $\mathcal{P}$, *i.e.* the extent of (item) *distortion* (see Fig. 17). There are 4 causes or sources of distortion: filtering, capping, bounding the loss, and use of an imperfect noise-marker. We show that any minimizer $\mathcal{W}^*$ is very constrained: only items in $\sup(\mathcal{P})$ with low probability, below or close to $p_{min}$, may have their probability reduced (wrt to $\mathcal{P}$), as LogLossNS has a ceiling on the penalty: moving their probability to other items results in net lowered loss. In the case of reduction, their probability is zero in $\mathcal{W}^*$, and their probability is *proportionately* spread onto remaining (positive probability) items (due to the properties of KL). Lemma 11 (distortion extent) then follows.

### C.1    Summary of the Properties of Bounded log-loss

The proofs are moved to Sect. C.3 to see the flow of arguments and the results below more easily. We begin with the basic properties surrounding the KL() function, extended to SDs. The correspondence of LogLoss to KL divergence (Kullback & Leibler, 1951; Cover & Thomas, 1991) extends to SDs:

**Definition 1.** *The entropy of a non-empty SD $\mathcal{P}$ is defined as:*

$$H(\mathcal{P}) = -\sum_{i \in \mathcal{I}} \mathcal{P}(i) \ln(\mathcal{P}(i)) \tag{5}$$

*The KL divergence of $\mathcal{W}$ from $\mathcal{P}$ (asymmetric), also known as the relative entropy, denoted $KL(\mathcal{P}||\mathcal{W})$, is a functional, defined here for non-empty SD $\mathcal{P}$ and SD $\mathcal{W}$:*

$$KL(\mathcal{P}||\mathcal{W}) := \sum_{i \in \mathcal{I}} \mathcal{P}(i) \ln \frac{\mathcal{P}(i)}{\mathcal{W}(i)}. \tag{6}$$

In both definitions, by convention (similar to the case for DIs in Cover & Thomas (1991)), when $\mathcal{P}(i) = 0$, we take the product $\mathcal{P}(i) \ln(x)$ to be 0. The divergence $KL(\mathcal{P}||\mathcal{W})$ can also be infinite (denoted $+\infty$) when $\mathcal{P}(i) > 0$ and $\mathcal{W}(i) = 0$.

**Lemma 4.** *Given DI $\mathcal{P}$ and SD $\mathcal{W}$, defined over the same finite set $\mathcal{I}$,*

$$LogLoss(\mathcal{W}|\mathcal{P}) = H(\mathcal{P}) + KL(\mathcal{P}||\mathcal{W}).$$

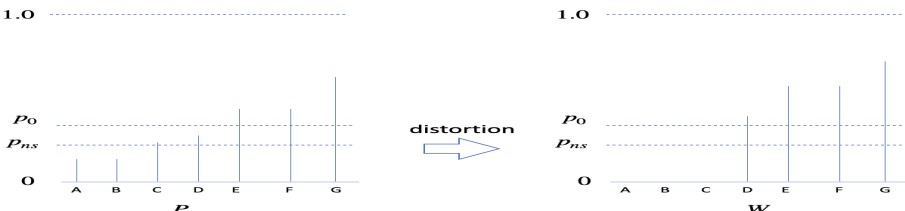

Figure 17: When using loglossRuleNS() *distortion* can occur, *i.e.* a SD $\mathcal{W}$ may obtain a lower loss than the true generating SD $\mathcal{P}$ (due to filtering, scaling, and bounding the loss). Section C.1 describes the extent of the distortion. In particular, Lemma 8 characterizes the properties of a minimizing SD $\mathcal{W}^*$: smaller items in $\mathcal{P}$ (those with lower probability, such as $A$, $B$ and $C$ above) may be zeroed, and have their probability mass transferred (spread, proportionately) onto other items in an optimal $\mathcal{W}^*$.

This is established when both are DIs, for example by Selten (1998), and the derivation does not change when $\mathcal{W}$ is a SD: $\mathrm{LogLoss}(\mathcal{W}|\mathcal{P}) = -\sum_{i\in\sup(\mathcal{P})} \mathcal{P}(i)\ln(\mathcal{W}(i)) = -\sum_{i\in\sup(\mathcal{P})} \mathcal{P}(i)\ln(\mathcal{W}(i)) - \mathrm{H}(\mathcal{P}) + \mathrm{H}(\mathcal{P})$ (*i.e.* add and subtract $\mathrm{H}(\mathcal{P})$), and noting that the first two terms is a rewriting of the KL(), establishes the lemma. Note that the entropy term in log-loss is a fixed positive offset: only KL() changes when we try different candidate SDs with the same underlying $\mathcal{P}$. From the properties of KL (Cover & Thomas, 1991), it follows that LogLoss is strictly proper (and asymmetric). We next formalize drawing items using a SD $\mathcal{P}$, which enables us to define taking expectations wrt a SD $\mathcal{P}$.

**Definition 2.** *Definitions of scaling, drawing from a SD, and a perfect marker:*

- *(scaling a SD) Let SD $\mathcal{P}$ be non-empty,* i.e. $a(\mathcal{P}) \in (0,1]$. *Then, for $\alpha > 0, \alpha \le \frac{1}{a(\mathcal{P})}$, $\mathcal{P}' := \alpha\mathcal{P}$ means the SD where $sup(\mathcal{P}') := sup(\mathcal{P})$ and $\forall i \in sup(\mathcal{P})$, $\mathcal{P}'(i) := \alpha\mathcal{P}(i)$. When $\alpha = \frac{1}{a(\mathcal{P})}$, $\alpha\mathcal{P}$ is a DI and one can repeatedly draw IID from it.*

- *Drawing item o from a non-empty SD $\mathcal{P}$,* i.e. $o \sim \mathcal{P}$, *means $a(\mathcal{P})$ of the time, drawing from $\alpha\mathcal{P}$, where $\alpha = \frac{1}{a(\mathcal{P})}$. and $1 - a(\mathcal{P})$ of the time generating a unique noise item. Repeatedly drawing items in this manner $N$ times generates a sequence $[o]_1^N$ using $\mathcal{P}$ ("IID drawing" from $\mathcal{P}$), and is denoted $[o]_1^N \sim \mathcal{P}$.*

- *(perfect marker) Given a non-empty SD $\mathcal{P}$, a perfect noise-marker wrt to $\mathcal{P}$, denoted $isNS_{\mathcal{P}}()$, marks an item i as noise iff $i \notin sup(\mathcal{P})$.*

Thus the perfect marker generates noise markings at about $u(\mathcal{P})$ fraction of the time on the stream $[o] \sim \mathcal{P}$. Next we show that LogLossNS() is equivalent to a *bounded* quasi-divergence denoted $\mathrm{KL}_{NS}()$, similar to plain log-loss reducing to plain KL divergence. First we define *augmentation* of an SD to a corresponding DI that has one extra item with all remaining probability mass. This allows us to draw from $\mathcal{P}$ as if it were a DI, in particular for the purpose of evaluating LogLossNS(): We show computing LogLossNS$(\mathcal{W}|\mathcal{P})$ is equivalent to computing a bounded KL() on the augmented versions[28] of $\mathcal{P}$ and FC$(\mathcal{W})$ (Lemma 5).

**Definition 3.** *Definitions of augmenting an SD to a DI, and bounded KL corresponding to LogLossNS():*

- *Given a non-empty SD $\mathcal{P}$ defined on $\mathcal{I} = \{1, \cdots, k\}, k \ge 1$, its augmentation (operation), denoted DI$(\mathcal{P})$, is a corresponding DI $\mathcal{P}'$, where $\mathcal{P}'$ is defined on $\mathcal{I}'$, $\mathcal{I}' = \mathcal{I} \cup \{0\}$, $\mathcal{P}'(0) = u(\mathcal{P})$, and $\mathcal{P}'(i) = \mathcal{P}(i)$, $\forall i \in I$.*

- *(**Bounded KL**) For non-empty SD $\mathcal{P}$ and SD $\mathcal{W}(p_{min} \in [0,1])$:*

$$KL_b(\mathcal{P}||\mathcal{W}) := \sum_{i \in sup(\mathcal{P})} \mathcal{P}(i)\ln\left(\frac{\mathcal{P}(i)}{\max(\mathcal{W}(i), p_{min})}\right) \ (bounded \ when \ p_{min} > 0).$$

---

[28]We are abusing notation in using DI() to also denote the augmentation operator applied to a SD $\mathcal{P}$ to make it a distribution, in addition to the related notion of referring to a distribution.

- (**KL() for LogLossNS**) $KL_{NS}(\mathcal{P}||\mathcal{W}) := KL_b(DI(\mathcal{P})||DI(FC(\mathcal{W})))$    *(where the $p_{min}$ of FC() is used in $KL_b()$).*

Both $KL_b(\mathcal{P}||\mathcal{W})$ and $KL_{NS}(\mathcal{P}||\mathcal{W})$ can be negative even when both $\mathcal{P}$ and $\mathcal{W}$ are DIs (the minimum is not 0 any more), and so they are not divergences, but still useful for scoring and comparisons. In particular, $KL_{NS}(\mathcal{P}||\mathcal{P})$ is not necessarily 0 ($FC()$ is only applied to the 2nd $\mathcal{P}$). As the lemma below shows, the addition of the positive entropy counteracts this growth and LogLossNS() is always in $[0, -\ln(p_{min})]$.

**Lemma 5.** *For any two SDs $\mathcal{P}$ and $\mathcal{W}$ defined on $\mathcal{I} = \{1, \cdots, k\}$, where $\mathcal{P}$ is non-empty, and using a perfect noise-marker , $isNS_{\mathcal{P}}()$, wrt to $\mathcal{P}$:*

$$LogLossNS(\mathcal{W}|\mathcal{P}) = H(DI(\mathcal{P})) + KL_{NS}(\mathcal{P}||\mathcal{W}).$$

The next property shows how plain KL() comparisons[29] (defined for SDs in Defn. 6) are affected by scaling, useful in seeing how FC() can affect the minimizer of $KL_{NS}()$.

**Lemma 6.** *(plain KL() scalings) For any non-empty SD $\mathcal{P}$ and SDs $\mathcal{W}$, and any $\alpha > 0$:*

1. $KL(\mathcal{P}||\alpha\mathcal{W}) = KL(\mathcal{P}||\mathcal{W}) + \ln(\alpha^{-1})a(\mathcal{P})$.

2. $KL(\alpha\mathcal{P}||\mathcal{W}) = \alpha KL(\mathcal{P}||\mathcal{W}) + \alpha \ln(\alpha)a(\mathcal{P})$.

3. $KL(\alpha\mathcal{P}||\alpha\mathcal{W}) = \alpha KL(\mathcal{P}||\mathcal{W})$.

As a consequence, if $KL(\mathcal{P}||\mathcal{W}_1) < KL(\mathcal{P}||\mathcal{W}_2)$ then $KL(\mathcal{P}||\alpha\mathcal{W}_1) < KL(\mathcal{P}||\alpha\mathcal{W}_2)$. More generally, the following, 'proportionate' spreading properties, can be established.

**Corollary 1.** *Scaling and spreading (adding or deducting mass) should be proportionate to SD $\mathcal{P}$ to minimize $KL(\mathcal{P}||.)$:*

1. *($\mathcal{P}$ is the unique minimizer over appropriate set) Given non-empty SD $\mathcal{P}$, $KL(\mathcal{P}||\mathcal{P}) = 0$, and for any $\mathcal{W} \neq \mathcal{P}$ with $a(\mathcal{W}) \leq a(\mathcal{P})$, $KL(\mathcal{P}||\mathcal{W}) > 0$.*

2. *(same minimizer for $\mathcal{P}$ and its multiple) Let non-empty SDs $\mathcal{P}_1$ and $\mathcal{P}_2$ be such that $\mathcal{P}_1 = \alpha\mathcal{P}_2$ for a scalar $\alpha > 0$, and consider any non-empty set $S$ of SDs . For any two SDs $\mathcal{W}_1, \mathcal{W}_2 \in S$, $KL(\mathcal{P}_1||\mathcal{W}_1) < KL(\mathcal{P}_1||\mathcal{W}_2) \Leftrightarrow KL(\mathcal{P}_2||\mathcal{W}_1) < KL(\mathcal{P}_2||\mathcal{W}_2)$ (thus, a SD $\mathcal{W} \in S$ is a minimizer for both or for neither).*

3. *Given a non-empty SD $\mathcal{P}$, among SD $\mathcal{W}$ such that $a(\mathcal{W}) = s$, the one that minimizes $KL(\mathcal{P}||\mathcal{W})$ is proportionate to (or a multiple of) $\mathcal{P}$, i.e. $\forall i \in \mathcal{I}, \mathcal{W}(i) = \frac{s}{a(\mathcal{P})}\mathcal{P}(i)$ (when $s = 0$ this becomes vacuous).*

We can now better describe what a minimizer $\mathcal{W}^*$ of $KL_b(\mathcal{P}||.)$ looks like in terms of $\mathcal{P}$, where $a(\mathcal{W}^*) = a(\mathcal{P})$ (and $p_{min} > 0$): we show that we have all-or-nothing deductions: an item's probability (in $\mathcal{W}^*$), if reduced (compared to $\mathcal{P}$), it is 0 (*zeroed*), otherwise stays the same (if no item zeroed) or increases. But, firstly, when $a(\mathcal{P}) \leq p_{min}$ (degeneracy) many $\mathcal{W}$s are minimizers:

**Lemma 7.** *(Minimizer in the degenerate case) Given threshold $p_{min}, p_{min} \in (0, 1)$ and nonempty SD $\mathcal{P}$ such that $a(\mathcal{P}) \leq p_{min}$ (degenerate), then for any SD $\mathcal{W}$ with $a(\mathcal{W}) \leq a(\mathcal{P})$ (including $\mathcal{P}$ and the empty SD ), $KL_b(\mathcal{P}||\mathcal{W}) = -a(\mathcal{P}) \ln(p_{min})$.*

If $\mathcal{P}$ is non-degenerate, we show that a minimizer must exist, and show some of the properties it must have (see also Fig. 17).

---

[29]For this lemma, we can extend the definition of KL() to non-negative valued $\mathcal{P}$ and $\mathcal{W}$ (no constraint on the sum, unlike plain SDs), or assume $\alpha$ is such that $\alpha\mathcal{W}$ and $\alpha\mathcal{P}$ remain a SD.

**Lemma 8.** *For threshold $p_{min} \in (0,1)$ and SD $\mathcal{P}$ such that $a(\mathcal{P}) > p_{min}$, where we seek a minimizer of $KL_b(\mathcal{P}||\mathcal{W})$ over the set $S$ of SDs $\mathcal{W}$ such that $a(\mathcal{W}_1) \leq a(\mathcal{P})$ (and wlog we need only consider $sup(\mathcal{W}) \subseteq sup(\mathcal{P})$):*

1. *(all positive probability items are above $p_{min}$) For any $\mathcal{W}_1 \in S$, if there is an item $i$ where $\mathcal{W}_1(i) \in (0, p_{min}]$, then there is a SD $\mathcal{W}_2 \in S$, such that $\forall i \in sup(\mathcal{W}_2), \mathcal{W}_2(i) > p_{min}$, and $KL_b(\mathcal{P}||\mathcal{W}_2) < KL_b(\mathcal{P}||\mathcal{W}_1)$. Therefore, in minimizing $KL_b(\mathcal{P}||.)$, we need only consider the set $S_2 = \{\mathcal{W}|\mathcal{W} \in S, a(\mathcal{W}) = a(\mathcal{P}), \text{ and } \forall i \in sup(\mathcal{W}), \mathcal{W}(i) > p_{min}\}$ ($S_2$ is not empty as $\mathcal{P}$ is non-degenerate).*

2. *(existence and proportionate increase) The set $S^* \subseteq S_2$ of minimizers of $KL_b(\mathcal{P}||.)$ is not empty, and for any $\mathcal{W}^* \in S^*$, and for some fixed multiple $r \geq 1$, $\forall i \in sup(\mathcal{W}^*), \mathcal{W}^*(i) = r\mathcal{P}(i)$.*

3. *(order is respected) For any minimizer $\mathcal{W}^*$ and any two items $i$ and $j$, when $\mathcal{P}(i) < \mathcal{P}(j)$, if $\mathcal{W}^*(i) > 0$, then $\mathcal{W}^*(j) > 0$ (and, from part 2, $\mathcal{W}^*(j) > \mathcal{W}^*(i)$).*

Thus, items in $sup(\mathcal{P})$ are zeroed in order of their probability magnitude and any non-zeroed item can only increase in probability, and proportionately (unique minimizer $\mathcal{W}^*$ if un-equal item probabilities), and the above suggest simple greedy methods for finding a $\mathcal{W}^*$. The next properties (existence of a threshold $p_0$) give us additional information about which items can be zeroed.

**Lemma 9.** *With a DI $\mathcal{P}$ and $p_{min} \in (0,1)$, if item $i$ has probability $\mathcal{P}(i) \geq p_0$, where $p_0$ is such that $p_{min} = p_0(1-p_0)^{\frac{1-p_0}{p_0}}$, then $\mathcal{W}^*(i) \geq \mathcal{P}(i)$ (item $i$ is not zeroed) in any minimizer $\mathcal{W}^*$ of $KL_b(\mathcal{P}||)$. If $i \in sup(\mathcal{P})$ has the smallest probability in $sup(\mathcal{P})$ and $\mathcal{P}(i) < p_0$, then $i$ is zeroed in some minimizer $\mathcal{W}^*$ ($\mathcal{W}^*(i) = 0$) and if it is the unique minimum, then $\mathcal{W}^*(i) = 0$ in any minimizer $\mathcal{W}^*$.*

$p_{min} = p_0(1-p_0)^{\frac{1-p_0}{p_0}}$ implies that $2p_{min} \leq p_0 \leq ep_{min} \approx 3p_{min}$. Thus, as a rule of thumb, as long as $p \geq 3p_{min}$, an item with $p$ is not zeroed. Note that the lemma is established for DIs only: for the analysis of extent of distortion below, we can assume $\mathcal{P}$ is a DI (the first argument in $KL_{NS}()$ is augmented to a distribution). However, we expect the bound can be extended to strict SDs as well (this is the case for SDs with only two items).

The following lemma specifies extent of increase in an item's probability in any minimizer of $KL_b(\mathcal{P}||.)$, and further characterizes the properties of a minimizer.

**Lemma 10.** *Given any DI $\mathcal{P}$ and $p_{min} \in (0,1)$, and any DI $\mathcal{W}$ with $sup(\mathcal{W}) \subseteq sup(\mathcal{P})$ and with proportionate spread according to $\mathcal{P}$, let $Z := sup(\mathcal{P}) - sup(\mathcal{W}^*)$ (the zeroed items or the difference of the two support sets), and $p_z(\mathcal{W}) := \sum_{i \in Z} \mathcal{P}(i)$, thus $p_z(\mathcal{W})$ is the total mass of the zeroed items (the shifted mass), $p_z(\mathcal{W}) \geq 0$. (**part 1**) We have for any $i \in sup(\mathcal{W}), \mathcal{W}(i) = \frac{\mathcal{P}(i)}{1 - p_z(\mathcal{W})}$. (**part 2**) Furthermore, let $S_g$ be the set of all such proportionate $\mathcal{W}$ with $min_{i \in sup(\mathcal{W})}\mathcal{P}(i) \geq p_0$. Then any minimizer DI $\mathcal{W}^*$ of $KL_b(\mathcal{P}||.)$ is in $S_g$, and has the largest support and the smallest shifted mass among such (i.e. $p_z(\mathcal{W}^*) \leq p_z(\mathcal{W})$ and $|sup(\mathcal{W}^*)| \geq |sup(\mathcal{W})|$ for any $\mathcal{W} \in S_g$).*

When the total probability shift $p_z$ is small, the increase $\frac{1}{1-p_z}$ in probability of any non-zeroed item is also small.

We have assumed a perfect noise-marker (referee) in all the above. Assuming $p_{min} = 0.01$, a simple practical noise-marker, *e.g.* keeping a history of the last 100 time points (a box predictor), will have some probability of making false positive markings (a salient item marked noise) and false negative errors (items below $p_{min}$). The error probability goes down as an item becomes more salient or more noisy (its probability gets farther from the $p_{min}$ threshold), and for items with probability near the boundary, the loss would be similar whether or not $\ln(p_{min})$ is used. Given a SD $\mathcal{P}$ on $\mathcal{I} = \{1, 2, \cdots\}$, the *ideal-threshold* noise-marker, or the **threshold marker** for short, marks an item $i$ noise iff $\mathcal{P}(i) \leq p_{min}$. Considering how LogLossNS() (or loglossRuleNS()) works when this threshold marker is used, the marker in effect converts a SD $\mathcal{P}$ to a DI $\mathcal{P}'$ where for any item $i$ with $\mathcal{P}(i) \leq p_{min}$, its probability, together with $u(\mathcal{P})$, is transferred to item 0 when computing $KL_{NS}(\mathcal{P}'||.)$ (recall that $DI(\mathcal{P})$, in definition 3, only transferred $u(\mathcal{P})$ to item 0). Thus $\mathcal{P}'$ is a

DI where $\forall i \in \sup(\mathcal{P}')$ if $i \neq 0$, then $\mathcal{P}'(i) > p_{min}$. It is possible that $\mathcal{P}'(0) < p_{min}$, but as we use FC(), we could limit our analysis to DI $\mathcal{P}'$ where $\forall i \in \sup(\mathcal{P}'), \mathcal{P}'(i) \geq p_{min}$. The next lemma shows that the distortion ratio $\frac{\mathcal{W}^*(i)}{\mathcal{P}(i)}$ is upper bounded by about 3 in this setting (where the probabilities are above $p_{min}$).

**Lemma 11.** *Given any DI $\mathcal{P}$ with $\min_{i \in sup(\mathcal{P})} \mathcal{P}(i) \geq p_{min}$, for any minimizer $\mathcal{W}^*$ of $KL_b(\mathcal{P}||.)$, the distortion ratio $\frac{\mathcal{W}^*(i)}{\mathcal{P}(i)} \leq \frac{p_0}{(1-p_0)p_{min}} < \frac{3}{(1-p_0)}$.*

When $p_{min} = 0.01$ the distortion ratio is bounded by 3.1. A practical marker can be viewed as a noisy version of the (ideal) threshold marker. Given threshold $p_{min} > 0$, if we find that the predictions $\mathcal{W}^{(t)}$ have mostly unallocated mass or probabilities close to $p_{min}$, then $p_{min}$ and $p_{min}$ may need to be lowered (*e.g.* from 0.01 to 0.001) for improved evaluation as well as better (finer) prediction.[30]

## C.2   Alternatives to Bounded Logloss

Quadratic (Brier) loss is proper and enjoys attractive properties such as symmetry and not blowing up on zero predictions, but is not sufficiently sensitive towards low probabilities for supporting a diversity of probability ranges. Take the actual distribution $P = \{A:0.5, B:0.05, C:0.45\}$, and two proposed ones $\mathcal{W}_2 = \{A:0.5, B:0.0, C:0.5\}$, and $\mathcal{W}_3 = \{A:0.45, B:0.05, C:0.5\}$. $\mathcal{W}_2$ completely ignores event $B$ (a one in 20 event), but $\mathcal{W}_2$ and $\mathcal{W}_3$ have the same Brier loss, of 0.55 (equivalently, equal Euclidean distance to $P$), and $\mathcal{W}_2' = \{A:0.46, B:0.0, C:0.54\}$ has a lower loss than $\mathcal{W}_3$ (for further development, see Appendix A.1 of Madani (2024)). log-loss is also sensitive to higher probabilities, but does not suffer this extent of insensitivity. If all probabilities were in a narrower range, say 0.1 to 0.5, perhaps Brier would be adequate. Our GitHub code supports reporting Brier loss as well, and comparisons based on Brier (*e.g.* in synthetic experiments) has not changed our findings. See also Sect. B.3, which includes Brier scores. We also looked at other options, such as separating zero- and non-zero-probability predictions (*e.g.* reporting two numbers), but such made comparisons among two or more SMAs difficult and could lead to impropriety.

## C.3   Proofs for Section C.1

*Proof of Lemma 5.* We first explain the different parts of LogLossNS$(\mathcal{W}|\mathcal{P})$, *i.e.* $\mathbb{E}_{o \sim \mathcal{P}}(\text{loglossRuleNS}(o, \mathcal{W}, isNS_{\mathcal{P}}(o)))$, to show that it is equivalent to comparing two augmented DIs under KL().

Let $\mathcal{W}' = FC(\mathcal{W})$ and $\mathcal{P}' = DI(\mathcal{P})$. An item $o \sim \mathcal{P}$ (drawn from $\mathcal{P}$), is salient with probability a$(\mathcal{P})$ ($o \in \sup(\mathcal{P})$), and otherwise is marked noise by $isNS_{\mathcal{P}}()$ (perfect noise-marker), and it is important to note that $o$ does not occur in $\mathcal{I}$ when marked noise ($\mathcal{W}'(o) = \mathcal{W}(o) = \mathcal{P}(o) = 0$ when $isNS_{\mathcal{P}}(o)$ is true), with our uniqueness assumption when generating noise items. When we use loglossRuleNS(), the score in the noise case is $-\ln(\text{u}(\mathcal{W}'))$. This case occurs u$(\mathcal{P})$ of the time, and a salient item $i$, $i \in \mathcal{I}$, occurs $\mathcal{P}(i)$ of the time, thus LogLossNS$(\mathcal{W}|\mathcal{P}) = -\text{u}(\mathcal{P})\ln(\max(\text{u}(\mathcal{W}'), p_{min})) - \sum_{i \in I} \mathcal{P}(i)\ln(\max(\mathcal{W}'(i), p_{min}))$. We have that u$(\mathcal{W}') \geq p_{min}$ (from the scaling down in FC()), or $\max(\text{u}(\mathcal{W}'), p_{min}) = \text{u}(\mathcal{W}')$. Adding and subtracting H$(\mathcal{P}')$ establishes the equivalence:

$$\text{LogLossNS}(\mathcal{W}|\mathcal{P}) = -\text{u}(\mathcal{P})\ln(\text{u}(\mathcal{W}')) - \sum_{i \in I} \mathcal{P}(i)\ln(\max(\mathcal{W}'(i), p_{min})) - \text{H}(\mathcal{P}') + \text{H}(\mathcal{P}')$$

$$= -\text{u}(\mathcal{P})\ln(\text{u}(\mathcal{W}')) - \sum_{i \in I} \mathcal{P}(i)\ln(\max(\mathcal{W}'(i), p_{min})) + \sum_{i \in I'} \mathcal{P}'(i)\ln(\mathcal{P}'(i)) + \text{H}(\mathcal{P}')$$

$$= \text{u}(\mathcal{P})\ln\frac{\text{u}(\mathcal{P})}{\text{u}(\mathcal{W}')} + \sum_{i \in I} \mathcal{P}(i)\ln\frac{\mathcal{P}(i)}{\max(\mathcal{W}'(i), p_{min})} + \text{H}(\mathcal{P}')$$

$$= \text{KL}(\mathcal{P}'||\text{DI}(\mathcal{W}')) + \text{H}(\mathcal{P}') = \text{KL}_{NS}(\mathcal{P}||\mathcal{W}) + \text{H}(\text{DI}(\mathcal{P})).$$

The 2nd to 3rd line follow from H$(\mathcal{P}') = -\sum_{i \in I'} \mathcal{P}'(i)\ln(\mathcal{P}'(i)) = -\text{u}(\mathcal{P})\ln(\text{u}(\mathcal{P})) - \sum_{i \in I} \mathcal{P}(i)\ln(P(i)))$.  $\square$

---

[30]However, this lowering, incurring extra memory, does not guarantee that further salient items are discovered. For instance, all remaining items may be (pure or absolute) noise, appearing only once in the stream.

*Proof of Lemma 6.* The $\alpha$ multiplier comes out, in all cases, yielding a fixed offset for first 2 cases, and a positive multiplier for the 2nd and 3rd cases:

$$\mathrm{KL}(\mathcal{P}||\alpha\mathcal{W}) = \sum_{i\in\mathcal{I}}\mathcal{P}(i)\ln(\frac{\mathcal{P}(i)}{\alpha\mathcal{W}(i)}) = \sum_{i\in\mathcal{I}}\mathcal{P}(i)\ln(\frac{\mathcal{P}(i)}{\mathcal{W}(i)}\frac{1}{\alpha}) = \sum_{i\in\mathcal{I}}\mathcal{P}(i)(\ln(\frac{\mathcal{P}(i)}{\mathcal{W}(i)}) + \ln(1/\alpha))$$

$$= \sum_{i\in\mathcal{I}}\mathcal{P}(i)\ln\frac{\mathcal{P}(i)}{\mathcal{W}(i)} + \ln(\alpha^{-1})\sum_{i\in\mathcal{I}}\mathcal{P}(i) = \mathrm{KL}(\mathcal{P}||\mathcal{W}) + \ln(\alpha^{-1})\mathrm{a}(\mathcal{P}).$$

$$\mathrm{KL}(\alpha\mathcal{P}||\mathcal{W}) = \sum_{i\in\mathcal{I}}\alpha\mathcal{P}(i)\ln(\frac{\alpha\mathcal{P}(i)}{\mathcal{W}(i)}) = \alpha\left(\sum_{i\in\mathcal{I}}\mathcal{P}(i)\ln\frac{\mathcal{P}(i)}{\mathcal{W}(i)} + \ln(\alpha)\sum_{i\in\mathcal{I}}\mathcal{P}(i)\right).$$

$$\mathrm{KL}(\alpha\mathcal{P}||\alpha\mathcal{W}) = \sum_{i\in\mathcal{I}}\alpha\mathcal{P}(i)\ln(\frac{\alpha\mathcal{P}(i)}{\alpha\mathcal{W}(i)}) = \alpha\sum_{i\in\mathcal{I}}\mathcal{P}(i)\ln(\frac{\mathcal{P}(i)}{\mathcal{W}(i)}) = \alpha\mathrm{KL}(\mathcal{P}||\mathcal{W}).$$

$\square$

*Proof of Corollary 1 .* (**part 1**) When $\mathcal{P}$ is a DI, among $\mathcal{W} \neq \mathcal{P}$ that are *DI* too, the property $\mathrm{KL}(\mathcal{P}||\mathcal{W}) > 0$ holds (Cover & Thomas, 1991). If $\mathcal{W}$ is a strict SD, on at least one item $i$, $\mathcal{W}(i) < \mathcal{P}(i)$, and we can repeat increasing all such $\mathcal{W}(i)$ in some order until $\mathcal{W}(i) = \mathcal{P}(i)$ or $\mathcal{W}$ becomes a DI (finitely many such $i$), lowering the distance (the log ratio $\frac{\mathcal{P}(i)}{\mathcal{W}(i)}$). We conclude for any $\mathcal{W} \neq \mathcal{P}$, $\mathrm{KL}(\mathcal{P}||\mathcal{W}) > \mathrm{KL}(\mathcal{P}||\mathcal{P}) = 0$. When $\mathcal{P}$ is a nonempty strict SD: We have $\mathrm{KL}(\mathcal{P}||\mathcal{P}) = \sum\mathcal{P}(i)\ln\frac{\mathcal{P}(i)}{\mathcal{P}(i)} = 0$. We can scale $\mathcal{P}$ by $\alpha = \frac{1}{\mathrm{a}(\mathcal{P})}$ to get a DI , and from the property of $\mathrm{KL}(\mathcal{P}||.)$ for DI $\mathcal{P}$, and using Lemma 6, we conclude that for any other SD $\mathcal{W} \neq \mathcal{P}$, with $\mathrm{a}(\mathcal{W}) \leq \mathrm{a}(\mathcal{P})$ (and thus $\alpha\mathcal{W}$ remains a SD), must yield a higher (positive) $\mathrm{KL}(\mathcal{P}||\mathcal{W})$: $\mathrm{KL}(\mathcal{P}||\mathcal{W}) = \frac{\mathrm{KL}(\alpha\mathcal{P}||\alpha\mathcal{W})}{\alpha} > 0$ (using Lemma 6, and $\mathrm{KL}(\alpha\mathcal{P}||\alpha\mathcal{W}) > 0$, from first part of this claim).

(**part 2**) Let $\Delta_1 := \mathrm{KL}(\mathcal{P}_1||\mathcal{W}_1) - \mathrm{KL}(\mathcal{P}_1||\mathcal{W}_2)$ and $\Delta_2 := \mathrm{KL}(\mathcal{P}_2||\mathcal{W}_1) - \mathrm{KL}(\mathcal{P}_2||\mathcal{W}_2)$. $\Delta_1 = \mathrm{KL}(\alpha\mathcal{P}_2||\mathcal{W}_1) - \mathrm{KL}(\alpha\mathcal{P}_2||\mathcal{W}_2) = \alpha(\mathrm{KL}(\mathcal{P}_2||\mathcal{W}_1) - \mathrm{KL}(\mathcal{P}_2||\mathcal{W}_2))$ (from part 2, Lemma 6, $\alpha\ln(\alpha)\ln(\mathrm{a}(\mathcal{P}_2))$ canceling). Therefore, $\Delta_1 < 0 \Leftrightarrow \Delta_2 < 0$.

(**part 3**) This is a consequence of parts 1 and 2 where the set $S$ includes $\mathcal{P}_1$, where $P_1$ is proportionate to $P_2$, and $P_1$ minimizes $\mathrm{KL}()$ to itself among $\mathcal{W} \in S$ (part 1). $\square$

*Proof of Lemma 7.* For any such $\mathcal{W}$, $\mathrm{KL}_b(\mathcal{P}||\mathcal{W}) = \sum_{i\in\sup(\mathcal{P})} -\mathcal{P}(i)\ln(p_{min})$ (as $\forall i, \mathcal{W}(i) \leq \mathrm{a}(\mathcal{W}) \leq p_{min}$), thus $\mathrm{KL}_b(\mathcal{P}||\mathcal{W}) = -\mathrm{a}(\mathcal{P})\ln(p_{min})$ $\square$

*Proof of Lemma 8.* (**part 1**) Say $\mathcal{W}_1$ has one or more items with low probability $\leq p_{min}$, call the set $\mathcal{I}_2$, $\mathcal{I}_2 := \{i|\mathcal{W}_1(i) \leq p_{min}\}$, with total probability $b$ (thus $b := \sum_{i\in\mathcal{I}_2}\mathcal{W}(i)$). We can also assume $\mathrm{a}(\mathcal{W}_1) = \mathrm{a}(\mathcal{P})$ (if less, we can also shift the difference onto receiving item $j$ in this argument). Then if $\mathcal{W}_1$ also has an item $j$ with probability above $p_{min}$, shift all the mass $b$ to item $j$. Otherwise, shift all the mass $b$ to a single item $j$ in $\mathcal{I}_2$ (pick any item). In either case, we have $\mathrm{KL}_b(\mathcal{P}||\mathcal{W}_2) < \mathrm{KL}_b(\mathcal{P}||\mathcal{W}_1)$: In the first case, the cost (*i.e.* $-\mathcal{P}(i)\ln(\frac{\mathcal{P}(i)}{\max(Q(i),p_{min})})$) is not changed for items in $\mathcal{I}_2$ ($-b\ln(p_{min})$), while for item $j$ the cost is lowered. In the second case, we must have $b > p_{min}$ ($\mathcal{P}$ is non-degenerate), and the cost for the receiving item $i$ improves (as we must have $Q_2(i) > p_{min}$), while for others in $\mathcal{I}_2$ it is not changed.

(**part 2**) The set $S_2$ (from part 1), can be partitioned into finitely many subsets (ie disjoint sets whose union is $S_2$), each partition member corresponding to a non-empty subset of $\sup(\mathcal{P})$. For instance all those that have positive probability on item 1 only (greater than $p_{min}$ by definition of $S_2$) (support of size 1) define one partition subset (we get $|\sup(\mathcal{P})|$ such subsets corresponding to singletons). The size of the support set $k$ of a SD in $S_2$, $k \leq |\sup(\mathcal{P})|$, can be large to the extent that $\frac{\mathrm{a}(\mathcal{P})}{k} \geq p_{min}$ is satisfied ($k = 1$ works, but larger $k$ may yield valid SDs in $S_2$ as well). On each such partition set $\mathcal{S}_k$), $\mathrm{KL}_b(\mathcal{P}||.)$ becomes equivalent to $\mathrm{KL}(\mathcal{P}||.)$, in the following sense: If $\mathcal{S}_k$ is a partition, then $\forall\mathcal{W}_1, \mathcal{W}_2 \in \mathcal{S}_k, \mathrm{KL}_b(\mathcal{P}||Q_1) - \mathrm{KL}_b(\mathcal{P}||Q_2) = \mathrm{KL}(\mathcal{P}||Q_1) - \mathrm{KL}(\mathcal{P}||Q_2)$. From Corollary 1, the total mass from elements that are zeroed (if any), *i.e.* $\sup(\mathcal{P}) - \sup(\mathcal{W}_1)$, is spread proportionately on $\sup(\mathcal{W}_1)$ to minimize $\mathrm{KL}(\mathcal{P}||.)$ over a partition subset. Since we have a finitely

many partitions (the size of a powerset at most), and each yields a well-defined minimizer of $\mathrm{KL}_b(\mathcal{P}||.)$, we obtain one or more minimizers for the entire $S_2$ and therefore $S$.

(**part 3**) Wlog consider items 1 and 2, $p_1 = \mathcal{P}(1)$ and $p_2 = \mathcal{P}(2)$, where $p_1 < p_2$, and assume in SD $\mathcal{W}_1 \in S_2$ (with support size $|\sup(\mathcal{W}_1)| \geq 1$), item 1 has an allocation $T > p_{min}$ (as $\mathcal{W}_1 \in S_2$), while $\mathcal{W}_1(2) < \mathcal{W}_1(1)$. We need only consider the case $\mathcal{W}_1(2) = 0$: if $\mathcal{W}_1(2) > 0$, then $\mathcal{W}_1(2) > p_{min}$ as $\mathcal{W}_1 \in S_2$, and proportionate increase from part 2 establishes the result. Assuming $\mathcal{W}_1(2) = 0$, we can 'swap' items 1 and 2, to get SD $\mathcal{W}_2 \in S_2$, and swapping improves $\mathrm{KL}_b()$, *i.e.* letting $\Delta := \mathrm{KL}_b(\mathcal{P}||\mathcal{W}_1) - \mathrm{KL}_b(\mathcal{P}||\mathcal{W}_2)$, we must have $\Delta > 0$:

$$\Delta = (p_2 \ln \frac{p_2}{p_{min}} + p_1 \ln \frac{p_1}{T}) - (p_1 \ln \frac{p_1}{p_{min}} + p_2 \ln \frac{p_2}{T}) \qquad \text{(all other terms cancel)}$$
$$= -p_2 \ln(p_{min}) - p_1 \ln(T) + p_1 \ln(p_{min}) + p_2 \ln(T) = (p_2 - p_1)(\ln(T) - \ln(p_{min})) > 0$$

The last step (conclusion) follows from our assumptions that $p_2 > p_1$ and $T > p_{min}$. Note that $\mathcal{W}_2$ can further be improved by a proportionate spread (the allotment $T$ to item 2 increased). $\qquad \square$

For establishing Lemma 9 suppose the DI $\mathcal{P}$ has two items, with probabilities $p$ and $1-p$, *i.e.* $\mathcal{P} = \{1:p, 2:1-p\}$, where $p \leq 1 - p$, and $p > p_{min}$, and we want to see how high $p$ can be and yet distortion remains possible, *i.e.* item 1 is zeroed and its mass shifted to item 2, and $\Delta := \mathrm{KL}_b(\mathcal{P}||\mathcal{P}) - \mathrm{KL}_b(\mathcal{P}||\mathcal{W}) > 0$: how high $p$, $p > p_{min}$, can be and yet we can get $\Delta > 0$. $\Delta = 0 - (p \ln \frac{p}{p_{min}} + (1-p) \ln \frac{1-p}{1})$, thus $\Delta > 0$, when $p \ln(p_{min}) > p \ln(p) + (1-p) \ln(1-p)$, or $p_{min} > p(1-p)^{\frac{1-p}{p}}$. Thus the threshold $p_0$ is such that $p_{min} = p_0(1-p_0)^{\frac{1-p_0}{p_0}}$. Now, with $r := (1-p_0)^{\frac{1-p_0}{p_0}}$, $r > 1 - p_0$ for $p_0 > 0.5$ ($r \to 1.0$ as $p_0 \to 1$), and $r < 1 - p_0 < 1$ when $p_0 < 0.5$ (and $r = 1 - p_0 = p_0 = 0.5$ when $p_0 = 0.5$) (note that in our set up, $p_0 \leq 0.5$). Thus, indeed $p_{min} < p_0$. We can set $p_0$ and see how low $p_{min}$ should be (or, otherwise, solve for $p_0$). Considering the ratio $\frac{p_{min}}{p_0} = \frac{p_0(1-p_0)^{(1-p_0)/p_0}}{p_0} = r$, where we have defined $r = (1-p_0)^{\frac{1-p_0}{p_0}}$, and as $p_0 \to 0$, $\frac{p_{min}}{p_0} \to \frac{1}{e}$ ($e$ denotes the base of the natural logarithm), while as $p_0 \to 0.5$, $\frac{p_{min}}{p_0} \to \frac{1}{2}$, or:

$$2p_{min} \leq p_0 \leq e p_{min} \approx 3 p_{min}. \tag{7}$$

The case of more items is similar to two items, and yields the same distortion threshold $p_0$: given DI $\mathcal{P}$ assume all its items have probability above $p_{min}$, and $\Delta := \mathrm{KL}_b(\mathcal{P}||\mathcal{P}) - \mathrm{KL}_b(\mathcal{P}||\mathcal{W}^*) = 0 - \mathrm{KL}_b(\mathcal{P}||\mathcal{W}^*)$, and we are wondering about the relation of say $p_1 = \mathcal{P}(1)$ and $p_{min}$ when $\Delta > 0$.

$$\Delta = -\left(p_1 \ln(p_1/p_{min}) + \sum_{i \geq 2} p_i \ln(p_i/(p_i + \frac{p_i}{1-p_1}p_1))\right)$$

(the above is the proportionate spread of $p_1$ onto others in $\mathcal{W}^*$)

$$= p_1 \ln(p_{min}) - p_1 \ln(p_1) - \sum_{i \geq 2} p_i \ln(p_i) + \sum_{i \geq 2} p_i \ln(p_i + \frac{p_i}{1-p_1}p_1)$$
$$= p_1 \ln(p_{min}) - \sum_{i \geq 1} p_i \ln(p_i) + \sum_{i \geq 2} p_i \ln(\frac{p_i}{1-p_1})$$
$$= p_1 \ln(p_{min}) - \sum_{i \geq 1} p_i \ln(p_i) + \sum_{i \geq 2} p_i \ln(p_i) - \ln(1-p_1) \sum_{i \geq 2} p_i$$
$$= p_1 \ln(p_{min}) - \ln(1-p_1) \sum_{i \geq 1} p_i - p_1 \ln(\frac{p_1}{1-p_1}) = p_1 \ln(p_{min}) - \ln(1-p_1) - p_1 \ln(\frac{p_1}{1-p_1})$$

And $\Delta > 0$ implies $p_1 \ln(p_{min}) > \ln(1-p_1) + p_1 \ln(\frac{p_1}{1-p_1})$, or when $p_1$ is low enough such that $p_{min} > p_1(1-p_1)^{\frac{1-p_1}{p_1}}$. This is the same bound or threshold as the two-item case, and we summarize the properties in the following lemma.

*Proof of Lemma 9.* The result follows from the above derivation and the ordering properties specified in Lemma 8. □

*Proof of Lemma 10.* For a DI $\mathcal{W}$, let $p_z$ denote $p_z(\mathcal{W})$, and let $s := \sum_{i \in \sup(\mathcal{W})} \mathcal{P}(i)$. From the definition of proportionate spread, we add $\frac{\mathcal{P}(i)}{s} p_z$ to $\mathcal{P}(i)$. We have $s = 1 - p_z$, therefore, $\mathcal{W}(i) = \mathcal{P}(i) + \frac{\mathcal{P}(i)}{1-p_z} p_z = \mathcal{P}(i)(1 + \frac{p_z}{1-p_z}) = \frac{\mathcal{P}(i)}{1-p_z}$. (**proof of part 2**) From Lemma 8, $p_z$ is proportionately spread onto non-zeroed items in $\mathcal{W}^*$ as well. Consider $Q_1$ and $\mathcal{W}^*$, both in $S_g$, and assume $\sup(\mathcal{W}_1) > \sup(\mathcal{W}^*)$ (proof by contradiction). By shifting the lowest probability in $S_1$ to remaining (higher probability) items in the support, and repeating, we should get to $\mathcal{W}^*$ (or an equivalent, in case of ties). But each shift results in an inferior $\mathcal{W}$ because of our assumption that all probabilities are $\geq p_0$ and the shifts only increase the probability on items that remain (non-zeroed items). □

*Proof of Lemma 11.* Let $m$ be the mass of non-zeroed items in $\mathcal{P}$ ($m = 1 - p_z(\mathcal{P})$, $p_z$ was also used in Lemma 10), then we need to bound the distortion ratio $\frac{1}{m}$ which is the ratio by which each item $i$ in $\sup(\mathcal{W}^*)$ goes up by ($\frac{\mathcal{W}^*(i)}{\mathcal{P}(i)}$). Let $j$ be any item that is zeroed (at least one such item exists, otherwise the distortion ratio is 1). Let $x := \mathcal{P}(j)$, and we have $x \geq p_{min}$. Then spreading 1.0 over one additional item, $\sup(\mathcal{W}^*) \cup \{j\}$, proportionately, would fail to take the probability of $j$ to $p_0$ (by Lemma 10, $\mathcal{W}^*$ would not be optimal). Proportional allocation over $\sup(\mathcal{W}^*) \cup \{j\}$ yields $\frac{x}{x+m}$ for item $j$ and we must have $\frac{x}{x+m} < p_0 \Rightarrow x < p_0 x + p_0 m \Rightarrow \frac{(1-p_0)x}{p_0} < m$ and as $x > p_{min}$, we get $\frac{1}{m} < \frac{p_0}{(1-p_0)x} \leq \frac{p_0}{(1-p_0)p_{min}}$. □

# D    Additional Synthetic Experiments

We first show comparisons in the stationary setting, then describe the muli-item generation process and present additional results, including pairing (for sign tests) in that setting.

## D.1    Tracking a Single Item, Stationary

All the prediction techniques are based on estimating the probability for each item separately (treating all other observations as negative outcomes), so we begin with assessing the quality of the predictions for a single item in the binary stationary setting of Sect. 2.1.1. Thus, when $p^* = 0.1$, about 10% of the sequence is 1, the rest 0. Table 9 presents the deviation rates of Qs, EMA, and DYAL, under a few parameter variations, and for $p^* \in \{0.1, 0.05\}$. Sequences are each 10k time points long, and deviation rates are averaged over 200 such sequences.[31]

Under this stationarity setting, higher qcap helps the Qs technique: Qs with qcap 10 does better than qcap of 5, but, specially for $d = 1.5$, both tend to substantially lag behind the best of the EMA variants. EMA with harmonic decay, with an appropriately low $\beta_{min}$, does best across all $p^*$. If we anticipate that the useful items to predict will have probabilities in the 0.01 to 1.0 range, in a stationary world, then setting $\beta_{min}$ for harmonic EMA to a low value, $\frac{0.01}{k}$, where $k \geq 10$, is adequate. In this stationary and binary setting, the complexity of DYAL is not needed, and harmonic EMA is sufficient. Still DYAL is the second best. Static EMA is not flexible enough, and one has to anticipate what $p^*$ is and set the (fixed) rate appropriately (for instance, when $p^* = 0.01$, EMA with the same $\beta = 0.01$ is not appropriate, resulting in too much variance). Finally, we observe that the deviation rates, as well as the variances, for any method, degrade (increase) somewhat as $p^*$ is lowered from 0.1 to 0.01. In ten thousand draws (during sequence generation), there are fewer positive observations with lower $p^*$, and estimates will have higher variance.

## D.2    Synthetic Non-Stationary Multi-Item Generation

Fig. 18 provides pseudocode of the main functions for generating item sequences under non-stationarity. As in the above, we think of a non-stationary sequence as a concatenation of "stable" (stationary) subsequences,

---

[31]The methods keep track of the probabilities of all the items they deem salient, in this case, both 0 and 1, but we focus on the probability estimates $\hat{p}^{(t)}$ for item 1.

| deviation threshold $\rightarrow$ | 1.5 | 2 | 1.5 | 2 |
|---|---|---|---|---|
| | Qs, 5 (qcap of 5) | | Qs, 10 (qcap of 10) | |
| $p^* = 0.10$ | $0.385 \pm 0.026$ | $0.129 \pm 0.020$ | $0.191 \pm 0.029$ | $0.026 \pm 0.010$ |
| $p^* = 0.05$ | $0.405 \pm 0.034$ | $0.142 \pm 0.028$ | $0.211 \pm 0.044$ | $0.035 \pm 0.016$ |
| | static EMA, 0.01 ($\beta$ of 0.01) | | static EMA, 0.001 ($\beta$ of 0.001) | |
| $p^* = 0.10$ | $0.075 \pm 0.021$ | $0.013 \pm 0.006$ | $0.113 \pm 0.020$ | $0.071 \pm 0.012$ |
| $p^* = 0.05$ | $0.211 \pm 0.034$ | $0.050 \pm 0.019$ | $0.118 \pm 0.029$ | $0.072 \pm 0.016$ |
| | harmonic EMA, 0.001 ($\beta_{min}$ of 0.001) | | DYAL , 0.001 ($\beta_{min}$ of 0.001) | |
| $p^* = 0.10$ | $0.006 \pm 0.007$ | $0.002 \pm 0.003$ | $0.018 \pm 0.015$ | $0.008 \pm 0.006$ |
| $p^* = 0.05$ | $0.012 \pm 0.013$ | $0.005 \pm 0.005$ | $0.028 \pm 0.023$ | $0.014 \pm 0.012$ |

Table 9: Synthetic single-item stationary: Deviation rates, for two deviation thresholds $d \in \{1.5, 2\}$, averaged over 200 randomly generated sequences of 10000 binary events (0 or 1), for target probability $p^* \in \{0.05, 0.1\}$. As an example, for $p^* = 0.1$, about 10% of the items will be 1, the rest are 0s in the sequence, and the SMA predicts a probability $\hat{p}^{(t)}$ at every time point $t$ for $o^{(t)} = 1$ (then updates), and we observe from the table that about 38% of time, $\max(\frac{\hat{p}^{(t)}}{p}, \frac{p}{\hat{p}^{(t)}}) > 1.5$ (*i.e.* $\hat{p}^{(t)} > 0.15$ or $\hat{p}^{(t)} < \frac{0.1}{1.5}$), for the Qs SMA with qcap 5 (top left). In this stationary setting, larger queues work better (as expected), and a lower $\beta_{min}$ of 0.001 performs best for the EMA variants: : harmonic EMA does best, and DYAL is a close second.

```
GenSequence(desiredLen, O_min, L_min)
    // Create & return a sequence of subsequences.
    seq ← [] // A sequence of items.
    prevSD ← {}
    While len(seq)< desiredLen:
        // Extend existing sequence
        P ← GenSD(prevSD) // get new SD.
        seq.extend(GenSubSeq(P), O_min, L_min)
        prevSD ← P
    Return seq

GenSubSeq(P, O_min, L_min)
    // Generate a subsequence, via repeated
    // sampling IID from SD P, long enough
    // so all item of P occurs ≥ O_min times in it.
    counts ← {} // An observation counter map.
    seq ← [] // A sequence of items.
    While min(counts) < O_min or len(seq) < L_min:
        o ← DrawItem(P) // item o drawn.
        seq.append(o)
        // Update counts only for salient items.
        If o ∈ P: // increment o's observed count.
            counts[o] ← counts.get(o, 0) + 1
    Return seq
```

```
GenSD(prevSD) // Generate a new SD.
    // Parameters: recycle, p_min, P_max, p_min.
    probs ← [] // probabilities of the SD.
    // Repeat while unallocated mass is sufficient.
    While u(probs) > p_min + p_min:
        // sample probabilities.
        p_max ← min(left − p_min, P_max)
        p ∼ U([p_min, p_max]) // uniformly at random.
        probs.append(p) // add p to SD.
    If recycle: // reuse items '1', '2', ..
        Random.shuffle(probs) // random permute
        // item '1' gets probs[0], '2' gets probs[1], etc.
        Return MakeMap(probs)
    Else: // Allocate new items (unused ids).
        Return MakeNewSDMap(probs, prevSD)

DrawItem(P) // Return a salient or noise item.
    sump ← 0.0 // via sampling.
    p ∼ U([0, 1.0]) // Uniformly at random.
    For o, prob ∈ P:
        sump ← sump + prob
        If p ≤ sump:
            Return o // Done.
    Return UniqueNoiseItem() // a unique noise id.
```

Figure 18: Pseudocode for generating a sequence of subsequences. Each subsequence corresponds to a stable period, subsequence $j \geq 1$ being the result of drawing IID from the $j$th SD, $\mathcal{P}^{(j)}$.

subsequence $j$, $j \geq 1$, corresponding to one SD $\mathcal{P}^{(j)}$. The subsequence is long enough (drawn IID from $\mathcal{P}^{(j)}$) so that $\min_{i \in \mathcal{P}^{(j)}} count(i) \geq O_{min}$, where $count(i)$ is the number of occurrences of item $i$ in the subsequence. We may have a minimum (overall) length requirement as well, $L_{min} \geq 0$. Each $\mathcal{P}^{(j)}$ is created using the **GenSD** function.[32]

---

[32]There are a variety of options for generating SDs and sequences. In particular, we also experimented with the option to change only one or a few items, once they become eligible (their observation count reaching $O_{min}$), and we obtained similar results. Because such a variation is more complex and requires specifying the details of how an item's change affects others (e.g. how it is replaced by zero or more new or old items), for simplicity, we use GenSD(), *i.e.* changing all the items, and only when all become eligible. Note that, under the reuse=1 setting, some items' probabilities may not change much, simulating a no-change for some items. Experiments on real streams provide further scenarios (Sect. 6).

| DYAL, 0.01 vs. Box, 100 | $O_{min}$=10, recycle | 10, new | $O_{min}$=50, recycle | 50, new |
|---|---|---|---|---|
| losses, wins of DYAL | 7, 43 | 0, 50 | 8, 42 | 0, 50 |
| log-loss (opt: 1.064 and 0.988) | 1.120,1.108 | 1.182,1.113 | 1.013,1.008 | 1.025,1.010 |

Table 10: Losses and wins of DYAL, with $\beta_{min} = 0.01$, pairing it against Box (Sect. B.2) with window size $K = 100$ (which performed better than $K$ of 50 or 200), on 50 sequences of 10k each (the setting of Table 3). Thus DYAL has the lower loss on 43 of 50 in the recycle mode with $O_{min} = 10$. Average losses are also shown (1st number is Box's), and average optimal LogLossNS is 1.064 ($O_{min}$=10) and 0.988 ($O_{min}$=50). All wins are highly significant (using the binomial sign test). With the lower $\beta_{min} = 0.001$ for DYAL (not shown), DYAL has statistically more wins under the more challenging (for Box) new-items settings but has mixed performance under the recycle settings.

In GenSD, some probability, $p_{min}$, is reserved for the noise items. Probabilities are drawn uniformly from what probability mass is available, initially $1 - p_{min}$, with the constraint that each drawn probability $p$ should be sufficiently large, $p \geq p_{min}$. Optionally, we may impose a maximum probability $P_{max}$ constraint as well ($P_{max} \leq 1$). Assume we get the set $S = \{p_1, p_2, \cdots, p_k\}$, once this loop in GenSD is finished, then we will have $\sum_{p_i \in S} p_i \leq 1 - p_{min}$, and each probability $p_k$ satisfies the minimum and maximum constraints.

Once the set $S$ of $k$ probabilities is generated, GenSD() then makes a SD from $S$. Under the **new-items** setting (when $recycle = 0$), new item ids, $|S|$ such, are generated, *e.g.* an item id count is incremented and assigned, and the items are assigned the probabilities (and all the old items, salient in $\mathcal{P}^{(j-1)}$, get zeroes, so discarded). Thus, as an example, the first few SDs could be $\mathcal{P}^{(1)} = \{1{:}0.37, 2{:}0.55, 3{:}0.065\}$, $\mathcal{P}^{(2)} = \{4{:}0.75, 5{:}0.21, 6{:}0.01, 7{:}0.017\}$, and $\mathcal{P}^{(3)} = \{8{:}0.8, 9{:}0.037, 10{:}0.15\}$. Under the **recycle** setting, with $k$ probabilities generated, items 1 through $k$ are assigned from a random permutation[33] $\pi()$ of $S$: $\mathcal{P}(i) \leftarrow p_{\pi(i)}$ (thus item 1 may get $p_3$, etc.). For example, with the previous probabilities, we could have $\mathcal{P}^{(1)} = \{1{:}0.065, 2{:}0.37, 3{:}0.55\}$ (and 0.015 is left for noise items), $\mathcal{P}^{(2)} = \{1{:}0.75, 2{:}0.01, 3{:}0.21, 4{:}0.017\}$ (a new item is added), $\mathcal{P}^{(3)} = \{1{:}0.037, 2{:}0.8, 3{:}0.15\}$. Thus, under the *recycle* setting, in addition to changes in probability, the support of the underlying SD $\mathcal{P}$ may expand (one or more new items added), or shrink, with every change (*i.e.* from one stable subsequence to the next one).

### D.3 Pairings on Synthetic Experiments and Additional Sensitivity Plots

We pair DYAL with another SMA and count the number of wins and losses, based on log-loss over each of the same 50 sequences of Table 3 (multi-item), and perform statistical sign tests. With $O_{min}$=50 and the new-items setting, pairing DYAL with $\beta_{min} = 0.01$ (best or near best of DYAL) against any of the other techniques, *i.e.* static or harmonic sparse EMA (with $\beta$ in $\{0.001, 0.01, 0.02, 0.05, 0.1\}$) or Qs, DYAL gets a lower log-loss on *all* 50 sequences. If we lower the $\beta_{min}$ rate for DYAL to 0.001, we still get dominating performance by DYAL, but harmonic can win on a few one or two sequences. With $O_{min}$=10 and again the new-items setting, pairing DYAL with $\beta_{min} = 0.01$ against all others, again we obtain the same dominating results for DYAL.

With the item-recycle setting, the differences between the best of the EMA variants and DYAL variants shrinks somewhat. For instance, with $O_{min}$=50, harmonic EMA with $\beta_{min} = 0.01$ gets 9 wins (DYAL , $\beta_{min} = 0.01$, gets the remaining 41 wins), and if we use $\beta_{min} = 0.001$ for DYAL , DYAL loses 42 times to harmonic EMA with $\beta_{min} = 0.01$. Similarly, with $O_{min}$=10 (high non-stationarity), in the recycle setting, we need to set $\beta_{min} = 0.02$ to get a dominating performance by DYAL, and setting it lower to $\beta_{min} = 0.01$ (which worked well for $O_{min}$=50) yields mixed performance. Thus, the choice of rate $\beta_{min}$ for DYAL can make a difference, although the operating range or the sensitivity to $\beta_{min}$ is substantially lower for DYAL than for other EMA variants (Fig. 11), in particular when setting $\beta_{min}$ to a low value. We observe this improved sensitivity on other data sources and settings as well.

---

[33]The probability generation process tends to generate smaller probabilities with each iteration of the loop, this shuffling ensures that the same items are not assigned consistently high or low probabilities.

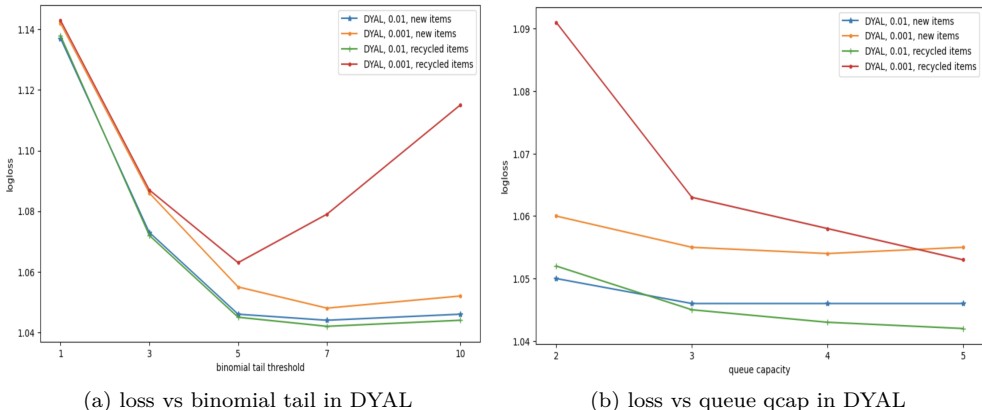

(a) loss vs binomial tail in DYAL

(b) loss vs queue qcap in DYAL

Figure 19: DYAL performance sensitivity in synthetic multi-item setting: (a) to the binomial tail threshold (b) to qcap (on the same 50 sequences of Table 3, under both new-item and recycle generation settings, and with $O_{min} = 50$).

The Box SMA (Sect. B.2) performed somewhat better than static EMA, with an appropriate window size $K$, which is about 100 for the setting of Table 3 ($K$ =50 and 500 were strictly inferior, and 200 is slightly worse, in terms of LogLossNS and other measures). Like EMA, Box is inflexible and requires tuning the fixed $K$ for different input streams, and unlike EMA its rigid space consumption is prohibitive. Table 10 summarizes paired comparisons with DYAL, under the two recycle and new-item settings and $O_{min} = 10$ and 50. DYAL with $\beta_{min} = 0.01$, consistently outperforms it.

Fig. 19 shows two plots of DYAL sensitivity to the binomial threshold and the queue capacity, showing DYAL performance is fairly stable under a reasonable range.

## E  Additional Experiments on Real-World Sequences

Of the 104 Expedition sequences, there are 19 sequences with length above 5k, median length being 13k. We averaged the log-loss performances of DYAL, as we change its $\beta_{min}$, over these 19, as well as over the sequences with length below 1k, of which there are 46 such, with median of 280 observations. Fig. 20(a) shows that lowering the $\beta_{min}$ works better or no worse, for longer sequences, as expected, while the best performance for the shorter sequences occurs with higher $\beta_{min} > 0.01$ (increased non-stationarity). A similar patterns is also seen in Fig. 20(a) for Qs technique. Larger queue capacities work better for longer sequences, but because shorter sequences dominate the 104 sequences, overall we get the result that a qcap of 3 works best for this data overall. We also note that the shorter sequences yield a lower log-loss (both figures of 20). This is expected and is due to our protocol for handling noise: for methods that allocate most their initial mass to noise (unallocated) and when this agrees with the noise marker referee, one gets low loss. For instance, at $t = 1$, the noise marker marks the next item as noise, and with a predictor that has all its probability mass unallocated, log-loss is 0 ( loglossRuleNS() in Fig. 2(b)). As the sequence grows longer, and more salient items are discovered, the average loss can (initially) go up (even though learning is taking place).[34]

### E.1  More on Nonstationarity

A way to see evidence of non-stationarity is to look at the maximum of the learning rates, in the *rateMap* of DYAL, as a function of time (similar to Fig. 15). Whenever this max-rate goes up, it is evidence that an item

---

[34]In learning under non-stationarity, it is expected that performance curve is not monotonic and can degrade at times even though learning is taking place all the time. In the above experiments, one can imagine all SMAs beginning with all mass allocated to a dummy item initially, forcing them to learn the appropriate probability allocation to the noise portion as well as the salient items, and in that case, we'd observe an initial worst-possible loss and the gradual lowering of the loss.

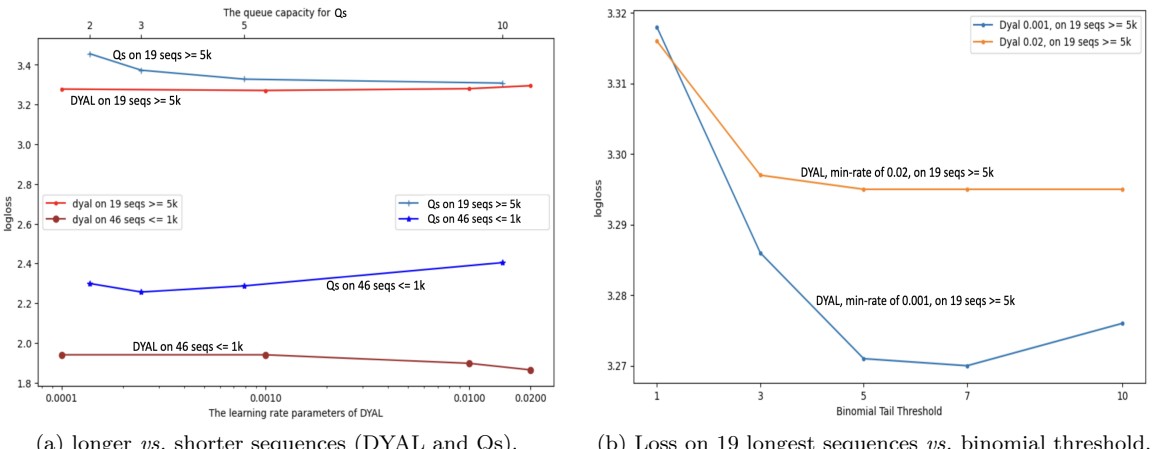

(a) longer *vs.* shorter sequences (DYAL and Qs).     (b) Loss on 19 longest sequences *vs.* binomial threshold.

Figure 20: log-loss (a) on 19 longest sequences (above 5k) vs 46 shorter sequences (below 1000). On the longer sequences lower $\beta_{min}$ for DYAL, and a higher qcap for Qs improves performance. (b) loss vs binomial threshold on 19 longest sequences. Setting the value in 3 to 7 works well, and $\beta_{min} = 0.001$ does better than $\beta_{min} = 0.02$ on these sequences.

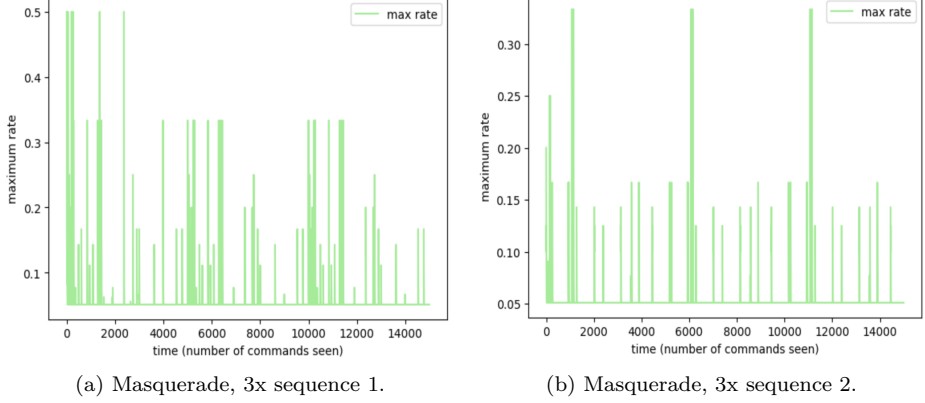

(a) Masquerade, 3x sequence 1.     (b) Masquerade, 3x sequence 2.

Figure 21: The maximum learning rate of DYAL as a function of time, on two sequences from Masquerade, where each sequence is concatenated with itself 3 times. The rate repeatedly jumps up, exhibiting pulsing or spiking patterns, indicating change.

is changing probability, for instance a new item is replacing an old. We took a sequence and concatenated with itself a few times, 3x in case of Masquerade sequences (Masquerade sequences are uniformly 5k long each). Fig. 21 shows two examples (thus [A, C], *i.e.* A followed by C, becomes [A,C,A,C,A,C]). If the start of a sequence has a distribution substantially different from the ending then, when concatenated, we should repeatedly see max-rate jump up.[35] We looked at many such plots. The max-rate repeatedly goes up in plots of all sequences from the 52-scientists (whether with default parameters or with $\beta_{min}$ of 0.05), and all except 1 of the 50 Masquerade sequences, and similar pattern on all except the shortest handful of sequences from 104 Expedition sequences. In other experiments (not shown), when updating (learning) is turned off midway along a sequence (whether on Unix or Expedition), the loss significantly increases (compared to continuing the updates).

---

[35]Non-stationarity at the level of predicting the next item given current priors. There may be higher-level *stationary* regularities, *e.g.* periodicities, that could explain the lower-level or short-term changes.

