# OpenReview forum: "Lifelong Open-Ended Probability Predictors"
_TMLR — Accepted by TMLR_

### Review · Reviewer_XK7T · 2025-12-12

**Summary Of Contributions:**

The paper studies online probabilistic multiclass prediction under lifelong, open-ended, and non-stationary streams with strict memory constraints. It introduces sparse moving averages (SMAs): (i) sparse EMA with fixed or harmonically decaying learning rates, (ii) a queue-based counting method (Qs), and (iii) DYAL, a hybrid that maintains per-item learning rates and small queues with binomial KL-based change tests to reset estimates. For evaluation with emerging items, the paper proposes a bounded variant of log-loss based on filtering/capping predictions and a simple noise referee; it provides a convergence bound for sparse EMA and empirical evidence that per-item dynamic rates (DYAL) often yield superior stability–plasticity trade-offs.

**Additional Comments:**

- How sensitive are results to the referee parameters (c_NS, windowing) and to p_min/p_NS choices? Could you report calibration metrics and log-loss across a grid of these settings to show robustness?
- In real-world datasets, what are the item cardinalities, memory budgets (as a function of p_min), and throughput? Can you include timing and memory profiles, as well as ablations on DYAL’s significance threshold and beta_min?

**Audience:**

Yes

**Audience Explanation:**

- The problem setting is well-motivated: open-ended, space-bounded, probabilistic prediction where the support evolves and can grow unbounded.
- The setting is practically relevant (UI agents, personalization, large categorical spaces) and under-explored relative to more common frequency/heavy-hitter tasks.
- The proposed evaluation methodology could be useful for broader work on probabilistic prediction with sparse supports, where unseen items are common.

**Claims And Evidence:**

Yes

**Claims Explanation:**

- The DYAL hybrid with per-item harmonic decay and queue-triggered resets via a binomial KL tail test is a pragmatic and interesting design that connects rate-based EMA with counting-based change detection.
-  Synthetic experiments explore regimes with large and frequent probability shifts versus heterogeneous “drifts,” and report deviation rates and log-loss under controlled stability windows.

**Requested Changes:**

- Theoretical results are limited to convergence of sparse EMA in stationary periods; DYAL’s central mechanisms (per-item rate, reset criteria) lack formal guarantees (e.g., bounds on detection delay, false alarms, regret under drift).
- Real-world evaluations are only briefly mentioned; details, datasets, and comparative results versus strong baselines are not fully evident in the excerpt.
- Some typographical errors and occasional digressions (e.g., philosophical epigraphs) distract from the core narrative; the semi-distribution vs distribution terminology could be streamlined.

---

> ### Author Response · Authors · 2025-12-23
> **response 1 to reviewer  XK7T**
>
> Thank you for your time and valuable feedback, which we hope to
> adequately address (in paper and responses).  We agree with the points
> (summaries, etc) made, and will only include cases where reviewer
> asked or requested or where added clarification could be useful.
>
> - A minor clarification on 'where the support evolves and can grow
> unbounded' (which could be related to the later cardinality comment):
> the support (or the item set deemed salient) indeed evolves but never
> grows beyond certain limit $O(1/p_{min})$ (however, it is true indeed
> that the set of items that were part of the support at some point in
> the past up to current time can grow unbounded).
>
> On the requested points:
>
> - 'Theoretical results are limited to convergence of sparse EMA in
> stationary periods..' (perhaps similar concern raised by Zdsi): we
> tried to connect results and enhancements on sparse EMA ) to DYAL
> (such as harmonic decay of rate) (referring back and forth in their
> sections), but perhaps certain technical properties need to be
> explicitly stated (please see footnote 11 for an informal
> statement ). For instance, for a new item (which had 0 probability, now
> probability say $p=0.1$ for some period of time), it is eventually
> (but reasonably soon) detected, with very high probability (the queues
> estimate in DIAL will be sufficiently high and close to true
> probability, and then EMA part of DYAL will pick it up, where harmonic
> decay has convergence within $O(1/p)$ ). The higher (the more distant)
> the $p$ is from $p_{min}$, and with increasing number of all
> observations (a multiple of 1/p), or positive observations of the
> item, the less likely that the item is pruned or does not make it to
> the EMA part (ie does not get the required number of queue cells) (see
> also our response to reviewer 79Vu's question on pruning).
>
> All such claims are probabilistic of course (including convergence of
> sparse EMA), and concisely formulating the properties and in a way
> that contributes to understanding, could be challenging (*eg* our near-propriety
> development): We wanted to have a significant empirical component
> to the paper too (and left certain aspects to informal 'probabilistic
> intuition').
>
> There are some theoretical properties (or certain tradeoffs) that a better understanding would help (we
> could devote a paragraph to open problems of interest), such as: (1)
> the granularity of change detection in DYAL (*eg* if EMA part of DYAL has
> an item already at ~0.5, but true probability is 0.9,
>  DYAL may not switch to queue estimate if queue is small, *eg* 3 cells, and binomial
> threshold is conservative (too high),
> so the EMA estimate needs to
> converge gradually using its own learning rate) (motivates keeping a positive $\beta_{min}$), (2) (undesired interaction
> among predictands) if a new item should in effect  replace another,
> is it possible evidence for evicting the old item is not strong (we
> compute evidences independently, so it is possible that evidences or
> the tests 'disagree' at times) (we expect these are low probability
> events, but analysis and quantification would be useful)
>
> ( The granularity of adaptation to change requires more careful
>   analysis, and improvement in techniques  may not easily reveal
>   themselves in empirical experiments. )
>
>
> - 'Some typographical errors and occasional digressions (e.g.,
> philosophical epigraphs) distract from the core narrative; the
> semi-distribution vs distribution terminology could be streamlined.'
>
> We hope to fix errors. If you can clarify further that would be
> helpful (if important): On philosophical, did you mean the quote
> before introduction? (or certain statements in summary or
> introduction?). On streamlining the semi-distribution vs distribution
> terminology, did you mean section 2.1 ? (or perhaps subsequent 2.1.1)
> (we think they are sufficiently concise).
>
>
> - Real-world evaluations are only briefly mentioned; details,
> datasets, and comparative results versus strong baselines are not
> fully evident in the excerpt.
>
> We tried to keep the paper short and to the point. For instance, on
> Unix data, we also ran experiments on predicting full commands (ie
> larger output spaces) and on all other users with shorter sequences
> (Greenberg data), and we have obtained results on additional text
> documents (Expedition ngram prediction sequences). The findings and
> insights have not changed (please see also response1 to reviewer Zdsi
> on other techniques, such as supervised ml).
>
> Update: please also see response 3 to reviewer Zdsi on experiments using ADWIN (for
> change detection+mean estimation on numeric streams).
>
> The problem of probability, and distribution, prediction in an open-ended categorical
> setting is new (and we found it tricky, even on settling on robust
> comparison techniques), and we hope to have motivated the area and
> introduced and described a general version sufficiently well.

---

> ### Author Response · Authors · 2025-12-23
> **response 2 to reviewer XK7T (on remaining requests)**
>
> Our responses to other requests & questions:
>
> - How sensitive are results to the referee parameters (c_{NS},
> windowing) and to $p_{min}/p_{NS}$ choices? Could you report calibration
> metrics and log-loss across a grid of these settings to show
> robustness?
>
> We have reported on changes in $c_{NS}$ in a few cases (6.1.1 on Unix sequences,
> Table 7 and section 6.2.2).  The referee window was left unbounded for
> simplicity, but for the case of Unix Masquerade, we reported on finite
> window sizes (which improved the standing of DYAL). Our focus has been
> in head-to-head comparisons, but we also state, in 6.2.2, that as $c_{NS}$ is
> raised (the referee becomes more relaxed, or a longer grace period), the losses (for all
> techniques) improve somewhat, and this is expected/explainable. (on
> 'robustness': the loss values remain close as we change referee
> parameters, if that's what you mean. But the number of wins/losses
> also change gradually with change in $c_{NS}$).
>
> Perhaps in the appendix, we can add a table (or plot) for a dataset or
> more on the combination of $c_{NS}$ and window size?
>
> $p_{min}$ is conceptually distinct from $p_{NS}$, but practically they are
> set equal (we will focus on $p_{min}$).
> $p_{min}$  controls down to what level of probabilities are modeled and is domain and task dependent (depends
> on the non-stationarity levels and how much space can be made available to each predictor).
> We expect 0.01 or 0.001 would be adequate  for many domains (based on experience with the current
> data sets). Setting it too high, eg 0.5 or 0.2 would hurt performance substantially: our
> $\beta_{min}$ figures shed light on the range of probabilities that are useful in different
> datasets we experimented with (relative to using DYAL ).
> (note: it may be possible to tune such parameters on the fly too).
>
> - In real-world datasets, what are the item cardinalities, memory
> budgets (as a function of $p_{min}$), and throughput?
>
> Theoretically memory and update times are $O(1/p_{min})$ (sections
> 3.3.1 and 3.2.2), and in practice, often a predictor's requirement
> grows to such extents with time (as its input stream grows) (however,
> it doesn't have to: imagine a predictor that always sees the
> same (or a few) observation(s), and variants of this do happen, both
> in text and Unix data).  Two typical examples (predictor stats such as size,
> that is number of edges) are given in Fig. 15.
>
> - Can you include timing and memory profiles, as well as ablations on
> DYAL’s significance threshold and $\beta _{min}$?
>
> Loss vs (minimum) learning-rate and binomial-tail thresholds in DYAL
> are reported in Fig 14 and in the appendix (Figs 19 and 20) (*eg*
> Binomial tail of 1 or 2 is too low of a confidence requirement for many cases, and
> results in too many switches to the queue, or false positive changes,
> at least in the Expedition domain. on Unix, 2 or 3 maybe best)
> (several figures show sensitivity to $\beta_{min}$, such as Fig 12).
>
> Experiments were fast (seconds, *eg* on 50 10k sequences, for any SMA).
> However, DYAL is the slowest of SMAs,
> so perhaps 2x or 3x as slow as plain (static/harmonic) sparse EMA (but a fixed bounded
> overhead, as for any SMA, *ie* not growing with stream length after a certain point ).

---

> > ### Comment · Reviewer_XK7T · 2026-01-14
> >
> > I appreciate the detailed responses and strongly support the proposal to include an appendix on parameter sensitivity and a discussion on open theoretical problems to address robustness and theoretical scope. I also accept the clarifications regarding computational complexity and confirm that my terminology comment simply referred to ensuring consistent usage of "semi-distribution" in Section 2.1 to prevent reader confusion. These revisions and the provided context adequately address my concerns.

---

### Review · Reviewer_Zdsi · 2025-12-13

**Summary Of Contributions:**

The paper studies online multiclass probability prediction on lifelong, open-ended streams, where the set of items is not known in advance and may grow without bound or even shrink. The predictor is space-limited and should only track "salient" items above threshold $p_{min}$. Some of the claimed contributions are:

- The proposal of Sparse Moving Average (SMA) techniques. Specifically, they introduce DYAL (Dynamic Adjustment of Learning), a hybrid predictor that combines a Sparse EMA with a Queue-based estimator (Qs). DYAL uses the queue to detect significant changes via a binomial-tail test and resets the EMA learning rates per-item (per-predictand) accordingly
- The formulation of a "bounded log-loss" metric and a protocol (Filter & Cap) to evaluate open-ended probability streams where new or noise items appear frequently, effectively preventing infinite loss on unseen items
- A convergence bound for sparse EMA in stationary settings, proving expected first-visit time to a probability band is $\mathcal{O}(\beta^{-2})$.
- Evaluations on synthetic data and real-world datasets (Unix commands and an internal "Expedition" system), claiming DYAL strikes a superior stability-plasticity tradeoff compared to static EMA or simple queuing.

**Audience:**

Yes

**Audience Explanation:**

Yes, but with a narrow and conditional appeal. This would be an interesting work for researchers working on streaming / online learning under concept drift, especially for discrete or open-vocabulary domains (e.g., event streams, logs, user actions). However, due to the following reasons, the interest would be limited:
- The novelty is modest, and the main algorithm is a heuristic hybrid of known methods.
- The main evidence is protocol-dependent and relies on a referee. Readers not already aligned with this framing may be unconvinced or skeptical.
- Weak engagement with related work, especially in the empirical investigations of the method.

**Broader Impact Concerns:**

The current manuscript focuses entirely on algorithmic performance and mathematical bounds. Given the stated applications in personalization and system monitoring, the lack of a dedicated Broader Impact Statement could be of concern.

**Claims And Evidence:**

No

**Claims Explanation:**

The claims in the paper are partially verified, and there are some gaps in the paper. I will summarize some of these gaps as follows:
- The central evaluation (AvgLogLossNS) depends on a noise-marker referee and hyperparameters ($p_{min}$, $p_{NS}$, $c_NS$), that can change the outcomes. Even the paper itself notes artifacts (e.g., low loss on short sequences due to "noise" handling). This makes it hard to interpret improvements as "better probability prediction" rather than "better alignment with the referee."
- The claim of "superiority" is insular. The paper completely fails to compare DYAL against established State-of-the-Art (SOTA) methods for concept drift and streaming probability estimation, such as ADWIN (Adaptive Windowing), Hoeffding Trees, or even standard Online Logistic Regression.
- The claim of superior performance is evaluated based on some 25 years old dataset of Unix commands, which is not satisfying. Modern "lifelong" learning involves high-dimensional, rapid-fire streams (e.g., clickstream data, IoT sensor logs, server metrics). Proving performance on small-vocabulary command logs is not convincing evidence of scalability or utility for modern open-ended problems. Even the "Expedition" dataset is proprietary, and hence, makes the evaluation harder.
- The convergence proof applies only to Static EMA, a standard technique. It does not cover the paper's main contribution, DYAL, which involves dynamic resetting of learning rates based on a queue trigger.

**Requested Changes:**

- Add at least two strong, conceptually different baselines and re-run core experiments. Some examples are: Adaptive window / drift-aware baseline / Bayesian smoothing baseline for categorical prediction.
- Compare against a simple Online Logistic Regression (e.g., via SGD) or a basic LSTM/GRU predictor.
- Evaluation on Modern Public Datasets: Report results on at least one recognized, modern, high-dimensional streaming dataset, like: Click-through Rate Prediction or IoT Sensor Streams.
- Provide Brier Score or standard Log-Loss (without the referee filter) on the subset of salient items

---

> ### Author Response · Authors · 2025-12-23
> **Response 1 to reviewer Zdsi**
>
> Thank you for your time and valuable feedback, which we hope to
> adequately follow up on. Summary of response:
>
> 1) on claims: we provide evidence (technical results, figures such as
>  6 and 10, and a range of experiments on synthetic and real) on why
>  certain techniques can converge faster. For example, the technical
>  results on EMA  *do apply* to DYAL.
>
> 2) on comparing to existing SOTA: we explain why SOTA techniques
> mentioned are not readily applicable (ADWIN and supervised ml) (we could
> include a ~2 page appendix on the differences, including an experiment
> with softmax under non-stationarity (but under a fixed known output
> set.. not open ended), showing a single learning rate is not adequate (when one item
> is stable, others swapping probabilities).
>
> 3) Evaluation and Brier (another post will include our response on Brier)
>
> 4) Other: choice of data, etc.
>
> ----
>
> 1) Contribution clarifications (in particular: 'convergence proof does
> not cover the paper's main contribution, DYAL').
>
> 1.1) Our contributions are stated in the introduction. All techniques,
> not just DYAL, offer some novelty (eg in some non-stationary
> applications, the probabilities provided by simple Qs might be
> sufficient).
>
> 1.2) The technical results/insights on EMA do apply to DYAL. For
> instance the harmonic lowering of the rate allows DYAL to converge
> faster for a high probability $p$ ($1/p$ instead of $\ge 1/\beta$)
> (Fig 6), and, similarly, if there is a floor on the learning rate (in
> our experiments, there is, eg 0.001), proof of convergence of
> fixed-rate EMA applies (in the stability period).
>
> 2) Reviewer stating insufficient comparisons and reviewer's
> recommendations.
>
> - comparing to  SOTA (such as logistic
>  regression). Supervised learning (sml) (such as Hoeffding trees or
>  logistic regression) assumes two inputs to the problem: the input
>  features (eg stream of vectors) and corresponding observations (a
>  class stream). Ours is one stream.  It can be modeled as a sml (assume
>  a dummy feature), but much of the richness of sml  becomes
>  irrelevant.  On the other hand, typical sml can have restrictions: a
>  major issue not easily lifted  is the assumption of
>  'closed-world' (and stationarity): the items (classes) should be a
>  fixed set and known to the learner (in related work, we mention that
>  predictors learned using SMAs could be useful in  open-ended sml). As explained above, we could add to
>  appendix on sml differences (and include the fixed fan-out softmax experiment mentioned above).
>
> - ADWIN and ADWIN2 are closer but not readily applicable either: the
>  solution themes are similar (change detection, averaging, a kind of
>  windowing .. as discussed in related work). There the problem is
>  stream of scalars/numeric and means and variances are computed in windows (with corresponding tests). In
>  our case, the observations are categorical, or sparse vectors. One could do ADWIN
>  *per item* (similar to our use of queues per item), but then one
>  needs to keep track of subsequences of 0s and 1s *per item* (not
>  sparse, highly space inefficient, and not plain ADWIN any more). Our
>  representation and binomial-tail tests are geared towards the special case
>  of binary sequences (for each item) and efficient proportion
>  estimation (of the 1s).
>
> - Unix data & data sizes: we looked for multiclass data with
>  definite non-stationarity in past work, such as the on same user's behavior (click prediction
>  suggested by reviewer appears binary class).  Sequences starting at
>  100s to 10k long are adequate for us. Eg if we are interested in
>  minimum 0.01, low 1000s is enough for stability period and few such
>  concatenated for verifying change and adaptation (10k to 20k).  The
>  obtained Expedition sequences in the paper are made available (please
>  note: *any text* corpus could be the input, such as a foreign
>  language). On 'lifelong': SMA memory is bounded (eg counts won't overflow),
> and the SMAs remain adaptive throughout the input stream.
>
> - on hard to tell whether 'better alignment with the referee':
>  we report on several referee thresholds (and on a rich range of
>  synthetic settings, where interpretable deviation rates are also
>  reported, and on real data).
>
> - 'paper itself notes artifacts (e.g., low loss on short sequences due
>  to 'noise' handling)': we should note that the problem is
>  non-stationary, and a performance sequence can show regions of
>  improvement, degradation, plateau, etc multiple times, even though
>  learning is continual  (unlike a learning curve for a typical stationary problem,
> which is monotone). Initially, the observed items don't get a
>  probability by a predictor (and the referee agrees), so the bounded
>  loss is near 0 (we can clarify in paper).
>
> - 'evaluation depends on ... hyperparameters ($p_{min}$..':
> Please note that the $p_{min}$ and $p_{NS}$ stem from the
> finite-memory assumption, and could be assumed to be part of domain
> knowledge and/or figured out with a bit of experimentation (or perhaps
> via meta learning).
>
> Thank you.

---

> ### Author Response · Authors · 2025-12-23
> **On using the Brier loss (response 2 to reviewer Zdsi)**
>
> On 'Provide Brier Score or standard Log-Loss (without the referee
>   filter) on the subset of salient items':
>
> Please see below for a few Brier (quadratic) loss comparisons (which we could add to appendix C.2).
> (as stated in the paper, the code supports reporting the
> Brier  score, where one can also compare two SMAs based on Brier on the same sequences).
>
> The paper  (2.2.2 and Appendix C.2), briefly mentions that this loss
> is not sufficiently sensitive to smaller probabilities. Take the
> actual distributions $D$={A:0.5, B:0.05, C:0.45}, and proposed ones
> $D_2$={A:0.5, B:0.0, C:0.5}, and $D_3$={A:0.45, B:0.05, C:0.5}.  $D_2$
> completely ignores B (a one in 20 event), but $D_2$ and $D_3$, according
> to Brier, have the same loss (of 0.55)! (and $D_4$={A:0.44, B:0.05,
> C:0.51} will have higher Brier loss than $D_2$) Logloss is also (rightly)
> sensitive to higher probabilities, but does not suffer this extent of
> insensitivity to the lower ones. (the longer version of our paper expands on this) (if
> all probabilities were in a narrower range, say 0.1 to 0.5, perhaps
> not an issue to use Brier)
>
> A more subtle issue is that even the appropriate use of the Brier
> may require a grace period and basically a referee (like logloss) may be
> necessary as otherwise, it could lead to 'cheating' or impropriety, by
> a prediction technique (via increasing the probabilities of salient
> items). We quickly describe this issue with an example, which also
> relates to the 2nd half of your plausible recommendation: even if
> salient items (for each stability period) are known (eg from post-processing
> a finite stream), we cannot expect the predictor to provide a positive
> probability the first time the salient item is seen (and log-loss
> blows up). There has to be a grace period, ie some positive
> observation-count parameter (and a history-window size) (in our
> experiments, the predictor should start providing a probability on the
> 4th observation by default, but we provided comparisons on other
> plausible low counts too).
>
> Brier does not blow up on the first observation(s) of an item, but
> consider the stream A:0.5 and otherwise absolute noise (50% A, and 50%
> of time an item is seen that's never seen again (here, we are assuming
> the underlying generator is a proper semidistribution).  The predictor
> that predicts A:1.0 (loss of 0 on A) does better than the truthful
> one, A:0.5 (loss of 0.125 on A), under plain Brier (ie without a
> referee, we are not crediting the truthful predictor for preserving
> some mass for the generic noise portion ) (if we assume the true
> generator is a distribution there won't be this issue).
>
> (note: for logloss too, if we skip say the first 2 occurrences of  salient items and only
> evaluate on remaining, then techniques that allocate all mass to salients do best, but
> this is not proper.)
>
> It may be useful to include Brier losses in the paper (eg to avoid any
> unexpected findings such as DYAL significantly underperforming).  A few plain Brier loss values and comparisons are
> given (we could add such results to appendix C.2)    (this is simple Brier, *ie*
> not using a referee, so with the caveats above) :
>
> On synthetic (Table 3) setting and 104 Expedition sequences:
>
> - On 50 synthetic sequences, each 10k long, $O_{min}$=10 ('all_new'
> setting), 'wins' is for DYAL (a few cases shown) (also includes logloss on
> the same sequences to compare with Brier) (shown parameters are the
> fixed learning rate for EMA, minimum learning-rate for dyal, 'cap' is
> queue size for Qs, etc) (on all 50 sequences, DYAL was better):
>
>
> 50 wins (mean1:0.848, mean2:0.534) (on Brier loss), ema 0.001 vs dyal 0.001
>
> 50 wins (mean1:2.362, mean2:1.134) (on loglossNS), ema 0.001 vs dyal 0.001
>
> 50 wins (mean1:0.589, mean2:0.534) (on Brier loss), ema 0.01, dyal 0.001
>
> 50 wins (mean1:1.268, mean2:1.134) (on loglossNS), ema 0.01, dyal 0.001
>
> 50 wins (mean1:0.547, mean2:0.534) (on Brier loss),  Qs cap=5, dyal 0.001
>
> 50 wins (mean1:0.738, mean2:0.534) (on Brier loss),  Box cap=500, dyal 0.001
>
> 50 wins (mean1:0.567, mean2:0.534) (on Brier loss),  Box cap=100, dyal 0.001
>
>
> - On 104 Expedition sequences, wins is for DYAL (again, also including
> logloss to compare) (the 2nd mean is DYAL's) (all wins are significant using sign test):
>
> 96 wins (mean1: 0.946, mean2: 0.935) (on Brier loss), ema 0.001 vs dyal 0.001
>
> 96 wins (mean1:2.796, mean2:2.392) (on loglossNS), ema 0.001 vs dyal 0.001
>
>
>
> 85 wins (mean1: 0.979, mean2: 0.938) (on Brier loss), Qs cap=5, dyal 0.001
>
> 103 wins (mean1:2.613, mean2:2.392) (on loglossNS), Qs cap=5, dyal 0.001
>
>
> 99 wins (mean1: 0.946, mean2: 0.935) (on Brier loss), box cap=500, dyal 0.001
>
> 99 wins (mean1:2.694, mean2:2.392) (on loglossNS),  box 500, dyal 0.001
>
>
> 76  wins (mean1:0.939, mean2:0.935) (on Brier loss), box cap=100, dyal 0.001
>
> 103 wins (mean1:2.666, mean2:2.392) (on loglossNS), box cap=100, dyal 0.001

---

> ### Author Response · Authors · 2025-12-27
> **response 3 (reviewer Zdsi and other reviewers), on comparing to ADWIN**
>
> (The discussion and experiments below can be added to a 1 page appendix.)
>
> We note that for open-ended probability prediction, the predictor has
> to allocate and initialize parameters whenever a new item is observed, and possibly discard
> certain old items (to keep memory in check).  None of this is
> supported by the current techniques to the best of our knowledge.
>
> We mentioned previously that perhaps ADWIN (for numeric or scalar sequences) could be
> used instead of queues for each single item within some technique,  akin to say DYAL.  We wanted to assess ADWINs
> accuracy in emitting probabilities: on binary sequences, the estimated mean $\hat{\mu}$ from the  selected window
>  is the probability of 1, and we used $1-\hat{\mu}$ for the probability of 0.  ADWIN uses
> window means and variances (and tests using the normal approximation, geared towards real numbers, *ie* not just limited to binary)
> to test which window to pick (in
> DYAL, we use the binomial tails via use of queues, and update via harmonic EMA, while Qs is
> simple: does no testing)
>
> We briefly discuss findings on loss comparisons, then computational complexity (space/time).
>
> ------------------
> - loss comparisons:
>
> We looked at the loss performance on binary sequences, where $p$ (true
> probability of 1) is picked at random in [0, 1] for periods governed
> by $O_{min}$ (as in Table 2) ($O_{min}$ occurrences of 1 before changing $p$). We also looked at oscillation $p_1 \leftrightarrow p_2$
> (*eg* $0.025 \leftrightarrow 0.25$ ) with similar results.
>
>
> We used ADWIN2 from the River online suite of algorithms,
> https://riverml.xyz/latest/api/drift/ADWIN/ (ADWIN2 is more space &
> time efficient than ADWIN, and both are described in Bifet and
> Gavalda, 2007)
>
> We wrote a wrapper to compare.  Here, we used a max loss of 5 for
> logarithmic loss (but 10 and a few higher values do not change
> results). Otherwise we did not use a referee (no noise items).  We
> normalized outputs of Qs and DYAL to make sure the two probabilities for 0 and 1
> add to exactly $1.0$  (as the case for ADWIN).
>
> We also looked at Brier (equivalent to root-mean-square error on binary), with similar results (*eg* last row in table below).
>
> We experimented with all ADWIN parameters (grace_period, $M$, clock,...), but confidence parameter $\delta$ made the most difference (otherwise, needed to use relatively low grace_period and clock such as 5, etc). DYAL ($\beta_{min}=0.001$) and Qs (capacity 5) have default parameters below (wins are head-to-head vs ADWIN, eg DYAL vs ADWIN).
>
>
> | logloss on 30 10k sequences |  $\delta=0.01$ | $\delta=0.1$  | $\delta=0.5$  | DYAL |Qs  | wins|
> | ---                          | ---                    | ---                  | ---                  | ---      | --- | ---    |
> | $O_{min}=50$  | 0.318  | 0.311|0.305|0.300| 0.297|DYAL and Qs win nearly all 30 over ADWIN|
> |  $O_{min}=10$ | 0.330  | 0.322 | 0.314 |0.306   | 0.282|DYAL and Qs win all ~30|
> | $O_{min}=5000$ (basically stationary)  | 0.542  |  0.541 | 0.542 |0.544|0.576|ADWIN, $\delta=0.01$, wins nearly all 30|
> |      Brier loss (RMSE)   |  $\delta=0.01$ | $\delta=0.1$  | $\delta=0.5$  | DYAL | Qs  |    |
> | $O_{min}=50$  | 0.306  | 0.301|0.297|0.297| 0.294|DYAL and Qs win all ~30|
>
>
> Very high $\delta$ (above 0.1)  loses its meaning for ADWIN (or any technique): we need $\delta$ to be below 0.1,
> eg 0.01  for reduced false positives (0.002 is default in River). But with low $\delta < 0.1$ ADWIN is slower to emit good changed probabilities than Qs, DYAL, etc.  With large *eg* $\delta \ge 0.1$,  ADWIN basically reduces to something like Qs (reverting to a small subwindow) and we note Qs does well in these binary experiments.  In multi-item synthetic experiments (not 'closed-world') Qs begins to trail DYAL substantially.    Because the main ADWIN window can grow unbounded (see below), ADWIN has an advantage on stationary.
>
> --------------
>
> - Computational Complexity: the main window in ADWIN, of size W, grows if there
> is no (detected) change in the stream, and the memory requirement is
> $O(M\log(\frac{W}{M})$, where $M >= 2$ is the maximum number of buckets (cells)
> with size at each power of 2 up to $W$ kept (if $M=2$ there are up to 2 windows of size $2^i$, $i=0,1,...$
> until W) (note: each (sub)window is compact keeping summary stats).  ADWIN needs to be modified
> (eg. largest main window cut at a max size of a few ~1000s if we are interested in
> $P_{min}=0.01$) so that its memory (and process time) doesn't grow unbounded. We expect this
> could be done (River doesn't allow us to limit the total number or size of windows).  We saw a noticeable slow down when
> testing on 100k sequences.   However, even with such capping
> main window to say the last 1000 elements observed, the memory
> requirement tends to be higher than our techniques (over 20 of buckets or cells   vs. say $\le 7$ cells for DYAL or Qs), and
> the test performance (to detect change and to pick a subwindow) are more complex in ADWIN (handling non-binary too).

---

### Review · Reviewer_79Vu · 2025-12-19

**Summary Of Contributions:**

This paper addresses the problem of online multiclass probabilistic prediction in a "lifelong" and "open-ended" setting. This setting is characterized by an unbounded stream of items where the set of possible items is not known in advance (open-ended), new items can appear at any time, and the underlying probability distributions are non-stationary (changing over time). The authors focus on resource-constrained predictors that must maintain space and time efficiency while adapting to these changes.

## The key contributions include:

1. **Problem Formalization:** The paper defines the "lifelong open-ended probability prediction" problem  , emphasizing the challenges of handling unbounded item sets, non-stationarity, and the "stability-plasticity" dilemma (balancing fast adaptation with stable estimation) .

2. **New Evaluation Protocol:** To handle the evaluation of open-ended probability predictors where the true set of items is unknown and potentially infinite, the authors propose a bounded version of logarithmic loss (log-loss) . This involves a protocol using "filtering and capping" (FC) to handle small probabilities and a "noise-marker" (referee) to handle new or infrequent items, ensuring the loss remains finite and informative .

3. **Algorithmic Contributions (SMAs):** The authors develop and analyze "Sparse Moving Average" (SMA) techniques :

    - **Sparse EMA:** An adaptation of Exponentiated Moving Average for sparse vectors, including a "harmonic decay" variant for learning rates .

    - **Qs (Queues):** A count-based method using small per-item queues to estimate probabilities .

    - **DYAL (Dynamic Adjustment of Learning):** A novel hybrid method that combines Sparse EMA with Qs . DYAL maintains per-item  learning rates that dynamically adjust. It uses the Qs estimator as a "gate" to detect changes (via a binomial-tail test based on KL divergence) and reset the EMA weights and learning rates when significant deviations occur.

4. **Empirical Evaluation:** The authors conduct extensive experiments on both synthetic data (with controlled non-stationarity)  and real-world datasets (Unix command sequences and "Expedition" system logs). They compare the proposed methods against baselines (static EMA, Qs) and analyze sensitivity to various parameters.

## Limitations
The primary limitations involve the assumption of intermittent stability, as the "listen-to-queue" logic requires sufficient stationary periods to accumulate evidence for resets . Furthermore, while DYAL exhibits reduced sensitivity to learning rate selection, its performance remains dependent on the calibration of the binomial-tail threshold and the minimum rate $\beta_{min}$. Developing more automated or adaptive mechanisms for these hyperparameters would significantly enhance the algorithm's robustness in highly volatile drift environments.

**Additional Comments:**

- There's a typo in page 1 introduction section instead of "We foucs" it should become "We focus"

- Color Blindness Accessibility: Figures 6, 10, and 12 rely heavily on distinguishing "blue," "green," and "orange" lines to compare EMA variants and true probabilities. Adding different line styles (dashed, dotted, or markers) would make these critical performance plots more accessible.

**Audience:**

Yes

**Audience Explanation:**

The problem of "open-ended" learning is increasingly relevant as machine learning moves towards continual, lifelong, and streaming settings where fixed vocabularies cannot be assumed (e.g., in large language models adapting to new tokens, user behavior modeling, or IoT data streams).
The specific combination of Sparse EMA with a queue-based change detector (DYAL) and the dynamic per-item learning rates is a novel contribution to the toolkit of online learning algorithms.

**Broader Impact Concerns:**

The authors do not explicitly list a Broader Impact statement, but the work is primarily algorithmic and theoretical.

- Positive: Efficient algorithms for open-ended prediction can improve the energy efficiency of learning systems  by allowing resource-constrained devices to adapt without expensive retraining.

-  Negative:  As with any user modeling technology (e.g., predicting Unix commands), there are potential privacy implications if improved prediction enables more  profiling of user behavior. However, this paper focuses on the fundamental probability estimation mechanics rather than specific surveillance applications.

**Claims And Evidence:**

Yes

**Claims Explanation:**

While the paper provides a good mix of theoretical motivation and empirical results, the evidence supports the claims with some caveats:

- **DYAL's Performance:** The claim that DYAL strikes a "superior stability-plasticity tradeoff"  is well-supported by the synthetic experiments where the nature of the change is controlled (e.g., oscillating probabilities) . In these settings, the mechanism of DYAL (resetting rates upon change detection) is shown to react faster than static EMA while remaining more stable than pure Qs .

- **Real-World Applicability:** The evidence on real-world data is mixed but honest. On the "Expedition" sequences (internal non-stationarity), DYAL outperforms others significantly. However, on the Unix traces (external non-stationarity), specifically the Masquerade dataset, static EMA with a high learning rate sometimes outperforms DYAL . The authors transparently discuss this, attributing it to the potential lack of sufficient "stable" periods for DYAL's mechanism to be advantageous over a consistently aggressive learner . This nuance adds credibility but slightly weakens the claim of universal superiority.

- **Evaluation Protocol:** The reliance on a "noise marker" (referee) introduces hyperparameters (like $c_{NS}$) that significantly influence the absolute loss values . While the authors show sensitivity analysis, the comparison of methods ultimately depends on this specific, somewhat complex, evaluation harness. The evidence is convincing that _within this harness_ DYAL performs well, but the external validity of the harness itself is a complex argument relying on the proofs in the appendix.

**Requested Changes:**

- **Clarify the "Noise Marker" Impact:** The evaluation relies heavily on the "noise marker" to bound the loss for new items. While Section 2.2.2 describes it, the impact of the specific choice (default $c_{NS}=2$) on the _relative_ ranking of algorithms should be made more explicit in the main text summary. The paper mentions that varying $c_{NS}$ changes the win counts , but a clearer "executive summary" of _when_ (under what noise definition) DYAL fails or succeeds would be helpful.

- **Threshold Sensitivity Analysis**: Provide a more explicit trade-off analysis between "False Alarms" (false changes) and "Missed Detections" (misses) for the binomial-tail threshold, currently set at a default of 5. This is vital for justifying the 99% confidence level which aims to avoid resetting while the EMA estimate should be kept across diverse drift intensities.

- **Pruning Protections**: Clarify if the "heart-beat" pruning logic—which removes items in order of least frequency when the map reaches a maximum capacity—contains mechanisms to protect lower-frequency salient items from being discarded during transient shifts.

- **Baselines Comparison:** The paper compares DYAL primarily against its own components (EMA, Qs) and a "Box" (sliding window) predictor . It would be beneficial to briefly explicitly state why more complex concept-drift baselines (e.g., ADWIN mentioned in related work  ) were not suitable or included as direct competitors .

---

> ### Author Response · Authors · 2025-12-23
> **response 1 to reviewer 79Vu (on broader impact concerns)**
>
> Thank you for your time and valuable feedback, which we hope to
> adequately address (in paper and in responses).  We very much agree
> with all the points/characterization made (in the summaries, contributions such as another
> tool in the toolkit, possible applications such as 'new tokens' on the
> fly, etc).
>
> In our next post we will engage with your requested changes (all valuable).
>
> This response is on 'Broader Impact Concerns':
>
> As reviewer XK7T and you indicated, this is a technical work, and we thought it
> not necessary at current point. However, both  reviewer Zdsi and you
> think it is important/useful to also make a statement. Below are further
> points (in additions to the valuable ones you already made):
>
> On privacy concerns: these type of approaches could lead to
> personalization agents that are better dedicated to users (they can learn
> on their own and without necessarily sharing their data with a central place/company), which could
> help allay privacy issues.
>
> Interpretability: here, a predictor's predictions are categorical and
> interpretable (consider, say in the Unix domain, the time of day or
> 'the morning' as a predictor, or the last command/action-performed as
> predictor: both could predict what the user will do next, and their predictions
> are interpretable) (understandable both in a snapshot in time, and as well as how a predictor's prediction change
> over time).  This has pros and cons: it could be useful for reliability & trouble-shooting,
> explainability and trust, and so on. But it also could be misused (ie open box, and
> a privacy concern).

---

> ### Author Response · Authors · 2025-12-24
> **response 2 to reviewer 79Vu (on requests)**
>
> On the requests:
>
> - 'Clarify the "Noise Marker" Impact: ...  more explicit in the main text summary. The paper mentions that
> varying changes the win counts , but a clearer "executive summary" of when (under what noise definition) DYAL fails or succeeds would be helpful.'
>
> We can add a summary: as the threshold is increased, *ie* longer grace
> periods, the problem becomes more relaxed, the losses generally improve, and
> comparisons get more focused on the more stable items. So the win
> numbers can change (eg a technique that is slower to adapt, such as
> sparse EMA with a low fixed rate, will generally benefit). For several
> plausible referee count choices (2, 3, 4,.. which correspond to short/plausible
> grace periods), DYAL often does best (and the synthetic experiments
> and figures provide insights as to why DYAL could do better). We will also include Brier
> (quadratic) losses in the appendix (please see responses to Zdsi and XK7T).
>
>
> - 'Threshold Sensitivity Analysis: Provide a more explicit trade-off
> analysis between "False Alarms" (false changes) and "Missed
> Detections" (misses) for the binomial-tail threshold, currently set at
> a default of 5. This is vital for justifying the 99% confidence level
> which aims to avoid resetting while the EMA estimate should be kept
> across diverse drift intensities.'
>
> Figure 14(b), and appendix, figs. 19 and 20 show performance (we can
> add  forward pointers) (thresholds 3 to 7 work well, dependent on how
> non-stationary the stream is). But perhaps you meant to expand the
> discussion (eg in an application with many, 1000s+ of predictors, we
> want the false alarm, jitter,  to be low, eg 0.01 or 0.001, but this will be at
> the cost of slower reaction time, ie a bit longer grace periods)
> (see also 'granularity' response to reviewer XK7T). Or perhaps
> include a technical lemma?
>
>
> - 'Pruning Protections: Clarify if the "heart-beat" pruning
> logic—which removes items in order of least frequency when the map
> reaches a maximum capacity—contains mechanisms to protect
> lower-frequency salient items from being discarded during transient
> shifts.'
>
> In footnote 11, we state that it is a probabilistic property, that the
> higher the item's probability the less likely it is pruned (and akin
> to the distortion analysis for near propriety: salient items far from
> boundary won't be zeroed in a new distorted optimal distribution,
> where distortion is caused by filter-and-cap).  When we use multiples
> of $1/p_{min}$ (always keep a few 100s when $p_{min}=0.01$), the
> chance of pruning a salient items becomes negligible).  We can
> formalize or add a sketch of the argument: we use queue cell 0
> counts for sorting and pruning, and it can be shown the kth lowest
> item has its count in cell 0 above k  (not seen for at least k-1 times). The binomial
> tail (same one for switching to queue) can be used here
> in this analysis to show the item, with high likelihood, cannot have
> a high probability.
>
> On 'transient shifts' (if we understood correctly, as spurts or
> undesired runs that may be too long): indeed they can occur in
> real-world non-iid settings (at an extreme, any protection can be
> foiled by an adversary). Techniques such as multi-level models and
> updates (eg per second and then daily) can offer some
> protection. (note also that even if an item is pruned, it is added the
> next time it is observed, and one can argue it wasn't salient before,
> *ie* over the period it was dropped..)
>
> - Baselines Comparison: The paper compares DYAL primarily against its
> own components (EMA, Qs) and a "Box" (sliding window) predictor . It
> would be beneficial to briefly explicitly state why more complex
> concept-drift baselines (e.g., ADWIN mentioned in related work ) were
> not suitable or included as direct competitors .
>
> We will make it explicit: direct comparison with small modifications
> is not possible to our knowledge.  ADWIN and ADWIN2 handle numeric
> sequences (computing means and variances), while we deal with
> categorical (or sparse vectors). To the best of our knowledge, we
> cannot use ADWIN out of the box. It may be possible to use ADWINs per
> item (for which we use queues and learning rates), but we claim for
> binary (Bernoulli) sequences, in per item streams, queues and binomial
> tail tests are more suited and efficient (than more general variance-based
> tests for scalar random variables, which would require some portions
> of the binary streams, per item, to be kept for testing).  (on
> other techniques, please also see discussion with reviewer
> Zdsi on supervised ml (often assuming close-world, ie a finite
> given class set, and adapting to unknown open-ended classes is not
> straight forward)
>
> Update: please see response 3 to reviewer Zdsi on experiments using ADWIN.
>
> - Suggestion to use different symbols in plots (improving accessibility):
>
> Thank you. Following your suggestion, we plan to also put plot labels close to each curve
> which should much improve differentiation (as is done already for a few
> plots) (and use different symbols when possible).

---

> > ### Comment · Reviewer_79Vu · 2026-01-16
> > **Thank you for the detailed clarifications and proposed revisions**
> >
> > I thank the authors for their detailed response and for engaging deeply with the specific points raised in my review.

---

### Decision · Action_Editor_TtiX · 2026-02-06

**Recommendation:** Accept as is

**Audience:**

Yes

**Audience Explanation:**

The paper tackles an important gap for streaming algorithms and the proposed algorithm is technically sound. It makes methodological contributions to open-ended settings, which is a setting of interest to the TMLR community. The evaluation and its somewhat particular protocol may not be up to par in terms of recency in contrast to NeurIPS/ICLR/ICML works, but some individuals may find the paper to be relevant due to its rigorous conceptual contribution.

**Claims And Evidence:**

Yes

**Claims Explanation:**

The paper has carefully revised its content based on the reviewer's feedback to provide solid empirical evidence for its claims. The proposed DYAL algorithm was initially corroborated on Unix data and the evaluation protocol has been detailed sufficiently, with extensions and relation to e.g. ADWIN having been added post rebuttal. Whereas reviewers find the significance to be modest, this fits the evaluation criteria for the paper to be accurate, clear, and convincing.